# Testing Conditional Mean Independence Using Generative Neural Networks

**Yi Zhang** [* 1]  **Linjun Huang** [* 1]  **Yun Yang** [2]  **Xiaofeng Shao** [3]

## Abstract

Conditional mean independence (CMI) testing is crucial for statistical tasks including model determination and variable importance evaluation. In this work, we introduce a novel population CMI measure and a bootstrap-based testing procedure that utilizes deep generative neural networks to estimate the conditional mean functions involved in the population measure. The test statistic is thoughtfully constructed to ensure that even slowly decaying nonparametric estimation errors do not affect the asymptotic accuracy of the test. Our approach demonstrates strong empirical performance in scenarios with high-dimensional covariates and response variable, can handle multivariate responses, and maintains nontrivial power against local alternatives outside an $n^{-1/2}$ neighborhood of the null hypothesis. We also use numerical simulations and real-world imaging data applications to highlight the efficacy and versatility of our testing procedure.

## 1. Introduction

Conditional mean independence (CMI) testing is a fundamental tool for model simplification and assessing variable importance, and plays a crucial role in statistics and machine learning. In traditional statistical applications, such as nonparametric regression, CMI testing identifies subsets or functions of covariates that meaningfully predict the response variable. This is essential for improving model efficiency, accuracy, and interpretability by avoiding redundant

*Equal contribution  [1]Department of Statistics, University of Illinois at Urbana-Champaign, Champaign, IL, USA. [2]Department of Mathematics, University of Maryland, College Park, MD, USA. [3]Department of Statistics and Data Science, and Department of Economics, Washington University in St Louis, St Louis, MO, USA. Correspondence to: Yi Zhang <yiz19@illinois.edu>, Linjun Huang <linjunh2@illinois.edu>, Yun Yang <yy84@umd.edu>, Xiaofeng Shao <shaox@wustl.edu>.

*Proceedings of the 42$^{nd}$ International Conference on Machine Learning*, Vancouver, Canada. PMLR 267, 2025. Copyright 2025 by the author(s).

variables. In machine learning, CMI testing has broad applications in areas like interpretable machine learning (Murdoch et al., 2019), representation learning (Bengio et al., 2013; Huang et al., 2024) and transfer learning (Maqsood et al., 2019; Zhuang et al., 2020).

In this paper, we address the problem of CMI testing for multivariate response variables and covariates. Specifically, for random vectors $X \in \mathbb{R}^{d_X}$, $Y \in \mathbb{R}^{d_Y}$, and $Z \in \mathbb{R}^{d_Z}$, we test the null hypothesis $H_0$ that $Y$ is conditionally mean independent of $X$ given $Z$, i.e., $\mathbb{E}[Y|X = x, Z = z] = \mathbb{E}[Y|Z = z]$, a.e. $(x, z) \in \mathbb{R}^{d_X + d_Z}$. For example, consider predicting age $Y$ from a facial image. To test whether $Y$ can be predicted using images with a specific facial region (potentially containing sensitive information) covered in black, $X$ can represent the covered region and $Z$ the remaining facial image. Similarly, to test whether $Y$ can be predicted using a low-resolution version or extracted features of a facial image, $X$ can represent the original image and $Z$ a low-dimensional feature vector derived from $X$ using standard extraction methods (e.g., Autoencoder, Average pooling, or PCA (Berahmand et al., 2024)). Under $H_0$, removing $X$ from the predictive model $f(X, Z)$, where $f : \mathbb{R}^{d_X + d_Z} \to \mathbb{R}^{d_Y}$ denotes the nonparametric regression function, does not reduce prediction accuracy.

For machine learning tasks, identifying which (functions of) covariates contribute to predicting the response is particularly important, as deep neural networks (DNNs) often process high-dimensional data, such as images and text, that may include irrelevant features (Li et al., 2017; Van Landeghem et al., 2010). Performing dimensionality reduction before DNN training provides several advantages: it enhances interpretability, improves prediction accuracy, and reduces computational costs (Cai et al., 2018). In evaluating DNN training, covariate significance is typically assessed by comparing performance metrics, such as mean squared error or classification accuracy, for DNNs trained with and without specific covariates (Dai et al., 2022; Williamson et al., 2023). As discussed in Section 2.3 of Williamson et al. (2023), many common performance metrics, such as the coefficient of determination ($R^2$) for regression and empirical risk with cross-entropy or 0-1 loss for binary classification, are functionals of the conditional mean functions of $Y$, which means $H_0$ will imply that the covariates are not significant. Thus, CMI testing provides a valuable tool for

assessing covariate importance across a variety of machine learning applications.

## 1.1. Related literature

To the best of our knowledge, most existing CMI tests focus on univariate $Y$ and face one or more of the following three major issues: (1) finite-sample performance deteriorates when some or all of $(X, Y, Z)$ are high-dimensional; (2) the tests lack theoretical size guarantees in general nonparametric settings; and (3) they exhibit weak power in detecting local alternatives. For a recent survey, see Lundborg (2022). We discuss how these challenges arise and how existing CMI tests have partially addressed them.

**Performance deterioration in high dimensional setting.** This issue primarily arises from the estimation of the conditional mean functions $r(z) := \mathbb{E}[Y|Z = z]$ and $m(x, z) := \mathbb{E}[Y|X = x, Z = z]$. Early CMI tests, such as those in Fan & Li (1996); Delgado & Manteiga (2001); Zhu & Zhu (2018); Lavergne & Vuong (2000); Aït-Sahalia et al. (2001), relied on kernel smoothing methods for these estimations. Consequently, these CMI tests suffer from the curse of dimensionality: their performance declines significantly as the dimensions $d_Z$, $d_X + d_Z$, and/or $d_Y$ increase (Zhou et al., 2022, Section 1). For instance, Figure 1 in Zhu & Zhu (2018) shows that the empirical power of tests proposed by Fan & Li (1996) and Delgado & Manteiga (2001) decreases sharply with increasing $d_Z$ and $d_X$, exhibiting trivial power under a sample size of $n = 200$ with moderate dimensionality ($d_Z = 4$ and $d_X \geq 8$). To address this, recent CMI tests utilize machine learning tools, such as deep neural networks (DNNs) and the kernel trick, to estimate conditional mean functions. These tools are effective in approximating complex, high-dimensional functions with underlying low-dimensional structures. For example, tests in Williamson et al. (2021); Dai et al. (2022); Lundborg et al. (2024); Cai et al. (2022); Williamson et al. (2023); Cai et al. (2024) leverage DNNs for conditional mean estimation, achieving better performance in high-dimensional settings.

**Theoretical size guarantee.** The main challenge in establishing a theoretical size guarantee stems from two key issues. First, most of the existing CMI tests (except the one proposed by Delgado & Manteiga (2001)) rely on the sample estimation of the population CMI measure $\Gamma := \mathbb{E}\big[(r(Z) - m(X, Z))^2 w(X, Z)\big]$ or its equivalent forms, where $w$ is a positive weight function. A common plug-in estimator of $\Gamma$ is given by $\widehat{\Gamma}(\widehat{r}, \widehat{m}) = n^{-1} \sum_{i=1}^{n} (\widehat{r}(Z_i) - \widehat{m}(X_i, Z_i))^2 w(X_i, Z_i)$, where $\widehat{r}$ and $\widehat{m}$ are estimators of the conditional mean functions. For a population CMI measure to be valid, it must uniquely characterize CMI, meaning that the measure equals zero if and only if $H_0$ holds. While $\Gamma$ satisfies this requirement, its estimator $\widehat{\Gamma}(\widehat{r}, \widehat{m})$ suffers from a degeneracy problem:

under $H_0$, $\widehat{\Gamma}(\widehat{r}, \widehat{m})$ converges to zero at a rate faster than the $n^{-1/2}$ rate at which $\widehat{\Gamma}(r, m) - \Gamma$ converges to a non-degenerate limiting distribution under the alternative (Fan & Li, 1996, Section 1).

Second, the nonparametric estimation errors for $r(z)$ and $m(x, z)$ typically decay slower than the $n^{-1/2}$ parametric rate, and the convergence rate of $\widehat{\Gamma}(\widehat{r}, \widehat{m})$ under $H_0$ depends heavily on how quickly these errors decay. For CMI tests that use kernel smoothing to estimate the conditional mean functions, the estimation error has an explicit form, allowing it to be addressed directly, along with the degeneracy issue, when deriving asymptotic results under $H_0$. As a result, all the aforementioned kernel smoothing-based CMI tests have theoretical size guarantees. In contrast, due to the black-box nature of DNNs, the estimation errors for $r(z)$ and $m(x, z)$ cannot be explicitly decomposed or handled in the same way as kernel smoothing estimators. Consequently, addressing the degeneracy issue requires additional debiasing procedures to mitigate the impact of these errors and achieve accurate size control. For example, Williamson et al. (2021) and Lundborg et al. (2024) constructed statistics based on alternative forms of $\Gamma$ to reduce bias. However, the degeneracy issue persists: as shown in Theorem 1 of Williamson et al. (2021), their estimator is $\sqrt{n}$-consistent when $\Gamma > 0$, but no asymptotic results are provided under $H_0$ (i.e., $\Gamma = 0$). Similarly, the test in Lundborg et al. (2024), which builds on the conditional independence test from Shah & Peters (2020), relies on strong assumptions to bypass the degeneracy issue, as outlined in Section 2.1 and Part (a) of Theorem 4 in Lundborg et al. (2024). To address the degeneracy issue, Dai et al. (2022) introduced additional noise of order $O_p(n^{-1/2})$ to the estimator of $\Gamma$, while Williamson et al. (2023) utilized sample splitting to estimate separate components of an equivalent form of $\Gamma$ on different subsamples. However, both approaches are ad hoc (Section 6.2 of Verdinelli & Wasserman (2024)) and, as discussed in Appendix S3 of Lundborg et al. (2024), suffer from significant power loss under the alternative hypothesis.

**Weak power against local alternatives.** On one hand, the CMI tests proposed in Fan & Li (1996); Zhu & Zhu (2018); Lavergne & Vuong (2000); Aït-Sahalia et al. (2001); Williamson et al. (2021); Dai et al. (2022); Williamson et al. (2023) fail to detect local alternatives with signal strength $\Delta_n := \sqrt{\mathbb{E}[(r(Z) - m(X, Z))^2]}$ of order $n^{-1/2}$, primarily due to their reliance on the population measure $\Gamma$. Specifically, the tests in Fan & Li (1996); Zhu & Zhu (2018); Lavergne & Vuong (2000); Aït-Sahalia et al. (2001) take the form $n h^{s/2} \widehat{\Gamma}$, where $h \to 0$ is a bandwidth parameter used in kernel smoothing, and $s = d_Z$ or $d_X + d_Z$. Consequently, these tests cannot detect local alternatives converging to the null faster than $n^{-1/2} h^{-s/4}$. The tests in Williamson et al. (2021); Dai et al. (2022); Williamson et al. (2023) use the population CMI measure $\Gamma_0 = \Gamma_1 - \Gamma_2$,

*Table 1.* Summary of existing CMI tests. High-Dim: whether the test has good empirical performance when some or all the dimensions of $X$, $Y$ and $Z$ are large; Size Guarantee: whether the test has theoretical results under $H_0$ that guarantee accurate asymptotic size control in general nonparametric setting; Local Alt: whether the test can detect local alternatives with signal strength $\Delta_n$ converging to zero at the parametric rate $n^{-1/2}$.

| Tests | High-Dim | Size Guarantee | Local Alt |
|---|---|---|---|
| Fan & Li (1996) | No | Yes | No |
| Delgado & Manteiga (2001) | No | Yes | Yes |
| Zhu & Zhu (2018) | No | Yes | No |
| Lavergne & Vuong (2000) | No | Yes | No |
| Ait-Sahalia et al. (2001) | No | Yes | No |
| Williamson et al. (2021) | Yes | No | No |
| Dai et al. (2022) | Yes | Yes | No |
| Lundborg et al. (2024) | Yes | No | Yes |
| Williamson et al. (2023) | Yes | Yes | No |
| Cai et al. (2024) | Yes | Yes | No |
| **Our method** | **Yes** | **Yes** | **Yes** |

where $\Gamma_1 = \mathbb{E}[(Y - r(Z))^2]$ and $\Gamma_2 = \mathbb{E}[(Y - m(X, Z))^2]$, which is equivalent to $\Gamma$. Since the quadratic terms $\Gamma_1$ and $\Gamma_2$ can only be estimated at the $n^{-1/2}$ rate, these tests are limited to detecting local alternatives with $\Delta_n$ of order $n^{-1/4}$. On the other hand, the test proposed in Cai et al. (2024) employs sample splitting and requires the size of the training subsample used to estimate the conditional mean functions to be substantially larger than that of the testing subsample. As a result, their test is limited to detecting local alternatives with $\Delta_n$ converging to zero slower than $n^{-1/2}$, which can still result in significant power loss in practice.

## 1.2. Our contributions

Table 1 summarizes limitations of existing CMI tests. To overcome these challenges, we propose a novel CMI testing procedure with the following advantages:

1. The test demonstrates strong empirical performance even when some or all dimensions of $X$, $Y$, and $Z$ are high. Notably, it is well-suited for scenarios where imaging data serve as covariates, responses, or both.

2. The test achieves precise asymptotic size control under $H_0$.

3. The test exhibits nontrivial power against local alternatives outside an $n^{-1/2}$-neighborhood of $H_0$.

To achieve these features, we propose a new population CMI measure closely related to the conditional independence measure introduced in Daudin (1980). Additionally, we develop a sample version of this population measure in a multiplicative form, which is key to mitigating the impact of estimation errors in nonparametric nuisance parameters (i.e., the conditional mean functions). Our test not only requires estimating $r(z)$ but also the conditional mean embedding (CME, Song et al., 2009, Definition 3) of $X$ given $Z$ into a reproducing kernel Hilbert space (RKHS) on the space of $X$. Instead of directly estimating the CME using DNNs, we

train a generative neural network (GNN) to sample from the (approximated) conditional distribution of $X$ given $Z$. The CME is then estimated using the Monte Carlo method with samples generated from the trained GNN.

## 1.3. Organization and notations

The paper is organized as follows: Section 2 introduces the proposed population CMI measure, the test statistic, and the bootstrap calibration procedure, along with its asymptotic properties and consistency results. Section 3 evaluates the test using finite sample simulations, comparing it with other methods. Section 4 presents two real data examples. Section 5 concludes with final remarks. All proofs and additional details are deferred to the Appendix. A Python implementation of our proposed test procedure is available at https://github.com/LinjunHuang86749/Testing-CMI-Using-Generative-NN.

The following notations will be used throughout the paper. For any positive integer $d$ and random vectors $(X^1, X^2, \ldots, X^d, Z)$ defined on the same probability space, $\mathbb{P}_{X^1 X^2 \cdots X^d}$ and $\mathbb{P}_{X^1 X^2 \cdots X^d | Z}$ denote the joint distribution of $X^1, X^2, \ldots, X^d$ and its conditional distribution given $Z$, respectively. Let $\mathbb{E}_Z$ represent the expectation with respect to $\mathbb{P}_Z$, and let $\mathbb{P}_m$ denote the Lebesgue measure on $\mathbb{R}^d$. For a positive integer $n$, define $[n] = \{1, 2, \ldots, n\}$. For a probability measure $\mu(\cdot)$ on $\mathbb{R}^d$ and $p \geq 1$, let $L_p(\mathbb{R}^d, \mu) = \{f : \mathbb{R}^d \to \mathbb{R} : \int |f(x)|^p d\mu(x) < \infty\}$. For any Hilbert spaces $A$ and $B$, let $A \otimes B$ denote their tensor product, and use $\langle \cdot, \cdot \rangle_A$ and $\| \cdot \|_A$ to denote the inner product and induced norm on $A$, respectively. For random vectors $a, b \in \mathbb{R}^d$, the Gaussian kernel is defined as $\mathcal{K}(a, b) = \exp(-\|a - b\|_2^2/(2\sigma^2))$ and the Laplacian kernel as $\mathcal{K}(a, b) = \exp(-\|a - b\|_1/\sigma)$, where $\sigma > 0$ is the bandwidth parameter, $\| \cdot \|_2$ is the Euclidean $\ell_2$-norm, and $\| \cdot \|_1$ is the $\ell_1$-norm.

## 2. Conditional Mean Independence Testing

In this section, we introduce a novel population measure for evaluating CMI and propose a corresponding sample-based statistic for conducting the CMI test. We then establish theoretical guarantees for the proposed procedure, including size control and power against local alternatives.

### 2.1. Population conditional mean independence measure

Recall that the goal is to test the null hypothesis of $H_0$ : $\mathbb{E}[Y|X, Z] = \mathbb{E}[Y|Z]$ a.s.-$\mathbb{P}_{XZ}$ against the alternative hypothesis of $H_1 : \mathbb{P}\big(\mathbb{E}[Y|X, Z] \neq \mathbb{E}[Y|Z]\big) > 0$. To motivate our population measure for evaluating CMI, we begin with the following result, which provides equivalent characterizations of CMI (Daudin, 1980).

**Proposition 1.** *If* $\mathbb{E}[\|Y\|_2^2] < \infty$, *then the following properties are equivalent to each other:*

*(a)* $\mathbb{E}[Y|X, Z] = \mathbb{E}[Y|Z]$ *a.s.-*$\mathbb{P}_{XZ}$.

*(b)* $\mathbb{E}\big[\big(f(X, Z) - \mathbb{E}[f(X, Z)|Z]\big) Y\big] = 0$ *for any* $f \in$

$L_2(\mathbb{R}^{d_X+d_Z}, \mathbb{P}_{XZ})$.

(c) $\mathbb{E}\big[\big(f(X,Z) - \mathbb{E}[f(X,Z)|Z]\big)\big(Y - \mathbb{E}[Y|Z]\big)\big] = 0$ for any $f \in L_2(\mathbb{R}^{d_X+d_Z}, \mathbb{P}_{XZ})$.

*Remark* 2. *It is straightforward to see that (a) $\Rightarrow$ (b) $\Rightarrow$ (c), and (c) implies (a) by taking $f(X,Z) = \mathbb{E}[Y^\top c \,|\, X, Z]$ over all $c \in \mathbb{R}^{d_Y}$. The only difference between (c) and (b) in Proposition 1 is that $Y$ is centered at $\mathbb{E}[Y|Z]$ in (c). This additional centering is crucial for reducing biases from the estimation of the conditional mean functions; our proposed population CMI measure will be derived from (c) by considering all $f$ within a dense subset of $L_2(\mathbb{R}^{d_X+d_Z}, \mathbb{P}_{XZ})$.*

Let $\mathcal{K}_X : \mathbb{R}^{d_X} \times \mathbb{R}^{d_X}$ and $\mathcal{K}_Z : \mathbb{R}^{d_Z} \times \mathbb{R}^{d_Z}$ denote two symmetric positive-definite kernel functions that define two reproducing kernel Hilbert spaces (RKHS) $\mathbb{H}_X$ and $\mathbb{H}_Z$ over the spaces of $X$ and $Z$, respectively. Additionally, let $\mathcal{K}_0 = \mathcal{K}_X \times \mathcal{K}_Z$ represent the product kernel, with $\mathbb{H}_0$ being the corresponding RKHS induced by $\mathcal{K}_0$ over the product space $\mathbb{R}^{d_X} \times \mathbb{R}^{d_Z}$. Motivated by part (c) of Proposition 1, we define linear operator $\Sigma : \mathbb{R}^{d_Y} \to \mathbb{H}_0$,

$$\Sigma c = \mathbb{E}\Big\{ \big[\mathcal{K}_0((X,Z), \cdot) - \mathbb{E}[\mathcal{K}_0((X,Z), \cdot)|Z]\big]$$
$$\big[Y - \mathbb{E}[Y|Z]\big]^\top c\Big\}, \quad \text{for any } c \in \mathbb{R}^{d_Y}.$$

From the reproducing property, we see that for any $f \in \mathbb{H}_0$ and any $c \in \mathbb{R}^{d_Y}$,

$$\langle f, \Sigma c\rangle_{\mathbb{H}_0} = \mathbb{E}\Big\{ \big[f(X,Z) - \mathbb{E}[f(X,Z)|Z]\big]\big[Y - \mathbb{E}[Y|Z]\big]^\top c\Big\}.$$

Under the assumption that the RKHS $\mathbb{H}_0$ is dense in $L_2(\mathbb{R}^{d_X+d_Z}, P_{XZ})$, which holds if $\mathcal{K}_X$ and $\mathcal{K}_Z$ are $L_2$- or $c_0$-universal kernels (Szabó & Sriperumbudur, 2018, Theorem 5), such as the Gaussian and Laplacian kernels considered in this paper, the preceding display along with Proposition 1 implies that the null hypothesis $H_0$ holds if and only if $\Sigma$ is the zero operator (i.e., $\Sigma c = 0 \in \mathbb{H}_0$ for any $c \in \mathbb{R}^{d_Y}$). The following proposition formalizes this intuition and serves as the foundation for our proposed population CMI measure,

$$\Gamma^* = \mathbb{E}\big[U(X, X')\, V(Y, Y')\, \mathcal{K}_Z(Z, Z')\big], \quad (1)$$

where $V(Y, Y') = \big[Y - g_Y(Z)\big]^\top \big[Y' - g_Y(Z')\big]$, and

$$U(X, X') = \mathcal{K}_X(X, X') - \langle g_X(Z), \mathcal{K}_X(X', \cdot)\rangle_{\mathbb{H}_X}$$
$$- \langle g_X(Z'), \mathcal{K}_X(X, \cdot)\rangle_{\mathbb{H}_X} + \langle g_X(Z), g_X(Z')\rangle_{\mathbb{H}_X}.$$

Here, $(X', Y', Z')$ is an independent copy of $(X, Y, Z)$, $g_Y$ and $g_X$ are defined as $g_Y(\cdot) = \mathbb{E}[Y|Z = \cdot] \in \mathbb{R}^{d_Y}$ and $g_X(\cdot) = \mathbb{E}[\mathcal{K}_X(X, \cdot) \,|\, Z = \cdot] \in \mathbb{H}_X$, respectively.

**Proposition 3.** *If Assumption 4(a) holds, then (see Definition 11 in Appendix C for the definitions of the Hilbert-Schmidt operator and norm)*

(a) *$\Sigma$ is a Hilbert-Schmidt operator, and its Hilbert-Schmidt norm, denoted as $\|\Sigma\|_{\mathrm{HS}}$, satisfies $\|\Sigma\|_{\mathrm{HS}}^2 = \Gamma^*$.*

(b) *The null $H_0$ holds if and only if $\Gamma^* = 0$.*

Due to the multiplicative form inside the expectation in equation (1), when constructing a sample version of $\Gamma^*$ as the test statistic, the estimation errors of the two nuisance parameters ($g_X$ and $g_Y$) do not affect the asymptotic properties of the statistic, as long as the product of these errors decays faster than $n^{-1/2}$; see Lemma 17 in Appendix E. This desirable property is commonly referred to as double robustness (Bang & Robins, 2005; Zhang et al., 2024).

Our $\Gamma^*$ bears similarity to the maximum mean discrepancy-based conditional independence measure (MMDCI) proposed in Zhang et al. (2024), which was designed for testing the stronger null hypothesis of conditional independence (CI). While CI assumes that the conditional distribution of the response variable $Y$ is independent of an additional covariate $X$ given an existing covariate $Z$, the CMI assumption only requires that the conditional mean function of $Y$ does not depend on $X$ given $Z$. This means that including $X$ in the regression function of $Y$ does not improve its predictive ability in the mean squared error sense. In contrast, rejecting CI does not provide a direct interpretation in terms of predictive ability (Lundborg et al., 2024, Section 1). Furthermore, the MMDCI statistic compares the joint distribution of $(X, Y, Z)$ under the null with its true distribution using maximum mean discrepancy (MMD) (Gretton et al., 2012), requiring stronger assumptions to fully characterize CI (see Assumption 1 in Zhang et al. (2024)). In contrast, our population CMI measure avoids using MMD to capture the full distributional properties of $(X, Y, Z)$, making it more suitable for CMI testing compared to existing population CMI measures. Computationally, the test proposed in Zhang et al. (2024) requires training two generative neural networks (GNNs) to sample from the conditional distributions of $X$ and $Y$ given $Z$, respectively. In contrast, our proposed test statistic only requires sampling from the conditional distribution of $X$ given $Z$, resulting in lower computational cost and reduced memory usage.

### 2.2. Testing procedure

To construct a sample version of $\Gamma^*$, it is necessary to estimate $g_Y$ and $g_X$. To address the curse of dimensionality, we use deep neural networks (DNNs) to estimate $g_Y$. For $g_X$, which is an RKHS-valued function, we estimate $g_X(z)$ for a fixed $z \in \mathbb{R}^{d_Z}$ by sampling $M$ copies $\{X_{z,i}\}_{i=1}^M$ from the conditional distribution of $X$ given $Z = z$. The estimate of $g_X(z)$ is then given by the sample average $M^{-1} \sum_{i=1}^M \mathcal{K}_X(X_{z,i}, \cdot) \in \mathcal{H}_X$. Specifically, using the noise-outsourcing lemma (Theorem 6.10 of Kallenberg (2002); see also Lemma 2.1 of Zhou et al. (2022)), for any integer $m \geq 1$, there exists a measurable function $\mathbb{G}$ such that, for any $\eta \sim N(0, I_m)$ independent of $Z$, we have $\mathbb{G}(\eta, Z) \,|\, Z \sim P_{X|Z}$. Based on this result, we train a generative moment matching network (GMMN, Dziugaite et al., 2015; Li et al., 2015) to approximate $\mathbb{G}$. For any fixed $z \in \mathbb{R}^{d_Z}$, we generate i.i.d. copies $\eta_i$ from $N(0, I_m)$,

input them into the trained GMMN $\widehat{\mathbb{G}}$ along with $z$, and treat the outputs $\widehat{\mathbb{G}}(\eta_i, z)$ as approximately sampled from the conditional distribution of $X$ given $Z = z$. We refer to the trained GMMN as the (conditional) generator, and detailed training procedures are provided in Appendix A. Note that other generative models, such as generative adversarial networks (GAN) (Goodfellow et al., 2014) and diffusion models (Yang et al., 2023), can also be applied to approximate $\mathbb{G}$, as long as the approximation errors satisfy mild decaying conditions. The field of generative modeling has recently seen significant advances, with notable examples including Athey et al. (2024); Baptista et al. (2020); Shi et al. (2022); Nguyen et al. (2024). Incorporating these newly developed models into our testing procedure is straightforward and may potentially improve both the computational cost and the performance of the proposed test.

In order to eliminate dependence on conditional generator estimation and improve test size accuracy, we follow Shi et al. (2021) and adopt a sample-splitting and cross-fitting framework to train the GMMN and DNN. For easy presentation, we consider two-fold splitting in this paper, although the test can be readily generalized to the multiple-fold splitting setting. Specifically, for positive integers $n$, $M$ and $m \in [M]$, let $\{(X_i, Y_i, Z_i)\}_{i=1}^n$ and $\{\eta_i^m\}_{i=1}^n$ be i.i.d. copies of $(X, Y, Z)$ and $\eta$ such that $\{\{(X_i, Y_i, Z_i)\}_{i=1}^n, \{\eta_i^1\}_{i=1}^n, \dots, \{\eta_i^m\}_{i=1}^n\}$ are mutually independent. Let $\widetilde{X}_i^{(m)} = \mathbb{G}(\eta_i^m, Z_i)$ denote the data sampled from $P_{X_i|Z_i}$. We divide $[n]$ into two equal folds $\mathcal{J}^{(1)}$ and $\mathcal{J}^{(2)}$ where $\mathcal{J}^{(1)} := \mathcal{J}^{(-2)} = \{1, 2, \dots, \lfloor n/2 \rfloor\}$ and $\mathcal{J}^{(2)} := \mathcal{J}^{(-1)} = [n]/\mathcal{J}^{(1)}$. For $j \in [2]$, we train a GMMN generator $\widehat{\mathbb{G}}_j$ using data $(X_i, Z_i)$ for $i \in \mathcal{J}^{(-j)}$. Similarly, we train a DNN $\widehat{g}_j$ using data $(Y_i, Z_i)$ for $i \in \mathcal{J}^{(-j)}$ as an estimator of $g_Y$ (four neural networks are trained in total). Let $\widehat{X}_i^{(m)} = \widehat{\mathbb{G}}_j(\eta_i^m, Z_i)$ and $\widehat{g}_Y(Z_i) = \widehat{g}_j(Z_i)$ if $i \in \mathcal{J}^{(j)}$, then we define the test statistic as

$$\widehat{T}_n = \frac{1}{2} \sum_{s=1}^2 \left[ \frac{1}{\frac{n}{2}(\frac{n}{2}-1)} \sum_{\substack{j \neq k \\ j,k \in \mathcal{J}^{(s)}}} \widehat{U}(X_j, X_k) \widehat{V}(Y_j, Y_k) \mathcal{K}_Z(Z_j, Z_k) \right],$$

where $\widehat{U}(X_j, X_k) = \mathcal{K}_X(X_j, X_k) - \frac{1}{M} \sum_{m=1}^M \mathcal{K}_X(X_j, \widehat{X}_k^{(m)})$

$$- \frac{1}{M} \sum_{m=1}^M \mathcal{K}_X(X_k, \widehat{X}_j^{(m)}) + \frac{1}{M} \sum_{m=1}^M \mathcal{K}_X(\widehat{X}_j^{(m)}, \widehat{X}_k^{(m)}),$$

and $\widehat{V}(Y_j, Y_k) = [Y_j - \widehat{g}_Y(Z_j)]^\top [Y_k - \widehat{g}_Y(Z_k)]$.

Given the trained neural networks and assuming Gaussian or Laplacian kernel functions are used, $\widehat{T}_n$ resembles the average of two U-statistics of degree two, with its value depending on pairwise distances between samples. As a result, the computational complexity scales linearly with the dimensions $(d_X + d_Y + d_Z)$ and quadratically with the sample size $n$. In comparison, the computational complexity of the statistics proposed in Cai et al. (2024) and Dai et al. (2022), both designed for univariate $Y$, scales linearly with $n$. As will be shown in Theorem 5 below, the limiting null distribution of $n\widehat{T}_n$ depends on the unknown distribution

$P_{XYZ}$. Consequently, we cannot directly determine the rejection threshold of $n\widehat{T}_n$ without knowing $P_{XYZ}$. Instead, we employ a wild bootstrap method to approximate the distribution of $n\widehat{T}_n$. Following Section 2.4 of Zhang et al. (2018), for a positive integer $B$ and $b \in [B]$, we generate $\{e_{bi}\}_{i=1}^n$ from a Rademacher distribution, and define the bootstrap version of $\widehat{T}_n$ as

$$\widehat{M}_n^b = \frac{1}{2} \sum_{s=1}^2 \left\{ \frac{1}{\frac{n}{2}(\frac{n}{2}-1)} \sum_{\substack{j \neq k \\ j,k \in \mathcal{J}^{(s)}}} \widehat{U}(X_j, X_k) \widehat{V}(Y_j, Y_k) \right.$$
$$\left. \cdot \mathcal{K}_Z(Z_j, Z_k) e_{bj} e_{bk} \right\}. \quad (2)$$

Since $\{\widehat{M}_n^1, \widehat{M}_n^2, \dots, \widehat{M}_n^B\}$ can be viewed as samples from the distribution of $\widehat{T}_n$ (c.f. Theorem 9), we reject $H_0$ at level $\gamma \in (0, 1)$ if $\frac{1}{B} \sum_{b=1}^B \mathbb{1}_{\{n\widehat{M}_n^b > n\widehat{T}_n\}} < \gamma$. As a default choice, we set $B = 1,000$ throughout our numerical experiments.

### 2.3. Theoretical properties

In this part, we evaluate the asymptotic performance of the test as the sample size increases, focusing on its empirical size (Type-I error) control and power against (local) alternatives. Specifically, we aim to determine whether the proposed test satisfies two desirable theoretical properties: 1. the probability of incorrectly rejecting the null hypothesis converges to the specified significance level $\gamma$ as the sample size grows; 2. the test maintains the capability to detect (local) alternative hypotheses with a deviation from null that diminishes at the parametric rate of $n^{-1/2}$.

From our procedure, we define the event $\text{Rej}_n := \{B^{-1} \sum_{b=1}^B \mathbb{1}_{\{n\widehat{M}_n^b > n\widehat{T}_n\}} < \gamma\}$ as rejecting $H_0$. To demonstrate that our proposed test has an accurate asymptotic size, it suffices to prove that the probability of rejecting $H_0$, given that $H_0$ is true, converges to the specified level $\gamma$, i.e., $\lim_{n \to \infty} \mathbb{P}(\text{Rej}_n | H_0) = \gamma$. To achieve this, we first derive the limiting null distribution of $n\widehat{T}_n$ in Theorem 5. Subsequently, we show the consistency of our bootstrap procedure in Theorem 9, that is, conditional on the sample, the rescaled bootstrap statistic converges to the same limiting null distribution. To study the asymptotic power of our proposed test, we define a local alternative hypothesis in equation (4) whose deviation from null scales as $n^{-\alpha}$ for some $\alpha \geq 0$. The asymptotic behavior of $\widehat{T}_n$ and the bootstrap counterpart under different values of $\alpha$ is analyzed in Theorem 8 and Theorem 9, respectively. Based on these results, we derive the asymptotic power of the test in Theorem 9. Let $T_n$ denote the oracle test statistic, defined similarly to $\widehat{T}_n$, except that the estimated nuisance parameters $(g_Y, g_X)$ are replaced with their true values; see equation (14) in Appendix D for the precise definition.

The following assumption is needed to derive the asymptotic properties of our statistic.

**Assumption 4.** Assume $M \to \infty$ as $n \to \infty$. For $C_0 > 0$, $\alpha_1, \alpha_2 \in (0, \frac{1}{2})$ such that $\alpha_1 + \alpha_2 > \frac{1}{2}$, $D_i \in$

$\{X_i, \widetilde{X}_i^{(1)}, \widehat{X}_i^{(1)}\}$, $E_i \in \{Y_i, g_Y(Z_i), \widehat{g}_Y(Z_i)\}$ and $i \in \{i_1, i_2\}$ where $i_s \in \mathcal{J}^{(s)}$ for each $s \in [2]$, we have:

(a) $\mathbb{E}\left\{\mathcal{K}_X(D_i, D_i) \|E_i\|_2^2 \mathcal{K}_Z(Z_i, Z_i)\right\} < C_0$;

(b) $\left[\mathbb{E}\left\{\|\mathbb{E}[\mathcal{K}_X(\cdot, X_i)|Z_i] - \mathbb{E}[\mathcal{K}_X(\cdot, \widehat{X}_i^{(1)})|Z_i]\|_{\mathbb{H}_X}^2 \left[\sqrt{\mathcal{K}_Z(Z_i, Z_i)}\right.\right.\right.$
$\left.\left.\left. + \|E_i\|_2^2 \mathcal{K}_Z(Z_i, Z_i)\right]\right\}\right]^{1/2} = O(n^{-\alpha_1})$;

(c) $\left[\mathbb{E}\left\{\|g_Y(Z_i) - \widehat{g}_Y(Z_i)\|_2^2 \left[\sqrt{\mathcal{K}_Z(Z_i, Z_i)} + \mathcal{K}_X(D_i, D_i)\right.\right.\right.$
$\left.\left.\left. \mathcal{K}_Z(Z_i, Z_i)\right]\right\}\right]^{1/2} = O(n^{-\alpha_2})$.

It it worth noting that the required technical assumptions are relatively mild and are commonly used in the literature of CMI testing. Part (a) of Assumption 4 ensures that the (conditional) mean embeddings into the RKHSs, as well as the operator $\Sigma$, are well-defined and holds for bounded kernels such as the Gaussian and Laplacian kernels. Part (b) of Assumption 4 requires the estimation errors of the neural networks to decay to zero at rates $n^{-\alpha_1}$ and $n^{-\alpha_2}$ for $\alpha_1, \alpha_2 \in (0, \infty)$ such that $\alpha_1 + \alpha_2 > 1/2$. Similar rate requirements appear in Cai et al. (2024) and Lundborg et al. (2024).

When estimating $g_Y$ with DNNs, Bauer & Kohler (2019) demonstrated that the decay rate for $\mathbb{E}\left[\|g_Y(Z) - \widehat{g}_Y(Z)\|_2^2\right]$ can be bounded from above by $n^{-2s/(2s+d^*)}$, where $s$ is the smoothness parameter of $g_Y$, and $d^*$ denotes its intrinsic dimensionality. Regarding the estimation error of the conditional mean embedding of $P_{X|Z}$, our requirement is less restrictive than the assumptions based on the total variation distance between $P_{X|Z}$ and its estimator, which are commonly adopted in the literature.

Specifically, if $\mathcal{K}_X$, $\mathcal{K}_Z$ are bounded kernels and $\|Y\|_2^2$ is bounded (without loss of generality, assume they are bounded by 1), then Assumption 4 reduces to $\mathbb{E}\left[\|\mathbb{E}[\mathcal{K}_X(\cdot, X_i) \mid Z_i] - \mathbb{E}[\mathcal{K}_X(\cdot, \widehat{X}_i^{(1)}) \mid Z_i]\|_{\mathcal{H}_X}^2\right] = O(n^{-2\alpha_1})$ and $\mathbb{E}\left[\|g_Y(Z_i) - \widehat{g}_Y(Z_i)\|_2^2\right] = O(n^{-2\alpha_2})$. Note that

$$\mathbb{E}\left[\|\mathbb{E}[\mathcal{K}_X(\cdot, X_i) \mid Z_i] - \mathbb{E}[\mathcal{K}_X(\cdot, \widehat{X}_i^{(1)}) \mid Z_i]\|_{\mathcal{H}_X}^2\right]$$
$$= \mathbb{E}\left[\left\{\sup_{f \in \mathcal{H}_X: \|f\|_{\mathcal{H}_X} \le 1} \mathbb{E}\left[f(\widehat{X}_i^{(1)}) - f(X_i) \mid Z_i\right]\right\}^2\right]$$
$$\le \mathbb{E}\left[\left\{\sup_{f: \mathbb{R}^{d_X} \to \mathbb{R}: \|f\|_\infty \le 1} \mathbb{E}\left[f(\widehat{X}_i^{(1)}) - f(X_i) \mid Z_i\right]\right\}^2\right]$$
$$= 2\,\mathbb{E}\left[d_{\text{TV}}^2(P_{\widehat{X}_i^{(1)}|Z_i}, P_{X_i|Z_i})\right], \qquad (3)$$

where $\|\cdot\|_\infty$ denotes the function supreme norm, and $d_{\text{TV}}(\cdot, \cdot)$ denotes the total variation distance. Here, the inequality in the second line is implied by the fact that $\|f\|_\infty = \sup_{x \in \mathbb{R}^{d_X}} |f(x)| = \sup_{x \in \mathbb{R}^{d_X}} |\langle f, \mathcal{K}_X(x, \cdot)\rangle_{\mathcal{H}_X}| \le \|f\|_{\mathcal{H}_X} \sqrt{\mathcal{K}_X(x, x)} \le \|f\|_{\mathcal{H}_X}$. Therefore, we can also replace the error metrics in

Assumption 4 by the total variation distance and the mean squared error, i.e., $\mathbb{E}\left[d_{\text{TV}}^2(P_{\widehat{X}_i^{(1)}|Z_i}, P_{X_i|Z_i})\right] = O(n^{-2\alpha_1})$ and $\mathbb{E}\left[\|g_Y(Z_i) - \widehat{g}_Y(Z_i)\|_2^2\right] = O(n^{-2\alpha_2})$, which are common assumptions made in existing works for characterizing qualities of conditional generators and nonparametric regression functions. However, the total variation metric may not be a suitable metric for characterizing the closeness between nearly mutually singular distributions, which happens when data are complex objects such as images or texts exhibiting low-dimensional manifold structures (Tang & Yang, 2023). Moreover, due to the double robustness property of our test statistic, we do not impose separate constraints on the respective estimation errors of $g_Y$ and the mean embedding of $P_{X|Z}$. Instead, we only require that their product decays faster than $n^{-1/2}$. Notably, when $g_Y$ is sufficiently smooth, $\alpha_2$ can approach $1/2$, allowing $\alpha_1$ to remain very small to accommodate complex and high-dimensional distributions of $P_{X|Z}$ (e.g., when both $X$ and $Z$ are images).

**Theorem 5.** *Suppose Assumption 4 holds, then under $H_0$, we have that $\widehat{T}_n - T_n = O_p\left(n^{-1}[M^{-1/2} + n^{-\alpha_1} + n^{-\alpha_2}] + n^{-1/2 - (\alpha_1 + \alpha_2)}\right)$ as $n \to \infty$; moreover, $n\widehat{T}_n \xrightarrow{D} T^\dagger = T_1^\dagger + T_2^\dagger$, where $\{T_1^\dagger, T_2^\dagger\}$ are i.i.d. random variables following a mixture of centered chi-squares distribution (see Appendix D.1).*

From this theorem, we observe that the "plugged-in" test statistic $\widehat{T}_n$ becomes asymptotically equivalent to the oracle statistic $T_n$ as long as $\alpha_1 + \alpha_2 > 1/2$. This demonstrates the property of *double robustness*: the slowly decaying nonparametric estimation errors in $(g_Y, g_X)$ do not compromise the asymptotic accuracy of the test, provided the product of these errors decays faster than $n^{-1/2}$.

Now let us switch to the asymptotic power of the test under local alternatives. We use the triple $(X^0, Y^0, Z^0)$ to denote $(X, Y, Z)$ under $H_0$ and consider a sequence of triples $(X^0, Y_n^A, Z^0)$ under the alternative hypothesis:

$$H_{1n}: Y_n^A = \mathbb{E}[Y^0|Z^0] + n^{-\alpha}\mathcal{G}(X^0, Z^0) + \mathcal{R}_n. \quad (4)$$

Here, the $\mathbb{R}^{d_Y}$-valued function $\mathcal{G}$ and the random vector $\mathcal{R}_n$ satisfy $\mathbb{E}[\mathcal{G}(X^0, Z^0)|Z^0] = 0$ and $\mathbb{E}[\mathcal{R}_n|X^0, Z^0] = 0$, respectively, so that the exponent $\alpha \ge 0$ determines the decay rate of the deviation from null under $H_{1n}$; for instance, setting $\alpha = 0$ and $\mathcal{R}_n \equiv \mathcal{R}$ corresponds to a fixed alternative. The following assumption is needed to derive the asymptotic results under $H_{1n}$ when $\alpha > 0$.

**Assumption 6.** For $D \in \{\mathcal{K}_X(X^0, \cdot), g_X(Z^0)\}$, there exists a random variable $\zeta \in \mathbb{R}^{d_Y}$ such that $\mathbb{E}\left\{\|D\|_{\mathbb{H}_X}^2 \|\mathcal{R}_n - \zeta\|_2^2 \mathcal{K}_Z(Z^0, Z^0)\right\} \to 0$.

*Remark 7. Assumption 6 implies that when $\alpha > 0$, the demeaned random vector $\mathcal{R}_n = Y_n^A - \mathbb{E}[Y_n^A|X^0, Z^0]$ converges to some random vector $\zeta$. Instead of fixing $\mathcal{R}_n = Y^0 - \mathbb{E}[Y^0|Z^0]$ as in nonparametric regression models (see equation (1.1) in Zhu & Zhu (2018)), we allow $\mathcal{R}_n$*

*to change with $n$ and $\zeta$ can be different from $Y^0 - \mathbb{E}[Y^0|Z^0]$.*

**Theorem 8.** *Suppose Assumptions 4 holds, then under $H_{1n}$ (see Appendix D.2 for precise definitions of relevant constants and random variables),*

1. *If $\alpha = 0$, then $\sqrt{n}(\widehat{T}_n - c_0) \xrightarrow{D} \frac{1}{\sqrt{2}} \sum_{j=1}^{2} \mathcal{G}_j^{(0)}$ as $n \to \infty$ for some positive constant $c_0$ and i.i.d. mean zero normal random variables $\{\mathcal{G}_1^{(0)}, \mathcal{G}_2^{(0)}\}$;*

*With Assumption 6 further satisfied, we have*

2. *If $0 < \alpha < 1/2$, then $n^{2\alpha}\widehat{T}_n \xrightarrow{p} c$ for some $c > 0$;*

3. *If $\alpha = 1/2$, then $n\widehat{T}_n \xrightarrow{D} c + T_A^\dagger + \frac{1}{\sqrt{2}} \sum_{j=1}^{2} \mathcal{G}_j$, where $T_A^\dagger = \sum_{j=1}^{2} T_{Aj}^\dagger$ and $\{T_{A1}^\dagger, T_{A2}^\dagger\}$ are i.i.d. random variables following a mixture of centered chi-squares distribution.*

4. *If $\alpha > 1/2$, then $n\widehat{T}_n \xrightarrow{D} T_A^\dagger$.*

Theorem 5 establishes that rejecting $H_0$ if $n\widehat{T}_n$ exceeds the rejection threshold as the $(1 - \gamma)$th quantile of the limiting distribution of $T^\dagger$ constitutes a valid test procedure for $H_0$ with an asymptotic size of $\gamma$ for any $\gamma \in (0, 1)$. Additionally, items 1 and 2 of Theorem 8 imply that $n\widehat{T}_n$ diverges to infinity under $H_{1n}$ if the deviation from the null hypothesis decays slower than $n^{-1/2}$, ensuring the power of this test procedure to approach one as $n \to \infty$. Together, these properties imply the minimax-optimality of the test procedure based on the test statistic $\widehat{T}_n$, provided the rejection threshold can be computed. On the other hand, items 3 and 4 of Theorem 8 show that the test has trivial power in detecting alternatives that are too close to the null. This is expected, as even for a parametric linear model, a meaningful CMI test cannot detect local alternatives with deviations from the null decaying faster than the $n^{-1/2}$ parametric rate (e.g., see Section 1.1 of Lundborg et al. (2024)).

Finally, we demonstrate that our proposed test in Section 2.2 based on bootstrap eliminates the need to compute the rejection threshold, and the resulting test procedure is asymptotically indistinguishable from the optimal test based on $n\widehat{T}_n$. For a generic bootstrapped statistic $\widehat{M}_n^M$ defined according to equation (2) with $\{e_{bi}\}_{i=1}^{n}$ replaced by independent sample $\{e_i\}_{i=1}^{n}$ from standard normal distribution, we say that $\widehat{M}_n^M$ converges in distribution in probability to a random variable $B^*$ if, for any subsequence $\widehat{M}_{n_k}^{M_k}$, there exists a further subsequence $\widehat{M}_{n_{k_j}}^{M_{k_j}}$ such that the conditional distribution of $\widehat{M}_{n_{k_j}}^{M_{k_j}}$ given the data $\{X_i, Y_i, Z_i, \eta_i^m\}_{i,m=1}^{\infty}$ converges in distribution to $B^*$ almost surely (e.g., see Definition 2.1 of Zhang et al. (2018)). We use the notation $\xrightarrow{D^*}$ to denote convergence in distribution in probability.

**Theorem 9.** *Suppose Assumptions 4 holds, then we have (see Appendix D.3 for precise definitions of relevant constants and random variables),*

1. *Under $H_0$, $n\widehat{M}_n^M \xrightarrow{D^*} T^\dagger$.*

2. *Under $H_{1n}$ with $\alpha = 0$, $n\widehat{M}_n^M \xrightarrow{D^*} T_1 = \sum_{j=1}^{2} \widetilde{T}_j$, where $\{\widetilde{T}_1, \widetilde{T}_2\}$ are i.i.d. random variables following a mixture of centered chi-squares distribution.*

*With Assumption 6 further satisfied, we have*

3. *Under $H_{1n}$ with $\alpha > 0$, $n\widehat{M}_n^M \xrightarrow{D^*} T_A^\dagger$.*

*Furthermore, if we let $M_{nM,\gamma}^*$ denote the $(1 - \gamma)$th quantile of $n\widehat{M}_n^M$ conditioning on the data, then the power (probability of detecting the alternative) of our testing procedure satisfies: if $\alpha < 1/2$, then $\mathbb{P}(n\widehat{T}_n \geq M_{nM,\gamma}^*) \to 1$; if $\alpha = 1/2$, then $\mathbb{P}(n\widehat{T}_n \geq M_{nM,\gamma}^*) \to \mathbb{P}(c + T_A^\dagger + \frac{1}{\sqrt{2}} \sum_{j=1}^{2} \mathcal{G}_j \geq T_{0,\gamma}^A)$, where $T_{0,\gamma}^A$ denotes the $(1 - \gamma)$th quantile of $T_A^\dagger$; if $\alpha > 1/2$, then $\mathbb{P}(n\widehat{T}_n \geq M_{nM,\gamma}^*) \to \gamma$.*

Since $\mathbb{P}(n\widehat{T}_n \geq M_{nM,\gamma}^*) = \mathbb{P}(\mathrm{Rej}_n)$, item 1 of Theorem 9 demonstrates that the proposed test achieves asymptotically correct size; while items 2 and 3, along with the second part of the theorem, establish that the proposed test is consistent (the probability of rejecting $H_0$ converges to 1) when the alternative lies outside an $n^{-1/2}$-neighborhood of $H_0$ (i.e., when $\alpha < 1/2$). Similar to the optimal test based on $n\widehat{T}_n$ discussed after Theorem 8, the proposed test will also exhibit trivial power in detecting alternatives that are too close to the null (i.e., when $\alpha > 1/2$).

## 3. Simulation Results

To evaluate the finite-sample performance of our test, we adopt the two examples from Cai et al. (2024), where the response variable $Y$ is univariate. Each experiment is repeated 500 times with sample sizes $n \in \{200, 400, 800\}$. The nominal significance level is set at 5%.

**Example A1:** *Consider the linear regression model $Y_i = \beta_Z^\top Z_i + \beta_X^\top X_i + \epsilon_i$ for $i \in [n]$, where $\epsilon_i \overset{i.i.d.}{\sim} N(0, 0.5^2)$ are independent of $\{X_i, Z_i\}_{i=1}^{n}$, $d_X = d_Z = 25$ and $(Z_i^\top, X_i^\top) \overset{i.i.d.}{\sim} N(0, \Sigma)$ with the $(i,j)$th element of $\Sigma$ being $\Sigma_{ij} = 0.3^{|i-j|}$ for $i, j \in [50]$. We set the first two components of $\beta_Z$ as one and the rest as zero. For this example, the null $H_0$ corresponds to $\beta_X \equiv 0$. Under the sparse alternative, the first two components of $\beta_X$ are $0.2/\sqrt{2}$ and the rest as zero. Under the dense alternative, every component of $\beta_X$ is fixed at $0.2/\sqrt{25} = 0.04$.*

**Example A2:** *Consider the nonlinear regression model $Y_i = \beta_Z^\top Z_i + (\beta_X^\top X_i)^2 + \epsilon_i$ for $i \in [n]$, under the same $\beta_Z$ and $\{X_i, Z_i, \epsilon_i\}_{i=1}^{n}$ as in Example A1. The null $H_0$ also corresponds to $\beta_X \equiv 0$. Under the sparse alternative, we set the first five components of $\beta_X$ to be $10^{-1/2}$ and the rest as zero. Under the dense alternative, we set the first twelve components of $\beta_X$ as $24^{-1/2}$ and the rest as zero.*

For $\widehat{T}_n$, we opt to use the Laplacian kernel and the bandwidth parameter for each kernel is selected according to the median heuristic (Gretton et al., 2012, Section 8). For comparison, we also include the simulation results for the CMI test proposed in (Williamson et al., 2023, Algorithm

*Table 2.* Empirical size and size adjusted power for Examples A1 and A2.

| | | $n$ | pMIT | | $\text{pMIT}_e$ | | $\text{pMIT}_M$ | | $\text{pMIT}_{e_M}$ | | PCM | $\text{PCM}_M$ | VIM | DSP | $\text{DSP}_M$ | $\widehat{T}_n$ Oracle | $\widehat{T}_n$ |
|---|---|---|---|---|---|---|---|---|---|---|---|---|---|---|---|---|---|
| | | | XGB | DNN | XGB | DNN | XGB | DNN | XGB | DNN | | | | | | | |
| | $H_0$ | 200 | 5.6 | 6.4 | 5.8 | 7.0 | 6.2 | 8.4 | 8.0 | 9.6 | 2.0 | 0.0 | 3.2 | 0.0 | 0.0 | 7.0 | 8.8 |
| | | 400 | 6.0 | 4.2 | 8.0 | 5.8 | 11.8 | 9.6 | 14.2 | 13.2 | 3.4 | 0.0 | 4.8 | 0.0 | 0.0 | 6.8 | 7.0 |
| | | 800 | 7.0 | 11.2 | 9.8 | 12.8 | 14.2 | 17.2 | 20.2 | 21.0 | 1.2 | 0.0 | 4.8 | 0.0 | 0.0 | 7.2 | 5.4 |
| Example A1 | $H_1$ sparse | 200 | 3.8 | 9.4 | 5.6 | 9.4 | 3.2 | 5.6 | 4.8 | 7.6 | 11.0 | 25.4 | 12.6 | 15.8 | 15.8 | 88.4 | 36.0 |
| | | 400 | 18.6 | 34.6 | 30.4 | 35.8 | 11.8 | 55.2 | 24.8 | 56.8 | 27.0 | 81.8 | 14.8 | 14.8 | 29.8 | 100 | 98.6 |
| | | 800 | 91.6 | 66.0 | 97.8 | 66.4 | 96.0 | 96.2 | 99.8 | 95.8 | 87.4 | 99.8 | 31.6 | 52.8 | 91.8 | 100 | 100 |
| | $H_1$ dense | 200 | 6.0 | 11.6 | 7.0 | 11.2 | 3.0 | 15.8 | 4.6 | 20.0 | 15.6 | 30.0 | 13.2 | 22.4 | 43.8 | 98.4 | 56.0 |
| | | 400 | 17.6 | 63.6 | 32.2 | 65.8 | 11.0 | 83.6 | 20.6 | 86.3 | 29.2 | 87.8 | 17.0 | 47.2 | 76.8 | 100 | 100 |
| | | 800 | 86.6 | 94.8 | 93.6 | 95.2 | 85.6 | 100 | 99.0 | 100 | 91.2 | 100 | 38.6 | 80.2 | 98.8 | 100 | 100 |
| | $H_0$ | 200 | 5.6 | 6.4 | 5.8 | 7.0 | 6.2 | 8.4 | 8.0 | 9.6 | 2.0 | 0.0 | 3.2 | 0.0 | 0.0 | 7.0 | 8.8 |
| | | 400 | 6.0 | 4.2 | 8.0 | 5.8 | 11.8 | 9.6 | 14.2 | 13.2 | 3.4 | 0.0 | 4.8 | 0.0 | 0.0 | 6.8 | 7.0 |
| | | 800 | 7.0 | 11.2 | 9.8 | 12.8 | 14.2 | 17.2 | 20.2 | 21.0 | 1.2 | 0.0 | 4.8 | 0.0 | 0.0 | 7.2 | 5.4 |
| Example A2 | $H_1$ sparse | 200 | 7.4 | 7.6 | 13.0 | 12.2 | 17.6 | 10.2 | 27.0 | 21.0 | 10.6 | 17.2 | 10.6 | 46.4 | 74.6 | 36.8 | 19.2 |
| | | 400 | 27.2 | 24.8 | 47.0 | 29.8 | 28.4 | 50.6 | 57.4 | 60.4 | 23.2 | 82.0 | 26.0 | 73.0 | 90.8 | 80.0 | 76.2 |
| | | 800 | 92.2 | 42.6 | 98.6 | 45.6 | 97.6 | 90.0 | 100 | 91.6 | 94.2 | 100 | 71.8 | 99.6 | 99.8 | 100 | 100 |
| | $H_1$ dense | 200 | 7.6 | 8.2 | 12.0 | 12.8 | 14.4 | 9.4 | 25.0 | 24.8 | 6.6 | 11.2 | 10.6 | 59.4 | 79.4 | 23.8 | 17.6 |
| | | 400 | 14.0 | 32.0 | 27.2 | 36.8 | 15.0 | 67.4 | 33.0 | 75.8 | 13.4 | 43.0 | 12.6 | 84.0 | 96.4 | 57.4 | 57.8 |
| | | 800 | 44.2 | 60.0 | 67.6 | 61.6 | 39.8 | 98.4 | 79.4 | 99.2 | 58.0 | 97.2 | 31.2 | 99.6 | 100 | 99.8 | 99.2 |

3) (denoted as VIM), the single and multiple split statistics proposed in (Lundborg et al., 2024, Algorithm 1 and $1^{DR}$) (denoted as PCM and $\text{PCM}_M$ respectively), the single and multiple split statistics proposed in (Dai et al., 2022, equations (3) and (7)) (denoted as DSP and $\text{DSP}_M$ respectively), the single/multiple split CMI tests proposed in Cai et al. (2024) (denoted as pMIT and $\text{pMIT}_M$ respectively) as well as their power enhanced versions (denoted as $\text{pMIT}_e$ and $\text{pMIT}e_M$ respectively). For the four tests proposed in Cai et al. (2024), we include the simulation results when the conditional mean functions are learned using eXtreme Gradient Boosting (XGB) and DNN. In addition, we also show the simulation results for an oracle version of our statistic $\widehat{T}_n$, where the true CME and conditional mean function of $Y$ are used instead of their estimators.

For Example A1, as shown in Table 2, VIM and $\widehat{T}_n$ have relatively accurate size under $H_0$, while PCM and DSP (as well as their multiple split versions) are severely undersized. The empirical size for pMIT with XGB estimation method is close to the nominal level, but it is oversized with DNN estimation method. The pMIT tests with multiple split and/or power enhancement all have large size distortions and the size distortion gets larger as $n$ increases. For the size adjusted power, our test $\widehat{T}_n$ outperforms all other tests for all values of $n$ under both sparse and dense alternatives, and VIM has the largest power loss although it also utilizes sample splitting and cross fitting. Note that $\text{pMIT}_M$ with DNN estimation method actually has larger power than $\text{pMIT}_e$, which is supposed to be the power enhanced version of pMIT.

For Example A2, as shown in Table 2, the empirical size results are the same as in Example A1. For the size adjusted power, DSP and $\text{DSP}_M$ have the best overall performances (especially when the sample size is small) and our test $\widehat{T}_n$ have similar power performance as DSP and $\text{DSP}_M$ when $n = 800$. Note that the power performances of the tests

proposed in Cai et al. (2024) depends heavily on the sparsity of the alternative and the estimation method used. The XGB estimation method has better performance under sparse alternative while the DNN estimation method outperforms under the dense alternative.

# 4. Imaging Data Applications

We apply our proposed CMI method to identify important facial regions for two computer vision tasks: recognizing facial expressions and predicting age.

## 4.1. Facial expression recognition

We examine whether covering specific facial regions affects facial expression prediction accuracy using the FER2013 dataset (Goodfellow et al., 2013), which contains $48 \times 48$ grayscale images labeled with one of seven expressions: angry, disgust, fear, happy, sad, surprise, neutral. Following Dai et al. (2022), we analyze seven hypothesized regions (HRs): top left corner (TL), nose, right eye, mouth, left eye, eyes, and face (see Figure 2). After preprocessing as in Section 6.D of Dai et al. (2022), we use 11,700 image-label pairs, denoted as $\{(X_i, Y_i)\}_{i=1}^{11700}$, and generate images $Z_i$ by masking HRs.

The test statistic $\widehat{T}_n$ is computed on ten subsamples (size $n = 2000$), with the resulting p-values plotted in Figure 1, alongside p-values for $\text{DSP}_M$ statistics (Dai et al., 2022) under 0-1 loss and CE loss. To assess testing results, we compare test accuracies from a VGG network (Khaireddin & Chen, 2021) trained on $(Y_i, Z_i)$ for each HR against the baseline accuracy using $(X_i, Y_i)$. As shown in Figure 1, $\widehat{T}_n$ correctly identifies the nose and TL as non-discriminative regions (p-values above the 5% level), while rejecting $H_0$ for other HRs, consistent with their lower test accuracies. $\text{DSP}_M$ p-values vary by loss function, with CE loss exhibiting stronger detecting power but inflated type-I error for TL and nose. Additionally, $\text{DSP}_M$ results do not align with accuracy trends, highlighting inconsistencies in region

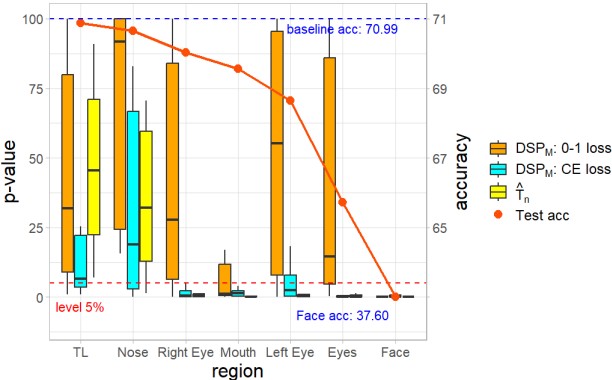

*Figure 1.* Box plot of the p-values (left y-axis) and the test accuracies (red line, right y-axis) for different HRs. The blue dashed line represents the baseline accuracy. The red dashed line represents the 5% nominal level. The test accuracy for the face-covered case (face acc: 37.60) is shown at the bottom right corner.

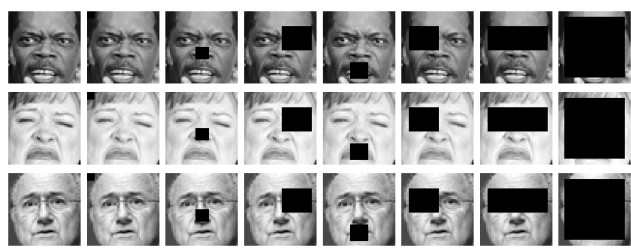

*Figure 2.* Original facial images in FER2013 (first column) and the covered images with HRs: TL, nose, right eye, mouth, left eye, eyes, face (Columns 2-8).

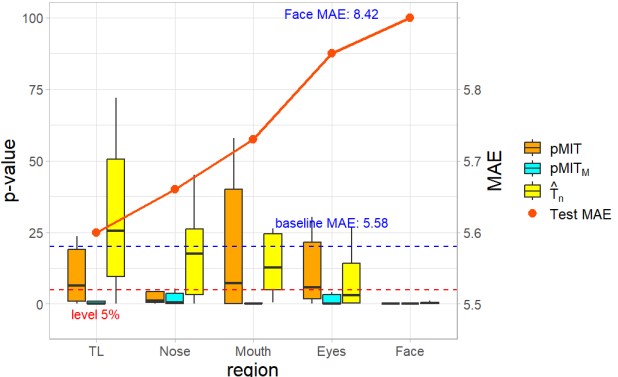

*Figure 3.* Box plot of the p-values (left y-axis) and the test MAE (red line, right y-axis) for different HRs. The blue dashed line represents the baseline MAE. The red dashed line represents the 5% nominal level. The test MAE for the face-covered case (face MAE: 8.42) is shown at the bottom right corner.

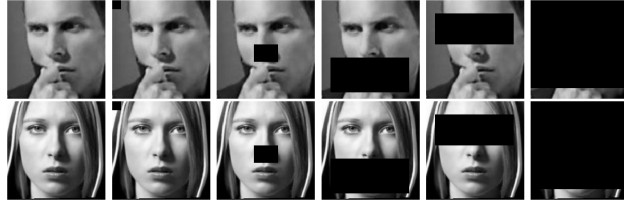

*Figure 4.* Original facial images in UTKFace (first column) and the covered images with HRs: TL, nose, mouth, eyes, face (Columns 2-6).

importance detection. Further details and a more in-depth discussion of the results are provided in Appendix B.1.

### 4.2. Facial age estimation

We investigate the impact of covering specific facial regions on age prediction accuracy using the cropped and aligned UTKFace dataset (Zhang et al., 2017), available at https://www.kaggle.com/datasets/abhikjha/utk-face-cropped. Five hypothesized regions (HRs) are analyzed: top left corner (TL), nose, mouth, eyes, and face (see Figure 4 and Appendix A.3). After converting images to grayscale and selecting age labels between 20–59 years, we obtain 16,425 image-label pairs, $\{(X_i, Y_i)\}_{i=1}^{16425}$, and the masked images $\{Z_i\}_{i=1}^{16425}$. The statistic $\widehat{T}_n$ is computed on ten subsamples ($n = 2000$ each), with p-values and mean absolute errors (MAE) from an EfficientNet B0 model (Tan & Le, 2019) plotted in Figure 3. Baseline MAEs from $(X_i, Y_i)$ are also included for comparison.

Figure 3 shows that $\widehat{T}_n$ p-values are above the 5% level when TL, nose, or mouth is covered, indicating these regions are not critical for age estimation. However, $H_0$ is

rejected for eyes and face due to significantly higher test MAEs. pMIT$_M$ consistently produces low p-values, even when TL is covered (test MAE close to baseline), indicating oversizing. Meanwhile, pMIT fails to reject $H_0$ in some cases (e.g., eyes, mouth) despite higher test MAEs than the nose-covered case, highlighting inconsistencies. Further details and more discussions of the results can be found in Appendix B.2

## 5. Conclusion

In this paper, we propose a novel conditional mean independence test that addresses limitations of existing methods. Using RKHS embedding, sample splitting, cross-fitting, and deep generative neural networks, our fully non-parametric test handles multivariate responses, performs well in high-dimensional settings, and maintains power against local alternatives converging at the $n^{-1/2}$ parametric rate. Simulations and data applications demonstrate its effectiveness. Some potential future directions include exploring conditional quantile independence testing, developing variable selection methods with false selection rate control, and performing diagnostic checks for high-dimensional regression models.

## Impact Statement

This paper presents work whose goal is to advance the field of Machine Learning. There are many potential societal consequences of our work, none which we feel must be specifically highlighted here.

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

## Appendix to "Testing Conditional Mean Independence Using Generative Neural Networks"

The Appendix is organized as follows: Appendix A details the training procedure and hyperparameter selection for the GMMN generator and DNN used in Sections 3 and 4. Appendix B provides an in-depth analysis and discussion of the results from Sections 4.1 and 4.2. Appendix C contains the proof of Proposition 3. Appendix D formalizes and proves Theorems 5-9. Finally, Appendix E includes an essential auxiliary lemma and its proof.

## A. Implementation details

In this section, we offer additional details about the training procedure and network structure of the networks used in the paper.

In practical applications, the conditional distribution $P_{X|Z}$ is often unknown, and we use a generative neural network (GNN) to train a conditional generator $\widehat{\mathbb{G}} : \mathbb{R}^m \times \mathbb{R}^{d_Z} \to \mathbb{R}^{d_X}$ for approximate sampling from $P_{X|Z}$. By sampling a latent variable $\eta$ from a simple distribution (e.g., standard normal) over $\mathbb{R}^m$, the conditional distribution of $\widehat{X} = \widehat{\mathbb{G}}(\eta, Z)$ given $Z$ (denoted as $P_{\widehat{X}|Z}$) serves as a good approximation of $P_{X|Z}$.

We adopt the GMMN framework for its strong performance and alignment with our test. The conditional generator $\widehat{\mathbb{G}}$ is trained by minimizing the sample-based squared MMD between $P_{XZ}$ and $P_{\widehat{X}Z}$, using training data $(X_i, Z_i)_{i=1}^{n_T}$ and $Mn_T$ latent variables $\{\eta_i^m : i = 1, \ldots, n_T, m = 1, \ldots, M\}$:

$$\widehat{\mathbb{G}} = \arg\min_{\mathbb{G} \in \mathcal{G}} \frac{1}{n_T(n_T - 1)} \sum_{\substack{k \neq \ell \\ k, \ell \in [n_T]}} \widehat{U}(X_k, X_\ell) \cdot \mathcal{K}_Z(Z_k, Z_\ell), \tag{5}$$

$$\text{with} \quad \widehat{U}(X_k, X_\ell) = \mathcal{K}_X(X_k, X_\ell) - \frac{1}{M} \sum_{m=1}^{M} \mathcal{K}_X\big(X_k, \mathbb{G}_X(\eta_\ell^m, Z_\ell)\big)$$

$$- \frac{1}{M} \sum_{m=1}^{M} \mathcal{K}_X\big(X_\ell, \mathbb{G}_X(\eta_k^m, Z_k)\big) + \frac{1}{M} \sum_{m=1}^{M} \mathcal{K}_X\big(\mathbb{G}_X(\eta_k^m, Z_k), \mathbb{G}_X(\eta_\ell^m, Z_\ell)\big),$$

where $\mathcal{G}$ represents an approximation family, such as (deep) neural networks. In the original GMMN framework by Dziugaite et al. (2015), the Monte Carlo sample size $M$ is fixed at one. However, our results indicate that increasing $M$ significantly enhances the empirical performance and stability of the training process.

Let $n_b$ denote the batch size, $N_T$ is the total training epoch, $\gamma_X$ is the learning rate of Adam optimizer, $M_b$ is the Monte Carlo sample size mentioned above, $d_m$ is the dimension of the input noise $\eta$, and $\rho$ is the distribution of the input noise. The table below shows how the above mentioned hyperparameters are selected in each section. Other hyperparameters of the GMMN used only in some specific section is discussed in Appendix A.1-A.3. The training procedure is outlined in Algorithm 1.

| Section | $n_b$ | $N_T$ | $\gamma_X$ | $M_b$ | $d_m$ | $\rho$ |
|---------|-------|-------|------------|-------|-------|--------|
| 3 | 128 | 1000 | $4.9 \cdot 10^{-3}$ | 10 | 50 | $\mathcal{N}(0, 1/3 \cdot I_{50})$ |
| 4.1 | 64 | 300 | $1 \cdot 10^{-3}$ | 10 | 2304 | $\mathcal{N}(0, 1/10 \cdot I_{2304})$ |
| 4.2 | 16 | 300 | $1 \cdot 10^{-3}$ | 10 | 50 | $\mathcal{N}(0, 1/10 \cdot I_{50})$ |

*Table 3.* Hyperparameter selection for $\widehat{\mathbb{G}}$

### A.1. Numerical simulation in Section 3

**Model structures and hyperparameters**
To train the DNN $\widehat{g}_Y$, we use the model structure and hyperparameters outlined in the following table with the MSE loss. To train the GMMN $\widehat{\mathbb{G}}$, we follow Algorithm 1 with loss $L_X = L_X^l + L_X^g$, where $L_X^l$ and $L_X^g$ are defined in equation (5) using the Laplace Kernel and Gaussian Kernel, respectively. Let $\gamma$ be the learning rate of Adam optimizer, BN represents whether batch normalization is used, and $\Gamma_{ReLU}$ is the activation coefficient of the leaky ReLU activation function. The hyperparameter values used in Section 3 is summarized in the following table.

---

**Algorithm 1** Training Conditional Generator $\widehat{\mathbb{G}}$

---

**Input:** batch size $n_b$, data $\{(X_i, Y_i, Z_i)\}_{i \in [n]}$, total training epoch $N_T$, learning rate of Adam optimizer $\gamma_X$, number of synthetic samples $M_T$, distribution of the input noise $\rho$

Initialize the conditional generator $\widehat{\mathbb{G}}$

Compute the batch number $b_n = \lceil n/n_b \rceil$

**for** $i = 1$ **to** $N_T$ **do**

    Randomly split the data $\{(X_i, Y_i, Z_i)\}_{i \in [n]}$ into $b_n$ batches

    **for** $b = 1$ **to** $b_n$ **do**

        Select $(X_i, Z_i)$ from the $b$-th batch

        Sample $\eta_i^m \sim \rho$ for $i \in [n_b]$, $m \in [M_T]$

        Compute $\widehat{X}_i^m = \widehat{\mathbb{G}}(\eta_i^m, Z_i)$ and loss $L_X$ based on equation (5)

        Update $\widehat{\mathbb{G}}$ based on loss $L_X$ using Adam optimizer with learning rate $\gamma_X$

    **end for**

**end for**

**Output:** $\widehat{\mathbb{G}}$

---

| | $n_b$ | $N_T$ | $\gamma$ | # of hidden layer | # of nodes | BN | $\Gamma_{ReLU}$ | drop out |
|---|---|---|---|---|---|---|---|---|
| $\widehat{\mathbb{G}}$ | 128 | 1000 | $4.9 \cdot 10^{-3}$ | 1 | 128 | No | 0.8 | 0.05 |
| $\widehat{g}_Y$ | $n$ | 1000 | $2.2 \cdot 10^{-3}$ | 1 | 256 | No | 0.7 | 0.15 |

*Table 4.* Network structure and hyperparameter selection for section 3

## Baseline algorithms

We implemented the pMIT, $\text{pMIT}_e$, $\text{pMIT}_M$, $\text{pMIT}_{eM}$, PCM, $\text{PCM}_M$, VIM, DSP, $\text{DSP}_M$ methods using the code provided by Cai et al. (Section 5.1, 2024) with the default settings.

### A.2. Facial expression recognition application in Section 4.1

#### Sampling procedure

To compute the test accuracy, we adopt the following train-test split procedure: select half of the images from each emotion label (recall that there are 7 emotions) as the testing set (with a sample size of 5850), and use the remaining images as the training set (with a sample size of 5850). We train the VGG network following the settings in Khaireddin & Chen (2021). The trained network is evaluated on the testing set to get the test accuracy.

To obtain a subsample with a size of 2000, we randomly sample 2000/11700 of the triples from each emotion. To compute $\widehat{T}_n$, we then split this subsample equally into two folds, with each fold containing approximately 1000/11700 of the triples from each emotion.

#### Model structures and hyperparameters

We use the VGG network from Khaireddin & Chen (2021) to estimate $g_Y$ and follow their default hyperparameters to train the network.

The GMMN generator is designed to process input images and noise vectors and it is trained by Algorithm 1 with loss $L_X = L_X^l$, where $L_X^l$ is defined in equation (5) using the Laplace Kernel. It starts with a fully connected layer that transforms the noise vector into a $48 \times 48$ feature map, followed by a ReLU activation. This feature map is then concatenated with the input image, resulting in a tensor with dimensions $2 \times 48 \times 48$. The concatenated tensor is passed through an intermediate block consisting of four convolutional layers. Each convolutional layer has 128 output channels, a kernel size of 3, and padding of 1, with ReLU activations applied after each layer. This output of the intermediate block is a feature map of dimensions $128 \times 48 \times 48$, which is passed through a final convolutional layer to reduce the number of channels from 128 to 1 (the final convolutional layer has kernel size of 3 and padding of 1). A Sigmoid activation function is applied to produce the final output, which has dimensions $1 \times 48 \times 48$.

For each HR, the synthetic data $\widehat{X}_i^{(m)}$ used to calculat the statistic $\widehat{T}_n$ is constructed by replacing the covered region in $Z_i$ with the corresponding region from the output image of the trained GMMN.

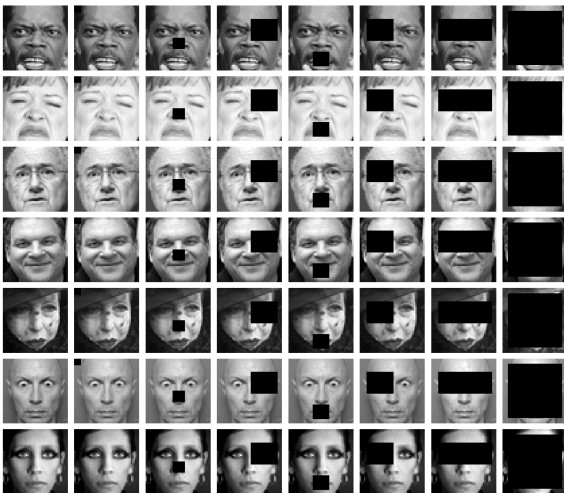

*Figure 5.* Original facial images in FER2013 (first column) and the covered images with HRs: TL, nose, right eye, mouth, left eye, eyes, face (Columns 2-8). From row 1 to 7, the expressions are 'angry', 'disgust', 'fear', 'happy', 'sad', 'surprise', 'neutral'. The pixel locations (height range, width range) for each HR are: **TL**: (1:6, 1:6); **nose**: (25:33, 21:30); **right eye**:(11:27, 26:46); **Case mouth**: (35:46, 19:31); Case **left eye**: (11:27, 6:26); **Case eyes**: (11:27, 6:46); **Case face**: (5:45, 5:45).

### Baseline algorithms

We implemented the $\text{DSP}_M$ (with 0-1 loss and Cross Entropy (CE) loss) using the code provided by Dai et al. (Section 6.D, 2022) with the default settings.

### A.3. Facial age estimation application in Section 4.2

#### Sampling procedure

To compute the test MAE, we adopt the following train-test split procedure: select 50 images from each age label (recall that age labels ranging from 20 to 59) as the testing set (with a sample size of 2000), and use the remaining images as the training set (with a sample size of 14,425). We train the EfficientNet B0 network (Tan & Le, 2019) (with the final layer modified to output a single value) on the training set by minimizing the MAE loss. The trained network is evaluated on the testing set to get the test MAE.

To obtain a subsample with a size of 2000, we randomly sample 50 triples from each age label. To compute $\widehat{T}_n$, we then split this subsample equally into two folds, with each fold containing approximately 25 triples from each age label.

#### Model structures and hyperparameters

We use the EfficientNet B0 network from (Tan & Le, 2019) to estimate $g_Y$, with the following hyperparameters to train the network: batch size $n_b = 128$, total epochs $N_T = 300$, learning rate $\gamma = 1 \times 10^{-3}$, and the loss metric is MAE loss. We employ the `ReduceLROnPlateau` learning rate scheduler, which reduces the learning rate by a factor of 0.2 if the MAE of the testing fold does not improve for 10 consecutive epochs. The minimum learning rate is set to $1 \times 10^{-6}$. To save computation time and avoid overfitting, we adopt the early stopping technique based on the MAE of the testing fold, with an early stopping patience of 25 epochs.

The GMMN generator is trained by Algorithm 1 with loss $L_X = L_X^l$, where $L_X^l$ is defined in equation (5) using the Laplace Kernel and it adopts the same `ReduceLROnPlateau` learning rate scheduler and early stopping technique mentioned above. It starts with a fully connected layer that transforms the noise vector into a $224 \times 224$ feature map, followed by a ReLU activation. This feature map is then concatenated with the input image, resulting in a tensor with dimensions $2 \times 224 \times 224$. The concatenated tensor is passed through an intermediate block consisting of four convolutional layers. Each convolutional layer has 128 output channels, a kernel size of 3, and padding of 1, with ReLU activations applied after each layer. This output of the intermediate block is a feature map of dimensions $128 \times 224 \times 224$, which is passed through

a final convolutional layer to reduce the number of channels from 128 to 1 (the final convolutional layer has kernel size of 3 and padding of 1). A Sigmoid activation function is applied to produce the final output, which has dimensions $1 \times 224 \times 224$.

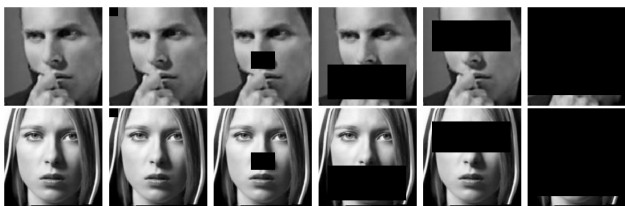

*Figure 6.* Original facial images in UTKFace (first column) and the covered images with HRs: TL, nose, mouth, eyes, face (Columns 2-6). The pixel locations (height range, width range) for each HR are: **TL**: (1:21, 1:21); **nose**: (101:141, 86:141); **mouth**: (131:211, 21:201); **eyes**: (31:101, 21:201); **face**: (1:201,1:224).

For each HR, the synthetic data $\widehat{X}_i^{(m)}$ used to calculate the statistic $\widehat{T}_n$ is constructed by replacing the covered region in $Z_i$ with the corresponding region from the output image of the trained GMMN.

**Baseline algorithms**

We implemented the pMIT, pMIT$_e$, pMIT$_M$, and pMIT$_{eM}$ methods using the code provided by Cai et al. (Section 6.1, 2024). We modified their model to use the same EfficientNet B0 architecture that we employed. Additionally, we changed the loss function to MAE loss and did not scale the age label $Y$.

## B. Detailed analysis of imaging data applications

### B.1. Facial expression recognition

In this application, we examine whether covering some region of a facial image will influence the prediction accuracy of facial expression. We use the Facial Expression Recognition 2013 Dataset (FER2013, Goodfellow et al., 2013) consisting of $48 \times 48$ pixel grayscale facial images, each attached with a label from one of seven facial expressions: angry, disgust, fear, happy, sad, surprise, neutral (denoted as Expression 1-7). As in (Dai et al., 2022), we consider seven cases where different hypothesized regions (HR) are covered: top left corner (TL), nose, right eye, mouth, left eye, eyes, face; see Appendix A.2 and Figure 2 for locations of these HRs.

After applying the same preprocessing procedure as outlined in Section 6.D of Dai et al. (2022), we obtain 11700 image-label pairs which will be the samples used in this application. Let $\{(X_i, Y_i)\}_{i=1}^{11700}$ denote the $48 \times 48$ pixel facial images and their corresponding labels. Note that for any $i \in [11700]$ and $j \in [7]$, $Y_i \in \mathbb{R}^7$ is an one-hot vector with the $j$th component being one and the rest being zero if Expression $j$ is associated with $X_i$. For each HR, we use $\{Z_i\}_{i=1}^{11700}$ to denote the facial images with the HR covered in black.

The statistic $\widehat{T}_n$ is evaluated ten times on different subsamples, with sample size $n = 2000$, from the 11700 triples $\{(X_i, Y_i, Z_i)\}_{i=1}^{11700}$ and the box plot of the ten p-values are plotted in Figure 1. As comparison, we also include the p-values of the DSP$_M$ statistics from (Dai et al., 2022) with different loss functions: 0-1 loss and Cross Entropy (CE) loss. To evaluate the testing results, we calculated the test accuracy of a VGG network (Khaireddin & Chen, 2021) trained/evaluated on the whole sample $\{(Y_i, Z_i)\}_{i=1}^{11700}$, since lower accuracy indicates alternative hypothesis with stronger signal. The resulting test accuracies for different HRs, as well as the accuracy for the VGG network trained/evaluated $\{(X_i, Y_i)\}_{i=1}^{11700}$ (denoted as baseline acc), are also plotted in Figure 1; see Appendix A.2 for the sampling procedure and other implementation details.

As shown in Figure 1, the lower quantiles of the p-values from $\widehat{T}_n$ are above the 5% nominal level when nose or top left corner of the facial image is covered, indicating that these regions are not discriminative to facial expressions. For all the other HRs, the $H_0$ is rejected since all the p-values from $\widehat{T}_n$ are smaller than the nominal level. The test results from $\widehat{T}_n$ is consistent with the test accuracies for different HRs, since the test accuracies when nose or TL is covered are very close to the baseline acc, while for other HRs the test accuracies are noticeably lower than the baseline acc. The test results from DSP$_M$ varies drastically when different loss functions are used. For DSP$_M$ with 0-1 loss, the lower quantiles of the p-values are above (close to) the nominal level for the cases when left eye or both eyes are covered, whereas the test accuracies

for these cases are significantly lower than baseline acc, indicating these regions are indeed important in detecting facial expressions. The $\mathrm{DSP}_M$ with CE loss has stronger detecting power than $\mathrm{DSP}_M$ with 0-1 loss since the p-values for the former is in general smaller than the p-values for the later, which means stronger . However, when TL or nose are covered (which correspond to the cases under $H_0$), the lower quantiles of the p-values from $\mathrm{DSP}_M$ with CE loss are smaller than the nominal level, which may result in inflated type-I error. In addition, the median p-values for different HRs from DSP and $\mathrm{DSP}_M$ do not decrease monotonously as the test accuracies decrease.

### B.2. Facial age estimation

In this application, we investigate the impact of covering specific regions of facial images on the accuracy of age prediction using a well-cropped and aligned version of the UTKFace dataset (Zhang et al., 2017), which is available at https://www.kaggle.com/datasets/abhikjha/utk-face-cropped. We examine five scenarios where different HRs are covered: the top left corner (TL), nose, mouth, eyes, face; see Appendix A.3 and Figure 4 for the locations of these HRs.

After converting the images to grayscale and selecting the age labels ranging from 20 to 59 years old, we obtain 16425 image-label pairs, which will be the samples used in this application. Let $\{(X_i, Y_i)\}_{i=1}^{16425}$ denote the $224 \times 224$ pixel facial images and their corresponding age label $Y_i \in \mathbb{R}$. For each HR, we use $\{Z_i\}_{i=1}^{16425}$ to denote the facial images with the HR covered in black.

The statistic $\widehat{T}_n$ is evaluated ten times on different subsamples, each with a sample size of $n = 2000$, from the triples $\{(X_i, Y_i, Z_i)\}_{i=1}^{16425}$. The box plot of the ten p-values is shown in Figure 3. For comparison, we also include the p-values of the pMIT and $\mathrm{pMIT}_M$ statistics from Cai et al. (2024). To evaluate the testing results, we calculated the MAE (mean absolute error) of a EfficientNet B0 network (Tan & Le, 2019) trained/evaluated on the whole sample $\{(Y_i, Z_i)\}_{i=1}^{16425}$. The resulting MAE for different HRs, as well as the MAE for the EfficientNet B0 network trained/evaluated using $\{(X_i, Y_i)\}_{i=1}^{16425}$ (denoted as baseline MAE), are also plotted in Figure 1; see Appendix A.3 for the sampling procedure and other implementation details.

As shown in Figure 3, the median of the p-values from $\widehat{T}_n$ for different HRs decrease as the MAEs increase, and they are above the 5% nominal level when TL, nose, or mouth (three HRs with the smallest MAEs) of the facial image is covered, indicating that these regions are not discriminative for age estimation. For the eyes- and face-covered cases, the null hypothesis $H_0$ is rejected since the median of p-values from $\widehat{T}_n$ are smaller than the nominal level, which makes sense since the test MAE in these two cases are significantly larger than the baseline MAE.

The p-values from $\mathrm{pMIT}_M$ are all lower than the nominal level, even for the TL-covered case where the test MAE is close to the baseline MAE. This is consistent with the simulation result in Section 3 where $\mathrm{pMIT}_M$ is severely oversized. The p-values from pMIT does not change monotonously with the test MAE. Based on the median p-values, pMIT rejects $H_0$ in the nose-covered case but failed to reject $H_0$ in the month- and eyes-covered cases, even though the later two have larger test MAE than the nose-covered case.

## C. Proof of Proposition 3

The following theorem comes from Theorem 2.2 in Park & Muandet (2020). Note that $\mathbb{H}_X$, $\mathbb{H}_Z$ and $\mathbb{H}_0$ are RKHSs defined in Section 2.1. Denote $\mathbb{H}_X \otimes \mathbb{H}_Z$ as the tensor product space of $\mathbb{H}_X$ and $\mathbb{H}_Z$ (Weidmann, 1980, page 47-48).

**Theorem 10.** $\mathbb{H}_X \otimes \mathbb{H}_Z$ *is generated by the functions* $f \otimes g : \mathbb{R}^{d_X + d_Z} \to \mathbb{R}$, *with* $f \in \mathbb{H}_X$ *and* $g \in \mathbb{H}_Z$ *defined by* $f \otimes g(x, y) = f(x)g(y)$. *Moreover,* $\mathbb{H}_X \otimes \mathbb{H}_Z$ *and* $\mathbb{H}_0$ *are identical.*

The following definition of Hilbert-Schmidt operator and Hilbert-Schmidt norm is from Definition 12.1.1 and Theorem 12.1.1 in Aubin (2000).

**Definition 11.** Let $A$ and $B$ be two separable Hilbert spaces and $\{r_i\}_{i=1}^{\infty}$ is an orthonormal base of $A$. A continuous linear operator $T : A \to B$ is a Hilbert-Schmidt operator if the series

$$\|A\|_{HS} = \Big( \sum_{i=1}^{\infty} \|Tr_i\|_B^2 \Big)^{1/2} \tag{6}$$

converges, and $\| \cdot \|_{HS}$ is called the Hilbert-Schmidt norm.

The following definition of the mean of a Hilbert space valued ranodm element is derived from Theorem 3.2.1 and Definition 7.2.1 in Hsing & Eubank (2015).

**Definition 12.** Let $\chi$ be a random element taking values in a separable Hilbert space $A$. If $\mathbb{E}\|\chi\|_A < 0$, then $\chi$ is Bochner integrable (Hsing & Eubank, 2015, Definition 2.6.3) and its mean $\mathbb{E}\chi$ is a unique element in $A$ such that $\langle f, \mathbb{E}\chi \rangle_A = \mathbb{E}\langle f, \chi \rangle_A$ for any $f \in A$.

### C.1. Proof of part (a) in Proposition 3

We prove $\Sigma$ is a Hilbert-Schmidt operator by showing that $\Sigma c \in \mathbb{H}_0$ for any $c \in \mathbb{R}^{d_Y}$ and $\|\Sigma\|_{HS}^2 < 0$.

To show $\Sigma c \in \mathbb{H}_0$, denote $h(x,y,z,c) = \left[ \mathcal{K}_0((x,z),\cdot) - \mathbb{E}\big[\mathcal{K}_0((X,z),\cdot)|z\big] \right]\big[y - \mathbb{E}[Y|z]\big]^\top c$, then we have $\Sigma c = \mathbb{E}h(X,Y,Z,c)$. By Definition 12, it suffice to show: (1), $h(x,y,z,c) \in \mathbb{H}_0$ for any $c \in \mathbb{R}^{d_Y}$ and almost every $(x,y,z) \in \mathbb{R}^{d_X + d_Y + d_Z}$; (2), $\mathbb{E}\|h(X,Y,Z,c)\|_{\mathbb{H}_0} < \infty$. For part (1), it is clear that $\mathcal{K}_0((x,z),\cdot)\big[y - \mathbb{E}[Y|z]\big]^\top c \in \mathbb{H}_0$, and $\mathbb{E}\big[\mathcal{K}_0((X,z),\cdot)|z\big] \in \mathbb{H}_0$ follows from Lemma 15. So part (1) is proved. For part (2), note that

$$
\begin{aligned}
\mathbb{E}\|h(X,Y,Z,c)\|_{\mathbb{H}_0}^2 =& \mathbb{E}\Big\{ \Big\|\mathcal{K}_0((X,Z),\cdot) - \mathbb{E}\big[\mathcal{K}_0((X,Z),\cdot)|Z\big]\Big\|_{\mathbb{H}_0}^2 \Big[\big[Y - \mathbb{E}[Y|Z]\big]^\top c\Big]^2 \Big\} \\
\leq& 4\|c\|_2^2 \mathbb{E}\Big\{ \Big[\big\|\mathcal{K}_0((X,Z),\cdot)\big\|_{\mathbb{H}_0}^2 + \big\|\mathbb{E}\big[\mathcal{K}_0((X,Z),\cdot)|Z\big]\big\|_{\mathbb{H}_0}^2\Big]\Big[\|Y\|_2^2 + \|g_Y(Z)\|_2^2\Big]\Big\} \\
\leq& 4\|c\|_2^2 \mathbb{E}\Big\{ \Big[\big\|\mathcal{K}_0((X,Z),\cdot)\big\|_{\mathbb{H}_0}^2 + \mathbb{E}\big[\|\mathcal{K}_0((X,Z),\cdot)\|_{\mathbb{H}_0}^2|Z\big]\Big]\Big[\|Y\|_2^2 + \|g_Y(Z)\|_2^2\Big]\Big\} \quad (7) \\
=& 4\|c\|_2^2 \mathbb{E}\Big\{ \Big[\big\|\mathcal{K}_0((X,Z),\cdot)\big\|_{\mathbb{H}_0}^2 + \big\|\mathcal{K}_0((\widetilde{X},Z),\cdot)\big\|_{\mathbb{H}_0}^2\Big]\Big[\|Y\|_2^2 + \|g_Y(Z)\|_2^2\Big]\Big\} \quad (8)
\end{aligned}
$$

where equation (7) follows from a generalized Jensen's inequality (Park & Muandet, 2020, Appendix A) and $\widetilde{X}$ is a random vector such that $(X,Z) \stackrel{d}{=} (\widetilde{X},Z)$ and $\widetilde{X}$ is conditionally independent of $Y$ given $Z$. By Assumption 4(a), the right hand side of equation (8) is finite and part (2) is proved since $\mathbb{E}\|h(X,Y,Z,c)\|_{\mathbb{H}_0} \leq \{\mathbb{E}\|h(X,Y,Z,c)\|_{\mathbb{H}_0}^2\}^{1/2}$.

To show $\|\Sigma\|_{HS}^2 < 0$, let $\{r_i\}_{i=1}^{d_Y}$ be an orthonormal base of $\mathbb{R}^{d_Y}$ where $r_i \in \mathbb{R}^{d_Y}$ is a vector with $i$th element being 1 and the rest being zero for any $i \in [d_Y]$. Then we have

$$
\begin{aligned}
\|\Sigma\|_{HS}^2 =& \sum_{i=1}^{d_Y} \|\Sigma r_i\|_B^2 \\
=& \sum_{i=1}^{d_Y} \Big\| \mathbb{E}\Big\{ \Big[\mathcal{K}_0((X,Z),\cdot) - \mathbb{E}\big[\mathcal{K}_0((X,Z),\cdot)|Z\big]\Big]\big[Y - \mathbb{E}[Y|Z]\big]^\top r_i\Big\}\Big\|_{\mathbb{H}_0}^2 \quad (9) \\
\leq& 2\mathbb{E}\Big\{ \Big[\big\|\mathcal{K}_0((X,Z),\cdot)\big\|_{\mathbb{H}_0}^2 + \big\|\mathbb{E}\big[\mathcal{K}_0((X,Z),\cdot)|Z\big]\big\|_{\mathbb{H}_0}^2 \|Y - \mathbb{E}[Y|Z]\|_2^2\Big]\Big\} \\
\leq& 4\mathbb{E}\Big\{ \Big[\big\|\mathcal{K}_0((X,Z),\cdot)\big\|_{\mathbb{H}_0}^2 + \mathbb{E}\big[\|\mathcal{K}_0((X,Z),\cdot)\|_{\mathbb{H}_0}^2|Z\big]\Big]\Big[\|Y\|_2^2 + \|g_Y(Z)\|_2^2\Big]\Big\} \\
<& \infty, \quad (10)
\end{aligned}
$$

where equation (10) follows from equation (8). So $\Sigma$ is a Hilbert-Schmidt operator.

To prove $\|\Sigma\|_{HS}^2 = \Gamma^*$, note that from equation (9) we have

$$
\begin{aligned}
\|\Sigma\|_{HS}^2 =& \mathbb{E}\Big\{ \Big\langle \mathcal{K}_0((X,Z),\cdot) - \mathbb{E}\big[\mathcal{K}_0((X,Z),\cdot)|Z\big], \mathcal{K}_0((X',Z'),\cdot) - \mathbb{E}\big[\mathcal{K}_0((X',Z'),\cdot)|Z'\big] \Big\rangle_{\mathbb{H}_0} V(Y,Y')\Big\} \\
=& \mathbb{E}\Big\{ \Big\langle \big[\mathcal{K}_X(X,\cdot) - \mathbb{E}\big[\mathcal{K}_X(X,\cdot)|Z\big]\big]\mathcal{K}_Z(Z,\cdot), \big[\mathcal{K}_X(X',\cdot) - \mathbb{E}\big[\mathcal{K}_X(X',\cdot)|Z'\big]\big]\mathcal{K}_Z(Z',\cdot) \Big\rangle_{\mathbb{H}_0} V(Y,Y')\Big\} \\
& \quad (11) \\
=& \mathbb{E}\Big\{ \Big\langle \mathcal{K}_X(X,\cdot) - \mathbb{E}\big[\mathcal{K}_X(X,\cdot)|Z\big], \mathcal{K}_X(X',\cdot) - \mathbb{E}\big[\mathcal{K}_X(X',\cdot)|Z'\big] \Big\rangle_{\mathbb{H}_X} V(Y,Y')\mathcal{K}_Z(Z,Z')\Big\} \\
=& \mathbb{E}\Big\{ U(X,X')V(Y,Y')\mathcal{K}_Z(Z,Z')\Big\}, \quad (12)
\end{aligned}
$$

where equation (11) follows from Lemma 15 and equation (12) follows from the reproducing property of RKHS. So part (a) of Proposition 3 is proved.

### C.2. Proof of part (b) in Proposition 3

The operator norm (Rudin, 1991, Theorem 4.4) of $\Sigma$ is defined as

$$\|\Sigma\|_{OP} = \sup\left\{\left|\langle f, \Sigma c\rangle_{\mathbb{H}_o}\right| : \|c\|_2 \leq 1, \|f\|_{\mathbb{H}_0} \leq 1\right\}. \tag{13}$$

According to Proposition 12.1.2 in (Hsing & Eubank, 2015), $\|\Sigma\|_{OP} \leq \|\Sigma\|_{HS}$. If $\Gamma^* = \|\Sigma\|_{HS}^2 = 0$, then we have $\|\Sigma\|_{OP} = 0$ which is equivalent to $\Sigma$ being a zero operator. If $\|\Sigma\|_{OP} = 0$, then we have $\Sigma r_i = 0 \in \mathbb{H}_0$ for any $i \in [d_Y]$, which implies $\Gamma^* = \|\Sigma\|_{HS}^2 = 0$. So we have that $\Gamma^* = 0$ if and only if $\Sigma$ is a zero operator, which is equivalent to $H_0$ holds according to part (c) of Proposition 1.

## D. Proofs of main results in Section 2.3

Define two oracle statistics as

$$T_n = \frac{1}{2}\sum_{s=1}^{2}\left\{\frac{1}{\frac{n}{2}\left(\frac{n}{2}-1\right)}\sum_{\substack{j\neq k \\ j,k\in\mathcal{J}^{(s)}}} U(X_j,X_k)V(Y_j,Y_k)\mathcal{K}_Z(Z_j,Z_k)\right\} \tag{14}$$

$$\widetilde{T}_n = \frac{1}{2}\sum_{s=1}^{2}\left\{\frac{1}{\frac{n}{2}\left(\frac{n}{2}-1\right)}\sum_{\substack{j\neq k \\ j,k\in\mathcal{J}^{(s)}}} \widetilde{U}(X_j,X_k)\widehat{V}(Y_j,Y_k)\mathcal{K}_Z(Z_j,Z_k)\right\}$$

where

$$\widetilde{U}(X_j,X_k) = \mathcal{K}_X(X_j,X_k) - \mathbb{E}_{\eta_k^m}\mathcal{K}_X(X_j,\widehat{X}_k^{(m)}) - \mathbb{E}_{\eta_j^m}\mathcal{K}_X(\widehat{X}_j^{(m)},X_k) + \mathbb{E}_{\eta_j^m\eta_k^m}\mathcal{K}_X(\widehat{X}_j^{(m)},\widehat{X}_k^{(m)}) \tag{15}$$

Denote $S_i = (X_i, Z_i)$, under $H_{1n}$, it is easy to see that Assumption 4 imply the following assumption:

**Assumption 13.** For $i$, $\alpha_1$ and $D_i$ defined in Assumption 4, we assume there is a large positive number $C_3$ such that

(a) $\mathbb{E}\left\{\mathcal{K}_X(D_i,D_i) + \mathcal{K}_X(D_i,D_i)\left[1 + \|\mathcal{G}(S_i)\|_2^2 + \|\mathcal{R}_{ni}\|_2^2\right]\mathcal{K}_Z(Z_i,Z_i)\right\} < C_3$.

(b) $\sqrt{\mathbb{E}\left\{\|\mathcal{G}(S_i)\|_2^2\|\mathbb{E}[\mathcal{K}_X(\cdot,\widetilde{X}_i^{(1)})|Z_i] - \mathbb{E}[\mathcal{K}_Y(\cdot,\widehat{X}_i^{(1)})|Z_i]\|_{\mathbb{H}_X}^2\mathcal{K}_Z(Z_i,Z_i)\right\}} = O(n^{-\alpha_1})$ and

$\sqrt{\mathbb{E}\left\{\|\mathcal{R}_{ni}\|_2^2\|\mathbb{E}[\mathcal{K}_X(\cdot,\widetilde{X}_i^{(1)})|Z_i] - \mathbb{E}[\mathcal{K}_Y(\cdot,\widehat{X}_i^{(1)})|Z_i]\|_{\mathcal{K}_Y}^2\mathcal{K}_Z(Z_i,Z_i)\right\}} = O(n^{-\alpha_1})$.

### D.1. Proof of Theorem 5

First, we give a formalized version of Theorem 5:

**Theorem 4** (formalized)**.** *Suppose Assumptions 4 holds, then under $H_0$,*

$$\widehat{T}_n - T_n = O_p\big(n^{-1}[M^{-1/2} + n^{-\alpha_1} + n^{-\alpha_2}] + n^{-1/2-(\alpha_1+\alpha_2)}\big)$$

*and $n\widehat{T}_n \xrightarrow{D} T^\dagger = T_1^\dagger + T_2^\dagger$, where $T_n$ is defined in the same way as $\widehat{T}_n$ with $\widehat{U}(X_j,X_k), \widehat{V}(Y_j,Y_k)$ replaced by $U(X_j,X_k), V(Y_j,Y_k)$ (see equation (14) in Appendix D) and $\{T_1^\dagger, T_2^\dagger\}$ are i.i.d. random variables with $T_1^\dagger = \sum_{s=1}^{\infty}\lambda_s(\chi_s^2 - 1)$. Here, $\chi_s^2$ are i.i.d. chi-square random variables with one degree of freedom, and $\lambda_s$'s are eigenvalues of the compact self-adjoint operator on $L_2(\mathbb{R}^{d_X+d_Y+d_Z}, P_{XYZ})$ induced by the kernel function $h((X_1,Y_1,Z_1),(X_2,Y_2,Z_2)) = U(X_1,X_2)V(Y_1,Y_2)\mathcal{K}_Z(Z_1,Z_2)$; that is, there exists orthonormal basis $\{f_i(X_1,Y_1,Z_1)\}_{i=1}^{\infty}$ of $L_2(\mathbb{R}^{d_X+d_Y+d_Z}, P_{XYZ})$ such that*

$$\mathbb{E}\big[h((X_1,Y_1,Z_1),(x,y,z))f_i(X_1,Y_1,Z_1)\big] = \lambda_i f_i(x,y,z).$$

*Proof.* Since $\widehat{T}_n - T_n = O_p\big(n^{-1}[M^{-1/2} + n^{-\alpha_1} + n^{-\alpha_2}] + n^{-1/2-(\alpha_1+\alpha_2)}\big) = o_p(n^{-1})$ by Lemma 17, it suffice to show

$nT_n \xrightarrow{D} T^\dagger$. Note that

$$nT_n = \sum_{s=1}^{2} \left\{ \frac{n}{2} \binom{\frac{n}{2}}{2}^{-1} \sum_{\substack{j<k \\ j,k \in \mathcal{J}^{(s)}}} U(X_j, X_k) V(Y_j, Y_k) \mathcal{K}_Z(Z_j, Z_k) \right\}$$

$$= \sum_{s=1}^{2} \frac{n}{2} T_s^{(n)}.$$

For any $s \in [2]$, $T_s^{(n)}$ is a degenerate U statistic of order two and $\frac{n}{2} T_s^{(n)} \xrightarrow{D} T_s^\dagger$ follows directly follows from asymptotic theory for degenerate U statistic, see Theorem 1 in Section 3.2.2 of Lee (2019) (also stated as Lemma 16 in Appendix E). Since $\{T_1^{(n)}, T_2^{(n)}\}$ are mutually independent, we have $nT_J \xrightarrow{D} T^\dagger$. $\qquad\square$

## D.2. Proof of Theorem 8

First, we give a formalized version of Theorem 8:

**Theorem 5** (formalized). *Suppose Assumptions 4 holds, then under $H_{1n}$,*

1. *If $\alpha = 0$, then $\sqrt{n}(\widehat{T}_n - c_0) \xrightarrow{D} \frac{1}{\sqrt{2}} \sum_{j=1}^{2} \mathcal{G}_j^{(0)}$, where $c_0 = \mathbb{E}\{U(X_1, X_2) V(Y_1, Y_2) \mathcal{K}_Z(Z_1, Z_2)\} > 0$, and $\{\mathcal{G}_1^{(0)}, \mathcal{G}_2^{(0)}\}$ are i.i.d. mean zero normal random variables with variance equal to $4\operatorname{Var}\left(\mathbb{E}\{U(X_1, X_2) V(Y_1, Y_2) \mathcal{K}_Z(Z_1, Z_2) \mid X_2, Y_2, Z_2\}\right)$.*

*With Assumption 6 further satisfied, we have*

2. *If $0 < \alpha < 1/2$, then $n^{2\alpha} \widehat{T}_n \xrightarrow{p} c$, where $c = \mathbb{E}\{U(X_1, X_2) \mathcal{G}(X_1, Z_1)^\top \mathcal{G}(X_2, Z_2) \mathcal{K}_Z(Z_1, Z_2)\} > 0$.*

3. *If $\alpha = 1/2$, then $n\widehat{T}_n \xrightarrow{D} c + T_A^\dagger + \frac{1}{\sqrt{2}} \sum_{j=1}^{2} \mathcal{G}_j$, where $T_A^\dagger = \sum_{j=1}^{2} T_{Aj}^\dagger$ and $\{T_{A1}^\dagger, T_{A2}^\dagger\}$ are i.i.d. random variables with $T_{A1}^\dagger = \sum_{i=1}^{\infty} \lambda_i^A (\chi_i^2 - 1)$, $\chi_i^2$ are i.i.d. chi-square random variables with one degree of freedom and $\lambda_i^A$'s are the eigenvalues corresponding to the kernel function $h((X_1, \zeta_1, Z_1), (X_2, \zeta_2, Z_2)) = U(X_1, X_2) \zeta_1^\top \zeta_2 \mathcal{K}_Z(Z_1, Z_2)$. Here $\mathcal{G}_j$ are independent mean zero normal random variables, possibly correlated with $T_{Aj}^\dagger$, with variance equal to $4\operatorname{Var}\left(\mathbb{E}\{U(X_1, X_2)[\mathcal{G}(X_1, Z_1)^\top \zeta_2 + \mathcal{G}(X_2, Z_2)^\top \zeta_1] \mathcal{K}_Z(Z_1, Z_2) \mid \zeta_2, X_2, Z_2\}\right)$.*

4. *If $\alpha > 1/2$, $n\widehat{T}_n \xrightarrow{D} T_A^\dagger$.*

*Proof.* If $\alpha = 0$, by Lemma 17 we have $\sqrt{n}\widehat{T}_n = \sqrt{n}T_n + o_p(1)$, so we only need to show that $\sqrt{n}(T_n - c_0) \xrightarrow{D} \frac{1}{\sqrt{2}} \sum_{j=1}^{2} \mathcal{G}_j^{(0)}$. Note that

$$\sqrt{n}(T_n - c_0) = \frac{1}{\sqrt{2}} \sum_{s=1}^{2} \sqrt{\frac{n}{2}} \left\{ \binom{\frac{n}{2}}{2}^{-1} \sum_{\substack{j<k \\ j,k \in \mathcal{J}^{(s)}}} U(X_j, X_k) V(Y_j, Y_k) \mathcal{K}_Z(Z_j, Z_k) - c_0 \right\}$$

$$= \frac{1}{\sqrt{2}} \sum_{s=1}^{2} \sqrt{\frac{n}{2}} (\hat{T}_s^{(n)} - c_0).$$

For any $s \in [2]$, $\hat{T}_s^{(n)}$ is a non-degenerate U statistic with $\mathbb{E}\hat{T}_s^{(n)} = c_0$ and $\sqrt{\frac{n}{2}}(\hat{T}_s^{(n)} - c_0) \xrightarrow{D} \mathcal{G}_s^{(0)}$ follows directly follows from asymptotic theory for non-degenerate U statistic, see Theorem 1 in Section 3.2.1 of Lee (2019) (also stated as Lemma 16 in Appendix E). Since $\{\hat{T}_1^{(n)}, \hat{T}_2^{(n)}\}$ are mutually independent, we have $\sqrt{n}(T_n - c_0) \xrightarrow{D} \frac{1}{\sqrt{2}} \sum_{j=1}^{2} \mathcal{G}_j^{(0)}$.

If $\alpha > 0$, by Lemma 17 and Assumption 6 we have

$$
\begin{aligned}
n\widehat{T}_n =&\, nT_n + o_p(n^{1/2-\alpha}+1)\\
=&\, \frac{n}{2}\sum_{s=1}^{2}\left\{\frac{n^{-2\alpha}}{\frac{n}{2}(\frac{n}{2}-1)}\sum_{\substack{j\neq k\\j,k\in\mathcal{J}^{(s)}}} U(X_j,X_k)\mathcal{G}(S_j)^\top\mathcal{G}(S_k)\mathcal{K}_Z(Z_j,Z_k)\right\}\\
&+\frac{n}{2}\sum_{s=1}^{2}\left\{\frac{n^{-\alpha}}{\frac{n}{2}(\frac{n}{2}-1)}\sum_{\substack{j\neq k\\j,k\in\mathcal{J}^{(s)}}} U(X_j,X_k)\big[\mathcal{G}(S_j)^\top\zeta_k+\mathcal{G}(S_k)^\top\zeta_j\big]\mathcal{K}_Z(Z_j,Z_k)\right\}\\
&+\frac{n}{2}\sum_{s=1}^{2}\left\{\frac{n^{-\alpha}}{\frac{n}{2}(\frac{n}{2}-1)}\sum_{\substack{j\neq k\\j,k\in\mathcal{J}^{(s)}}} U(X_j,X_k)\big[\mathcal{G}(S_j)^\top(\mathcal{R}_{nk}-\zeta_k)+\mathcal{G}(S_k)^\top(\mathcal{R}_{nj}-\zeta_j)\big]\mathcal{K}_Z(Z_j,Z_k)\right\}\\
&+\frac{n}{2}\sum_{s=1}^{2}\left\{\frac{1}{\frac{n}{2}(\frac{n}{2}-1)}\sum_{\substack{j\neq k\\j,k\in\mathcal{J}^{(s)}}} U(X_j,X_k)\big[(\mathcal{R}_{nj}-\zeta_j)^\top\zeta_k+(\mathcal{R}_{nk}-\zeta_k)^\top\zeta_j\big]\mathcal{K}_Z(Z_j,Z_k)\right\}\\
&+\frac{n}{2}\sum_{s=1}^{2}\left\{\frac{1}{\frac{n}{2}(\frac{n}{2}-1)}\sum_{\substack{j\neq k\\j,k\in\mathcal{J}^{(s)}}} U(X_j,X_k)(\mathcal{R}_{nj}-\zeta_j)^\top(\mathcal{R}_{nk}-\zeta_k)\mathcal{K}_Z(Z_j,Z_k)\right\}\\
&+\frac{n}{2}\sum_{s=1}^{2}\left\{\frac{1}{\frac{n}{2}(\frac{n}{2}-1)}\sum_{\substack{j\neq k\\j,k\in\mathcal{J}^{(s)}}} U(X_j,X_k)\zeta_j^\top\zeta_k\mathcal{K}_Z(Z_j,Z_k)\right\}+o_p(n^{1/2-\alpha}+1)\\
=&\, n^{1-2\alpha}J_1 + n^{1-\alpha}J_2 + n^{1-\alpha}J_3 + nJ_4 + nJ_5 + nJ_6 + o_p(n^{1/2-\alpha}+1).
\end{aligned}
\tag{16}
$$

Note that $J_1 \xrightarrow{P} \mathbb{E}\big\{U(X_1,X_2)\mathcal{G}(S_1)^\top\mathcal{G}(S_2)\mathcal{K}_Z(Z_1,Z_2)\big\} = c > 0$. $J_2$ is mean zero non-degenerate U statistics and $J_6$ is degenerate U statistics. By Assumption 6 and part (a) of Assumption 13, we have $J_3 = o_p(n^{-1/2})$ and $J_4, J_5 = o_p(n^{-1})$. If $0<\alpha<1/2$, $n\widehat{T}_n = n^{1-2\alpha}J_1(1+o_p(1)) \to \infty$. If $\alpha > 1/2$, $n\widehat{T}_n = nJ_6 + o_p(1) \xrightarrow{D} T_A^\dagger$ follows in the same way as in the proof of Theorem 5. If $\alpha = 1/2$, $n\widehat{T}_n = J_1 + \sqrt{n}J_2 + nJ_6 + o_p(1)$. Following similar approach as in the proof for Theorem 2 of Lee et al. (2020), we have $\sqrt{n}J_2 + nJ_6 \xrightarrow{D} \frac{1}{\sqrt{2}}\sum_{j=1}^{2}\mathcal{G}_j + T_A^\dagger$ and the proof is finished.

$\square$

### D.3. Proof of Theorem 9

First, we give a formalized version of Theorem 9:

**Theorem 6** (formalized). *Suppose Assumptions 4 holds, then we have,*

1. *Under $H_0$, $n\widehat{M}_n^M \xrightarrow{D^*} T^\dagger$.*

2. *Under $H_{1n}$ with $\alpha = 0$, $n\widehat{M}_n^M \xrightarrow{D^*} T_1 = \sum_{j=1}^{2}\widetilde{T}_j$, where $\{\widetilde{T}_1,\widetilde{T}_2\}$ are i.i.d random variables with $\widetilde{T}_1 = \sum_{i=1}^{\infty}\gamma_i(\chi_i^2-1)$, $\chi_i^2$ are i.i.d chi-square random variables with one degree of freedom and $\gamma_i$s are eigenvalues of $h((X_1,Y_1,Z_1),(X_2,Y_2,Z_2)) = U(X_1,X_2)V(Y_1,Y_2)\mathcal{K}_Z(Z_1,Z_2)$.*

*With Assumption 6 further satisfied, we have*

3. *Under $H_{1n}$ with $\alpha > 0$, $n\widehat{M}_n^M \xrightarrow{D^*} T_A^\dagger$.*

*Furthermore, if we let $M_{nM,\gamma}^*$ denote the $(1-\gamma)$th quantile of $n\widehat{M}_n^M$ conditioning on the data, then the power (probability of detecting the alternative) of our testing procedure satisfies: if $\alpha<1/2$, then $\mathbb{P}(n\widehat{T}_n \geq M_{nM,\gamma}^*) \to 1$; if $\alpha = 1/2$, then $\mathbb{P}(n\widehat{T}_n \geq M_{nM,\gamma}^*) \to \mathbb{P}(c+T_A^\dagger+\frac{1}{\sqrt{2}}\sum_{j=1}^{2}\mathcal{G}_j \geq T_{0,\gamma}^A)$, where $T_{0,\gamma}^A$ denotes the $(1-\gamma)$th quantile of $T_A^\dagger$; if $\alpha > 1/2$, then $\mathbb{P}(n\widehat{T}_n \geq M_{nM,\gamma}^*) \to \gamma$.*

*Proof.* Define

$$
M_n = \frac{1}{J}\sum_{s=1}^{J}\left\{\frac{1}{\frac{n}{J}(\frac{n}{J}-1)}\sum_{\substack{j\neq k\\j,k\in\mathcal{J}^{(s)}}} U(X_j,X_k)V(Y_j,Y_k)\mathcal{K}_Z(Z_j,Z_k)e_je_k\right\}
$$

and $\{\mathcal{I}_1, \widehat{\mathcal{I}}_1, \widehat{\mathcal{I}}_2, \widehat{\mathcal{I}}_3\}$ according to the same equations as $\{I_1, \widehat{I}_1, \widehat{I}_2, \widehat{I}_3\}$ in Appendix E with $\mathcal{K}_Z(Z_j, Z_k)$ replaced by $\mathcal{K}_Z(Z_j, Z_k)e_j e_k$. Denote $\mathbb{E}^*$ and $P^*$ as the conditional expectation and probability of a random variable conditioning on $\{X_i, Y_i, Z_i, \eta_i^m\}_{i=1, m=1}^{n, M}$. First, we show that the difference between $\widehat{M}_n^M$ and $M_n$ is asymptotically negligible in the sense that

$$P\big(P^*(|n\widehat{M}_n^M - nM_n| \geq \epsilon) \geq \delta\big) \to 0 \text{ for any } \epsilon > 0, \delta > 0. \tag{17}$$

By Chebyshev's inequality, it suffice to show $\mathbb{E}^*|n\widehat{M}_n^M - nM_n|^2 = o_p(1)$, which is implied by $\mathbb{E}^*[\mathcal{I}_1^2 + \widehat{\mathcal{I}}_1^2 + \widehat{\mathcal{I}}_2^2 + \widehat{\mathcal{I}}_3^2] = o_p(n^{-2})$. Note that $\mathbb{E}^*[\mathcal{I}_1^2 + \widehat{\mathcal{I}}_1^2 + \widehat{\mathcal{I}}_2^2 + \widehat{\mathcal{I}}_3^2] = o_p(n^{-2})$ is already shown in the proof of Lemma 17. For example,

$$\mathbb{E}^*\mathcal{I}_1^2 = \frac{1}{\left[\frac{n}{2}(\frac{n}{2} - 1)\right]^2} \sum_{\substack{j \neq k \\ j,k \in \mathcal{J}^{(1)}}} H_1(X_j, X_k)^2 \widehat{V}(Y_j, Y_k)^2 \mathcal{K}_Z(Z_j, Z_k)^2,$$

and the right hand side is $o_p(n^{-2})$ according to part (a) of the proof of Lemma 17 for $I_1$.

$$\mathbb{E}^*\widehat{\mathcal{I}}_1^2 = \frac{1}{\left[\frac{n}{2}(\frac{n}{2} - 1)\right]^2} \sum_{\substack{j \neq k \\ j,k \in \mathcal{J}^{(1)}}} \widehat{H}_1(X_j, X_k)^2 V(Y_j, Y_k)^2 \mathcal{K}_Z(Z_j, Z_k)^2,$$

and the right hand side is $o_p(n^{-2})$ according to part (b) of the proof of Lemma 17 for $\widehat{I}_1$.

Note that $nM_n \xrightarrow{D^*} T^\dagger$ under $H_0$ and $nM_J^n \xrightarrow{D^*} T_1$ under $H_{1n}$ with $\alpha = 0$ is a standard result in bootstrapping for U-statistics (see Theorem 4 in Lee et al. (2020) or Theorem 3.1 in Dehling & Mikosch (1994)).

It remains to show $nM_n \xrightarrow{D^*} T_A^\dagger$ under $H_{1n}$ with $\alpha > 0$. For $i \in [6]$, define $\mathcal{D}_i$ according to the same formulas as $J_i$ in equation (16) with $\mathcal{K}_Z(Z_j, Z_k)$ replaced by $\mathcal{K}_Z(Z_j, Z_k)e_j e_k$. Then we have

$$nM_n = n^{1-2\alpha}\mathcal{D}_1 + n^{1-\alpha}\mathcal{D}_2 + n^{1-\alpha}\mathcal{D}_3 + n\mathcal{D}_4 + n\mathcal{D}_5 + n\mathcal{D}_6.$$

It is easy to see that $\mathbb{E}^*\mathcal{D}_i^2 = O_p(n^{-2})$ for $i \in [3]$ and $\mathbb{E}^*\mathcal{D}_i^2 = o_p(n^{-2})$ for $i = 4, 5$ (by part (a) of Assumption 13). So the difference between $nM_n$ and $n\mathcal{D}_6$ is asymptotically negligible and $n\mathcal{D}_6 \xrightarrow{D^*} T_A^\dagger$ follows from Theorem 4 in Lee et al. (2020) or Theorem 3.1 in Dehling & Mikosch (1994).

To show the asymptotic power of our test, let $M_{nM,\gamma}^*$ denote the $(1 - \gamma)$th quantile of $n\widehat{M}_n^M$ conditioning on $\{X_i, Y_i, Z_i, \eta_i^m\}_{i=1, m=1}^{n, M}$. By theorem 9 and part (ii) of Lemma 11.2.1 in Lehmann & Romano (2006), $M_{nM,\gamma}^* \xrightarrow{P} T_{0,\gamma}$, $M_{nM,\gamma}^* \xrightarrow{P} T_{0,\gamma}^A$ $M_{nM,\gamma}^* \xrightarrow{P} T_{1,\gamma}$ under $H_0$, $H_{1n}$ with $\alpha > 0$ and $H_{1n}$ with $\alpha = 0$ respectively, where $T_{0,\gamma}$, $T_{0,\gamma}^A$ and $T_{1,\gamma}$ are the $(1 - \gamma)$th quantiles of $T^\dagger$, $T_A^\dagger$ and $T_1$ respectively. Then the asymptotic power results in Theorem 9 follows. $\qquad\square$

## E. Auxiliary Lemmas and Its Proofs

In this section, we give three lemmas that are used in the proofs of Proposition 3 and Theorems 5-9.

The following lemma is used in the proof of Proposition 3.

**Lemma 15.** *If Assumption 4(a) holds, then* $\mathbb{E}\big[\mathcal{K}_0((X, z), \cdot)\big|z\big] \in \mathbb{H}_0$ *is well defined and* $\mathbb{E}\big[\mathcal{K}_0((X, z), \cdot)\big|z\big] = \mathbb{E}\big[\mathcal{K}_X(X, \cdot)\big|z\big]\mathcal{K}_Z(z, \cdot)$ *for almost every* $z \in \mathbb{R}^{d_Z}$.

*Proof.* By Assumption 4(a),

$$\mathbb{E}\Big\{\mathbb{E}\big[\|\mathcal{K}_0((X, Z), \cdot)\|_{\mathbb{H}_0}\big|Z\big]\Big\} = \mathbb{E}\Big\{\sqrt{\mathcal{K}_X(X, X)\mathcal{K}_Z(Z, Z)}\Big\}$$

$$\leq \Big\{\mathbb{E}\big[\mathcal{K}_X(X, X)\mathcal{K}_Z(Z, Z)\big]\Big\}^{1/2} < \infty, \tag{18}$$

which implies $\mathbb{E}\big[\|\mathcal{K}_0((X, z), \cdot)\|_{\mathbb{H}_0}\big|z\big] < \infty$ for almost every $z \in \mathbb{R}^{d_Z}$. So from Definition 12 we have that $\mathbb{E}\big[\mathcal{K}_0((X, z), \cdot)\big|z\big] \in \mathbb{H}_0$ is well defined for almost every $z \in \mathbb{R}^{d_Z}$. Similarly, we can show $\mathbb{E}\big[\mathcal{K}_X(X, \cdot)\big|z\big] \in \mathbb{H}_X$

is well defined for almost every $z \in \mathbb{R}^{d_Z}$. For any $f \in \mathbb{H}_X$ and $g \in \mathbb{H}_Z$, by Assumption 4(a) we have $\mathbb{E}|f(X)| \leq \|f\|_{\mathbb{H}_X} \mathbb{E}\sqrt{\mathcal{K}_X(X, X)} < \infty$ and similarly $\mathbb{E}|f(X)g(Z)| < \infty$. By Theorem 10 we have

$$
\begin{aligned}
\langle \mathbb{E}[\mathcal{K}_0((X, z), \cdot)|z], f \otimes g \rangle_{\mathbb{H}_0} &= \mathbb{E}\Big[f(X)g(z)\Big|z\Big] \\
&= \mathbb{E}\Big[f(X)\Big|z\Big]g(z) \\
&= \mathbb{E}\Big[\langle \mathcal{K}_X(X, \cdot), f \rangle_{\mathbb{H}_X}\Big|z\Big]\langle \mathcal{K}_Z(z, \cdot), g \rangle_{\mathbb{H}_Z} \\
&= \langle \mathbb{E}[\mathcal{K}_X(X, \cdot)|z]\mathcal{K}_Z(z, \cdot), f \otimes g \rangle_{\mathbb{H}_0},
\end{aligned}
$$

so we have $\mathbb{E}[\mathcal{K}_0((X, z), \cdot)|z] = \mathbb{E}[\mathcal{K}_X(X, \cdot)|z]\mathcal{K}_Z(z, \cdot)$ for almost every $z \in \mathbb{R}^{d_Z}$ and the lemma is proved. $\qquad\square$

The following lemma summarizes Theorem 1 in Section 3.2.1 and Theorem 1 in Section 3.2.2 of Lee (2019), and is used in the proofs of Theorems 5 and 8.

**Lemma 16.** *Let $U_n$ be a U-statistic based on kernel $h(x_1, x_2)$ with $\sigma_1^2 = \mathrm{Var}(\mathbb{E}[h(X_1, X_2)|X_2])$ and $\theta = \mathbb{E}[h(X_1, X_2)]$, where $\{X_i\}_{i=1}^n$ are i.i.d. copies from distribution $P_X$. We have*

1. *If $\sigma_1^2 > 0$, then $\sqrt{n}(U_n - \theta) \xrightarrow{d} \mathcal{N}(0, 4\sigma_1^2)$.*

2. *If $\sigma_1^2 = 0$ and $\mathbb{E}[h^2(X_1, X_2)]$, then $nU_n \xrightarrow{d} \sum_{s=1}^{\infty}\lambda_s(Z_s^2 - 1)$, where $\{Z_s\}_{s=1}^{\infty}$ are independent standard normal random variables, and the $\lambda_s$ are the eigenvalues of the integral equation*

$$
\int h(x_1, x_2)f(x_2)dP_X(x_2) = \lambda f(x_1).
$$

The following lemma gives the rate at which the difference between the statistic $\widehat{T}_n$ and the oracle statistic $T_n$ decays to zero, which is used in the proofs of Theorems 5-9.

**Lemma 17.** *Suppose Assumption 4 holds, then we have*

1. $\widehat{T}_n - T_n = O_p(n^{-1}[M^{-1/2} + n^{-\alpha_1} + n^{-\alpha_2}] + n^{-1/2 - (\alpha_1 + \alpha_2)})$ *under $H_0$.*

2. $\widehat{T}_n - T_n = O_p(n^{-\min\{\alpha + 1/2, 1\}}[M^{-1/2} + n^{-\alpha_1} + n^{-\alpha_2}] + n^{-\min\{\alpha, 1/2, \alpha_1 + \alpha_2\} - (\alpha_1 + \alpha_2)})$ *under $H_{1n}$.*

*Proof.* The proof is divided into two parts:

(a) $\widehat{T}_n - \widetilde{T}_n = O_p(M^{-1/2}[n^{-1} + n^{-1/2 - (\alpha_1 + \alpha_2)}])$ under $H_0$ and $\widehat{T}_n - \widetilde{T}_n = O_p(M^{-1/2}[n^{-1} + n^{-1/2 - (\alpha_1 + \alpha_2)} + n^{-1/2 - \alpha}])$ under $H_{1n}$.

(b) $\widetilde{T}_n - T_n = O_p(n^{-1 - \min\{\alpha_1, \alpha_2, \alpha_1 + \alpha_2 - 1/2\}})$ under $H_0$ and $\widetilde{T}_n - T_n = O_p(n^{-\min\{\alpha + 1/2, 1\} - \min\{\alpha_1, \alpha_2, \alpha_1 + \alpha_2 - 1/2\}})$ under $H_{1n}$.

Note that in the following proof, we use $C$ to denote a generic positive constant which may vary each time it appears.

***Proof of part (a):***

It suffice to prove the asymptotic order in part (a) for $\widehat{T}_n^{(1)} - \widetilde{T}_n^{(1)}$ where

$$
\widehat{T}_n^{(1)} = \frac{1}{\frac{n}{2}(\frac{n}{2} - 1)} \sum_{\substack{j \neq k \\ j, k \in \mathcal{J}^{(1)}}} \widehat{U}(X_j, X_k)\widehat{V}(Y_j, Y_k)\mathcal{K}_Z(Z_j, Z_k),
$$

$$
\widetilde{T}_n^{(1)} = \frac{1}{\frac{n}{2}(\frac{n}{2} - 1)} \sum_{\substack{j \neq k \\ j, k \in \mathcal{J}^{(1)}}} \widetilde{U}(X_j, X_k)\widehat{V}(Y_j, Y_k)\mathcal{K}_Z(Z_j, Z_k).
$$

Since $\widehat{U}(X_j, X_k) = \widetilde{U}(X_j, X_k) - H_1(X_j, X_k)$, where

$$
\begin{aligned}
W_1(j, k, m) =& \mathcal{K}_X(X_j, \widehat{X}_k^{(m)}) + \mathcal{K}_X(X_k, \widehat{X}_j^{(m)}) - \mathcal{K}_X(\widehat{X}_j^{(m)}, \widehat{X}_k^{(m)}) \\
& - \mathbb{E}_{\eta_k^1}\mathcal{K}_X(X_j, \widehat{X}_k^{(1)}) - \mathbb{E}_{\eta_j^1}\mathcal{K}_X(\widehat{X}_j^{(1)}, X_k) + \mathbb{E}_{\eta_j^1\eta_k^1}\mathcal{K}_X(\widehat{X}_j^{(1)}, \widehat{X}_k^{(1)}),
\end{aligned}
$$

$$
H_1(X_j, X_k) = \frac{1}{M}\sum_{m=1}^{M}W_1(j, k, m),
$$

we have $\widehat{T}_n^{(1)} - \widetilde{T}_n^{(1)} = -I_1$, where

$$I_1 = \frac{1}{\frac{n}{2}\left(\frac{n}{2}-1\right)} \sum_{\substack{j \neq k \\ j,k \in \mathcal{J}^{(1)}}} H_1(X_j, X_k)\widehat{V}(Y_j, Y_k)\mathcal{K}_Z(Z_j, Z_k).$$

For any $\epsilon \in (0,1)$, by Assumptions 4, 13 and Chebyshev's inequality, there exist a large positive number $C_4$ such that for any positive integer $n$, $\mathbb{P}(\mathcal{A}_n) > 1 - \epsilon$ where

$$
\begin{aligned}
\mathcal{A}_n = &\Bigg\{ \{(X_i, Y_i, Z_i)\}_{i \in \mathcal{J}^{(-1)}} \Bigg| \mathbb{E}\Big\{ \mathcal{K}_X(D_1, D_1)\|E_1\|_2^2 \mathcal{K}_Z(Z_1, Z_1) \Big| \{(X_i, Y_i, Z_i)\}_{i \in \mathcal{J}^{(-1)}} \Big\} < C_4, \\
&\sqrt{\mathbb{E}\Big\{ \|\mathbb{E}[\mathcal{K}_X(\cdot, X_1)|Z_1] - \mathbb{E}[\mathcal{K}_X(\cdot, \widehat{X}_1^{(1)})|Z_1]\|_{\mathbb{H}_X}^2 \big[\sqrt{\mathcal{K}_Z(Z_1, Z_1)} + \|E_1\|_2^2\mathcal{K}_Z(Z_1, Z_1)\big] \Big\}} \leq C_4 n^{-\alpha_1}, \\
&\sqrt{\mathbb{E}\Big\{ \|g_Y(Z_1) - \widehat{g}_Y(Z_1)\|_2^2 \big[\sqrt{\mathcal{K}_Z(Z_1, Z_1)} + \mathcal{K}_X(D_1, D_1)\mathcal{K}_Z(Z_1, Z_1)\big] \Big\}} \leq C_4 n^{-\alpha_2}, \\
&\mathbb{E}\Big\{ \mathcal{K}_X(D_1, D_1)\big[\|\mathcal{G}(S_1)\|_2^2 + \|\mathcal{R}_{n1}\|_2^2\big]\mathcal{K}_Z(Z_1, Z_1) \Big\} < C_4, \\
&\sqrt{\mathbb{E}\Big\{ \|\mathcal{G}(S_1)\|_2^2 \|\mathbb{E}[\mathcal{K}_X(\cdot, \widetilde{X}_1^{(1)})|Z_1] - \mathbb{E}[\mathcal{K}_Y(\cdot, \widehat{X}_1^{(1)})|Z_1]\|_{\mathbb{H}_X}^2 \mathcal{K}_Z(Z_1, Z_1) \Big\}} \leq C_4 n^{-\alpha_2}, \\
&\sqrt{\mathbb{E}\Big\{ \|\mathcal{R}_{n1}\|_2^2 \|\mathbb{E}[\mathcal{K}_X(\cdot, \widetilde{X}_1^{(1)})|Z_1] - \mathbb{E}[\mathcal{K}_Y(\cdot, \widehat{X}_1^{(1)})|Z_1]\|_{\mathcal{K}_Y}^2 \mathcal{K}_Z(Z_1, Z_1) \Big\}} \leq C_4 n^{-\alpha_2} \Bigg\}
\end{aligned}
\tag{19}
$$

In the following proof, we fix $\{(X_i, Y_i, Z_i)\}_{i \in \mathcal{J}^{(-1)}} \in \mathcal{A}_n$ and treat $\widehat{\mathbb{G}}_1$ and $\widehat{g}_1$ as fixed functions. Note that $\mathbb{E}[H_1(X_j, X_k)\widehat{V}(Y_j, Y_k)\mathcal{K}_Z(Z_j, Z_k)] = 0$, so

$$
\begin{aligned}
\mathrm{Var}(I_1) \leq &\frac{C}{n^2}\mathbb{E}\Big[ H_1(X_1, X_2)\widehat{V}(Y_1, Y_2)\mathcal{K}_Z(Z_1, Z_2) \Big]^2 \\
&+ \frac{C}{n}\Big| \mathbb{E}\Big[ H_1(X_1, X_2)H_1(X_1, X_3)\widehat{V}(Y_1, Y_2)\widehat{V}(Y_1, Y_3)\mathcal{K}_Z(Z_1, Z_2)\mathcal{K}_Z(Z_1, Z_3) \Big] \Big| \\
= &I_1^{(1)} + I_1^{(2)}
\end{aligned}
$$

First, we show that $I_1^{(1)} \leq \frac{C}{n^2 M}$ under $H_0$ or $H_{1n}$. For integers $m_1, m_2 \in [M]$ such that $m_1 \neq m_2$, we have $\mathbb{E}\big\{ W_1(1, 2, m_1)W_1(1, 2, m_2)|Y_1, S_1, Y_2, S_2 \big\} = 0$, which implies

$$
\begin{aligned}
&\mathbb{E}\big\{ H_1(X_1, X_2)^2|Y_1, S_1, Y_2, S_2 \big\} \\
=&\frac{1}{M}\mathbb{E}\big\{ W_1(1, 2, 1)^2|Y_1, S_1, Y_2, S_2 \big\} \\
\leq&\frac{C}{M}\mathbb{E}\big\{ \mathcal{K}_X(X_1, X_1)\mathcal{K}_X(X_2, X_2) + \mathcal{K}_X(\widehat{X}_1^{(1)}, \widehat{X}_1^{(1)})\mathcal{K}_X(X_2, X_2) + \mathcal{K}_X(X_1, X_1)\mathcal{K}_X(\widehat{X}_2^{(1)}, \widehat{X}_2^{(1)}) \\
&+ \mathcal{K}_X(\widehat{X}_1^{(1)}, \widehat{X}_1^{(1)})\mathcal{K}_X(\widehat{X}_2^{(1)}, \widehat{X}_2^{(1)})|Y_1, S_1, Y_2, S_2 \big\}.
\end{aligned}
$$

It can also be shown that $\widehat{V}(Y_1, Y_2)^2 \leq C\mathbb{E}\big\{ \|Y_1\|_2^2\|Y_2\|_2^2 + \|\widehat{g}_Y(Z_1)\|_2^2\|Y_2\|_2^2 + \|Y_1\|_2^2\|\widehat{g}_Y(Z_2)\|_2^2 + \|\widehat{g}_Y(Z_1)\|_2^2\|\widehat{g}_Y(Z_2)\|_2^2 \big| S_1, S_2 \big\}$. So from part (a) of Assumption 4 (conditioning on $\{(X_i, Y_i, Z_i)\}_{i \in \mathcal{J}^{(-1)}} \in \mathcal{A}_n$), we know $I_1^{(1)} \leq \frac{C}{n^2 M}$. For $I_1^{(2)}$, we have that under

$H_0$

$$I_1^{(2)} = \frac{C}{nM} \left| \mathbb{E}\left[ W_1(1,2,1) W_1(1,3,1) \widehat{V}(Y_1, Y_2) \widehat{V}(Y_1, Y_3) \mathcal{K}_Z(Z_1, Z_2) \mathcal{K}_Z(Z_1, Z_3) \right] \right|$$

$$\leq \frac{C}{nM} \mathbb{E} \left| \langle \mathcal{K}_X(\widehat{X}_1^{(1)}, \cdot) - \mathbb{E}_{\eta_1^1} \mathcal{K}_X(\widehat{X}_1^{(1)}, \cdot), \mathbb{E}_{\eta_2^1} \mathcal{K}_X(\widetilde{X}_2^{(1)}, \cdot) - \mathbb{E}_{\eta_2^1} \mathcal{K}_X(\widehat{X}_2^{(1)}, \cdot) \rangle_{\mathbb{H}_X} \right.$$
$$\times \langle \mathcal{K}_X(\widehat{X}_1^{(1)}, \cdot) - \mathbb{E}_{\eta_1^1} \mathcal{K}_X(\widehat{X}_1^{(1)}, \cdot), \mathbb{E}_{\eta_3^1} \mathcal{K}_X(\widetilde{X}_3^{(1)}, \cdot) - \mathbb{E}_{\eta_3^1} \mathcal{K}_X(\widehat{X}_3^{(1)}, \cdot) \rangle_{\mathbb{H}_X}$$
$$\times \left[ Y_1 - \widehat{g}_Y(Z_1) \right]^\top \left[ g_Y(Z_2) - \widehat{g}_Y(Z_2) \right]$$
$$\left. \times \left[ Y_1 - \widehat{g}_Y(Z_1) \right]^\top \left[ g_Y(Z_3) - \widehat{g}_Y(Z_3) \right] \mathcal{K}_Z(Z_1, Z_2) \mathcal{K}_Z(Z_1, Z_3) \right|$$

$$\leq \frac{C}{nM} \mathbb{E} \left[ \| \mathcal{K}_X(\widehat{X}_1^{(1)}, \cdot) - \mathbb{E}_{\eta_1^1} \mathcal{K}_X(\widehat{X}_1^{(1)}, \cdot) \|_{\mathbb{H}_X}^2 \| Y_1 - \widehat{g}_Y(Z_1) \|_2^2 \mathcal{K}_Z(Z_1, Z_1) \right]$$
$$\times \mathbb{E} \left[ \| \mathbb{E}_{\eta_2^1} \mathcal{K}_X(\widetilde{X}_2^{(1)}, \cdot) - \mathbb{E}_{\eta_2^1} \mathcal{K}_X(\widehat{X}_2^{(1)}, \cdot) \|_{\mathbb{H}_X} \| g_Y(Z_2) - \widehat{g}_Y(Z_2) \|_2 \sqrt{\mathcal{K}_Z(Z_2, Z_2)} \right]$$
$$\times \mathbb{E} \left[ \| \mathbb{E}_{\eta_3^1} \mathcal{K}_X(\widetilde{X}_3^{(1)}, \cdot) - \mathbb{E}_{\eta_3^1} \mathcal{K}_X(\widehat{X}_3^{(1)}, \cdot) \|_{\mathbb{H}_X} \| g_Y(Z_3) - \widehat{g}_Y(Z_3) \|_2 \sqrt{\mathcal{K}_Z(Z_3, Z_3)} \right].$$

By part (a) of Assumption 4, $\mathbb{E}\left[ \| \mathcal{K}_X(\widehat{X}_1^{(1)}, \cdot) - \mathbb{E}_{\eta_1^1} \mathcal{K}_X(\widehat{X}_1^{(1)}, \cdot) \|_{\mathbb{H}_X}^2 \| Y_1 - \widehat{g}_Y(Z_1) \|_2^2 \mathcal{K}_Z(Z_1, Z_1) \right] < CC_4$. By part (b) of Assumption 4 and Cauchy–Schwarz inequality,

$$\mathbb{E}\left[ \| \mathbb{E}_{\eta_2^1} \mathcal{K}_X(\widetilde{X}_2^{(1)}, \cdot) - \mathbb{E}_{\eta_2^1} \mathcal{K}_X(\widehat{X}_2^{(1)}, \cdot) \|_{\mathbb{H}_X} \| g_Y(Z_2) - \widehat{g}_Y(Z_2) \|_2 \sqrt{\mathcal{K}_Z(Z_2, Z_2)} \right] \leq C n^{-\alpha_1 - \alpha_2},$$
$$\mathbb{E}\left[ \| \mathbb{E}_{\eta_3^1} \mathcal{K}_X(\widetilde{X}_3^{(1)}, \cdot) - \mathbb{E}_{\eta_3^1} \mathcal{K}_X(\widehat{X}_3^{(1)}, \cdot) \|_{\mathbb{H}_X} \| g_Y(Z_3) - \widehat{g}_Y(Z_3) \|_2 \sqrt{\mathcal{K}_Z(Z_3, Z_3)} \right] \leq C n^{-\alpha_1 - \alpha_2}.$$

So we have $I_1^{(2)} \leq \frac{C}{n^{1+2(\alpha_1+\alpha_2)} M}$ under $H_0$. Under $H_{1n}$, since $\mathbb{E}[\mathcal{R}_{ni} | S_i] = 0$,

$$I_1^{(2)} = \frac{C}{nM} \left| \mathbb{E}\left[ \langle \mathcal{K}_X(\widehat{X}_1^{(1)}, \cdot) - \mathbb{E}_{\eta_1^1} \mathcal{K}_X(\widehat{X}_1^{(1)}, \cdot), \mathcal{K}_X(X_2, \cdot) - \mathbb{E}_{\eta_2^1} \mathcal{K}_X(\widehat{X}_2^{(1)}, \cdot) \rangle_{\mathbb{H}_X} \right. \right.$$
$$\times \langle \mathcal{K}_X(\widehat{X}_1^{(1)}, \cdot) - \mathbb{E}_{\eta_1^1} \mathcal{K}_X(\widehat{X}_1^{(1)}, \cdot), \mathcal{K}_X(X_3, \cdot) - \mathbb{E}_{\eta_3^1} \mathcal{K}_X(\widehat{X}_3^{(1)}, \cdot) \rangle_{\mathbb{H}_X}$$
$$\times \left[ Y_1 - \widehat{g}_Y(Z_1) \right]^\top \left[ g_Y(Z_2) - \widehat{g}_Y(Z_2) + n^{-\alpha} \mathcal{G}(S_2) \right]$$
$$\left. \left. \times \left[ Y_1 - \widehat{g}_Y(Z_1) \right]^\top \left[ g_Y(Z_3) - \widehat{g}_Y(Z_3) + n^{-\alpha} \mathcal{G}(S_3) \right] \mathcal{K}_Z(Z_1, Z_2) \mathcal{K}_Z(Z_1, Z_3) \right] \right|. \tag{20}$$

By Assumptions 4 and 13, we have

$$\left| \mathbb{E}\left[ \langle \mathcal{K}_X(\widehat{X}_1^{(1)}, \cdot) - \mathbb{E}_{\eta_1^1} \mathcal{K}_X(\widehat{X}_1^{(1)}, \cdot), \mathcal{K}_X(X_2, \cdot) - \mathbb{E}_{\eta_2^1} \mathcal{K}_X(\widehat{X}_2^{(1)}, \cdot) \rangle_{\mathbb{H}_X} \langle \mathcal{K}_X(\widehat{X}_1^{(1)}, \cdot) - \mathbb{E}_{\eta_1^1} \mathcal{K}_X(\widehat{X}_1^{(1)}, \cdot), \mathcal{K}_X(X_3, \cdot) - \mathbb{E}_{\eta_3^1} \mathcal{K}_X(\widehat{X}_3^{(1)}, \cdot) \rangle_{\mathbb{H}_X} \right. \right.$$
$$\left. \left. \times \left[ Y_1 - \widehat{g}_Y(Z_1) \right]^\top \left[ n^{-\alpha} \mathcal{G}(S_2) \right] \left[ Y_1 - \widehat{g}_Y(Z_1) \right]^\top \left[ g_Y(Z_3) - \widehat{g}_Y(Z_3) \right] \mathcal{K}_Z(Z_1, Z_2) \mathcal{K}_Z(Z_1, Z_3) \right] \right|.$$
$$\leq \mathbb{E}\left[ \| \mathcal{K}_X(\widehat{X}_1^{(1)}, \cdot) - \mathbb{E}_{\eta_1^1} \mathcal{K}_X(\widehat{X}_1^{(1)}, \cdot) \|_{\mathbb{H}_X}^2 \| Y_1 - \widehat{g}_Y(Z_1) \|_2^2 \mathcal{K}_Z(Z_1, Z_1) \right]$$
$$\times \mathbb{E}\left[ \| \mathcal{K}_X(X_2, \cdot) - \mathbb{E}_{\eta_2^1} \mathcal{K}_X(\widehat{X}_2^{(1)}, \cdot) \|_{\mathbb{H}_X} \| n^{-\alpha} \mathcal{G}(S_2) \|_2 \sqrt{\mathcal{K}_Z(Z_2, Z_2)} \right]$$
$$\times \mathbb{E}\left[ \| \mathbb{E}_{\eta_3^1} \mathcal{K}_X(\widetilde{X}_3^{(1)}, \cdot) - \mathbb{E}_{\eta_3^1} \mathcal{K}_X(\widehat{X}_3^{(1)}, \cdot) \|_{\mathbb{H}_X} \| g_Y(Z_3) - \widehat{g}_Y(Z_3) \|_2 \sqrt{\mathcal{K}_Z(Z_3, Z_3)} \right]$$
$$\leq C n^{-\alpha - \alpha_1 - \alpha_2},$$

$$\left| \mathbb{E}\left[ \langle \mathcal{K}_X(\widehat{X}_1^{(1)}, \cdot) - \mathbb{E}_{\eta_1^1} \mathcal{K}_X(\widehat{X}_1^{(1)}, \cdot), \mathcal{K}_X(X_2, \cdot) - \mathbb{E}_{\eta_2^1} \mathcal{K}_X(\widehat{X}_2^{(1)}, \cdot) \rangle_{\mathbb{H}_X} \langle \mathcal{K}_X(\widehat{X}_1^{(1)}, \cdot) - \mathbb{E}_{\eta_1^1} \mathcal{K}_X(\widehat{X}_1^{(1)}, \cdot), \mathcal{K}_X(X_3, \cdot) - \mathbb{E}_{\eta_3^1} \mathcal{K}_X(\widehat{X}_3^{(1)}, \cdot) \rangle_{\mathbb{H}_X} \right. \right.$$
$$\left. \left. \times \left[ Y_1 - \widehat{g}_Y(Z_1) \right]^\top \left[ g_Y(Z_2) - \widehat{g}_Y(Z_2) \right] \left[ Y_1 - \widehat{g}_Y(Z_1) \right]^\top \left[ g_Y(Z_3) - \widehat{g}_Y(Z_3) \right] \mathcal{K}_Z(Z_1, Z_2) \mathcal{K}_Z(Z_1, Z_3) \right] \right|.$$
$$\leq \mathbb{E}\left[ \| \mathcal{K}_X(\widehat{X}_1^{(1)}, \cdot) - \mathbb{E}_{\eta_1^1} \mathcal{K}_X(\widehat{X}_1^{(1)}, \cdot) \|_{\mathbb{H}_X}^2 \| Y_1 - \widehat{g}_Y(Z_1) \|_2^2 \mathcal{K}_Z(Z_1, Z_1) \right]$$
$$\times \mathbb{E}\left[ \| \mathbb{E}_{\eta_2^1} \mathcal{K}_X(\widetilde{X}_2^{(1)}, \cdot) - \mathbb{E}_{\eta_2^1} \mathcal{K}_X(\widehat{X}_2^{(1)}, \cdot) \|_{\mathbb{H}_X} \| g_Y(Z_2) - \widehat{g}_Y(Z_2) \|_2 \sqrt{\mathcal{K}_Z(Z_2, Z_2)} \right]$$
$$\times \mathbb{E}\left[ \| \mathbb{E}_{\eta_3^1} \mathcal{K}_X(\widetilde{X}_3^{(1)}, \cdot) - \mathbb{E}_{\eta_3^1} \mathcal{K}_X(\widehat{X}_3^{(1)}, \cdot) \|_{\mathbb{H}_X} \| g_Y(Z_3) - \widehat{g}_Y(Z_3) \|_2 \sqrt{\mathcal{K}_Z(Z_3, Z_3)} \right]$$
$$\leq C n^{-2(\alpha_1 + \alpha_2)}$$

and

$$\left| \mathbb{E}\Big[ \langle \mathcal{K}_X(\widehat{X}_1^{(1)},\cdot) - \mathbb{E}_{\eta_1^1}\mathcal{K}_X(\widehat{X}_1^{(1)},\cdot), \mathcal{K}_X(X_2,\cdot) - \mathbb{E}_{\eta_2^1}\mathcal{K}_X(\widehat{X}_2^{(1)},\cdot) \rangle_{\mathbb{H}_X} \langle \mathcal{K}_X(\widehat{X}_1^{(1)},\cdot) - \mathbb{E}_{\eta_1^1}\mathcal{K}_X(\widehat{X}_1^{(1)},\cdot), \mathcal{K}_X(X_3,\cdot) - \mathbb{E}_{\eta_3^1}\mathcal{K}_X(\widehat{X}_3^{(1)},\cdot) \rangle_{\mathbb{H}_X} \right.$$
$$\left. \times \big[Y_1 - \widehat{g}_Y(Z_1)\big]^\top \big[n^{-\alpha}\mathcal{G}(S_2)\big]\big[Y_1 - \widehat{g}_Y(Z_1)\big]^\top \big[n^{-\alpha}\mathcal{G}(S_3)\big]\mathcal{K}_Z(Z_1,Z_2)\mathcal{K}_Z(Z_1,Z_3) \Big] \right|.$$
$$\leq \mathbb{E}\Big[ \|\mathcal{K}_X(\widehat{X}_1^{(1)},\cdot) - \mathbb{E}_{\eta_1^1}\mathcal{K}_X(\widehat{X}_1^{(1)},\cdot)\|_{\mathbb{H}_X}^2 \|Y_1 - \widehat{g}_Y(Z_1)\|_2^2 \mathcal{K}_Z(Z_1,Z_1) \Big]$$
$$\times \mathbb{E}\Big[ \|\mathcal{K}_X(X_2,\cdot) - \mathbb{E}_{\eta_2^1}\mathcal{K}_X(\widehat{X}_2^{(1)},\cdot)\|_{\mathbb{H}_X} \|n^{-\alpha}\mathcal{G}(S_2)\|_2 \sqrt{\mathcal{K}_Z(Z_2,Z_2)} \Big]$$
$$\times \mathbb{E}\Big[ \|\mathcal{K}_X(X_3,\cdot) - \mathbb{E}_{\eta_3^1}\mathcal{K}_X(\widehat{X}_3^{(1)},\cdot)\|_{\mathbb{H}_X} \|n^{-\alpha}\mathcal{G}(S_3)\|_2 \sqrt{\mathcal{K}_Z(Z_3,Z_3)} \Big]$$
$$\leq Cn^{-2\alpha},$$

which implies $I_1^{(2)} \leq CM^{-1}(n^{-1-2(\alpha_1+\alpha_2)} + n^{-1-(\alpha+\alpha_1+\alpha_2)} + n^{-1-2\alpha}) \leq CM^{-1}(n^{-2} + n^{-1-2\alpha})$ under $H_{1n}$. Combining with the result for $I_1^{(1)}$, we can conclude that conditioning on $\{(X_i,Y_i,Z_i)\}_{i\in\mathcal{J}^{(-1)}} \in \mathcal{A}_n$, $\mathbb{E}[I_1]^2 \leq CM^{-1}(n^{-2} + n^{-1-2(\alpha_1+\alpha_2)})$ under $H_0$ and $\mathbb{E}[I_1]^2 \leq CM^{-1}(n^{-1-2\alpha} + n^{-2} + n^{-1-2(\alpha_1+\alpha_2)})$ under $H_{1n}$. Let $\mathcal{A}_n^c$ denote the complement set of $\mathcal{A}_n$, Under $H_0$, since

$$\mathbb{P}(|\frac{M^{1/2}I_1}{n^{-1}+n^{-1/2-\alpha_1-\alpha_2}}| > C) \leq \mathbb{P}(|\frac{M^{1/2}I_1}{n^{-1}+n^{-1/2-\alpha_1-\alpha_2}}| > C, \mathcal{A}_n) + P(A_n^c)$$
$$= \mathbb{E}\Big[ P(|\frac{M^{1/2}I_1}{n^{-1}+n^{-1/2-\alpha_1-\alpha_2}}| > C | \{(X_i,Y_i,Z_i)\}_{i\in\mathcal{J}^{(-1)}}) \mathbb{1}_{\mathcal{A}_n}(\{(X_i,Y_i,Z_i)\}_{i\in\mathcal{J}^{(-1)}}) \Big] + \mathbb{P}(A_n^c)$$
$$\leq \frac{1}{C^2} + \epsilon \tag{21}$$

and $\epsilon$ can be arbitrarily small, we can conclude that, without conditioning on $\{(X_i,Y_i,Z_i)\}_{i\in\mathcal{J}^{(-1)}}$, $I_1 = O_p(M^{-1/2}[n^{-1}+n^{-1/2-(\alpha_1+\alpha_2)}])$ under $H_0$. Similarly, we can prove $I_1 = O_p(M^{-1/2}[n^{-1}+n^{-1/2-(\alpha_1+\alpha_2)}+n^{-1/2-\alpha}])$ under $H_{1n}$. So part (a) is proved.

Since the same conditioning argument as in equation (21) can be used to derive the asymptotic order without conditioning on the training data, from now on, we treat all the neural networks as fixed functions.

***Proof of part (b)***:

It suffice to prove the asymptotic order in part (b) for $\widetilde{T}_n^{(1)} - T_n^{(1)}$ where

$$T_n^{(1)} = \frac{1}{\frac{n}{2}(\frac{n}{2}-1)} \sum_{\substack{j\neq k \\ j,k\in\mathcal{J}^{(1)}}} U(X_j,X_k)V(Y_j,Y_k)\mathcal{K}_Z(Z_j,Z_k).$$

Since $\widetilde{U}(X_j,X_k) = U(X_j,X_k) - \widehat{H}_1(X_j,X_k)$ and $\widehat{V}(Y_j,Y_k) = V(Y_j,Y_k) - \widehat{H}_2(Y_j,Y_k)$, where

$$\widehat{H}_1(X_j,X_k) = \mathbb{E}_{\eta_k^1}\mathcal{K}_X(X_j,\widehat{X}_k^{(1)}) + \mathbb{E}_{\eta_j^1}\mathcal{K}_X(\widehat{X}_j^{(1)},X_k) - \mathbb{E}_{\eta_j^1\eta_k^1}\mathcal{K}_X(\widehat{X}_j^{(1)},\widehat{X}_k^{(1)})$$
$$- \mathbb{E}_{\eta_k^1}\mathcal{K}_X(X_j,\widetilde{X}_k^{(1)}) - \mathbb{E}_{\eta_j^1}\mathcal{K}_X(\widetilde{X}_j^{(1)},X_k) + \mathbb{E}_{\eta_j^1\eta_k^1}\mathcal{K}_X(\widetilde{X}_j^{(1)},\widetilde{X}_k^{(1)}),$$
$$\widehat{H}_2(Y_j,Y_k) = Y_j^\top \widehat{g}_Y(Z_k) + \widehat{g}_Y(Z_j)^\top Y_k - \widehat{g}_Y(Z_j)^\top \widehat{g}_Y(Z_k)$$
$$- Y_j^\top g_Y(Z_k) - g_Y(Z_j)^\top Y_k + g_Y(Z_j)^\top g_Y(Z_k),$$

then we have $\widetilde{T}_n^{(1)} - T_n^{(1)} = -\widehat{I}_1 - \widehat{I}_2 + \widehat{I}_3$, where

$$\widehat{I}_1 = \frac{1}{\frac{n}{2}(\frac{n}{2}-1)} \sum_{\substack{j\neq k \\ j,k\in\mathcal{J}^{(1)}}} \widehat{H}_1(X_j,X_k)V(Y_j,Y_k)\mathcal{K}_Z(Z_j,Z_k),$$
$$\widehat{I}_2 = \frac{1}{\frac{n}{2}(\frac{n}{2}-1)} \sum_{\substack{j\neq k \\ j,k\in\mathcal{J}^{(1)}}} U(X_j,X_k)\widehat{H}_2(Y_j,Y_k)\mathcal{K}_Z(Z_j,Z_k),$$
$$\widehat{I}_3 = \frac{1}{\frac{n}{2}(\frac{n}{2}-1)} \sum_{\substack{j\neq k \\ j,k\in\mathcal{J}^{(1)}}} \widehat{H}_1(X_j,X_k)\widehat{H}_2(Y_j,Y_k)\mathcal{K}_Z(Z_j,Z_k).$$

Consider $\widehat{I}_1$. Under $H_0$,

$$\mathbb{E}[\widehat{H}_1(X_j, X_k)V(Y_j, Y_k)\mathcal{K}_Z(Z_j, Z_k)|Y_j, S_j] = \mathbb{E}[\widehat{H}_1(X_j, X_k)V(Y_j, Y_k)\mathcal{K}_Z(Z_j, Z_k)|Y_k, S_k] = 0,$$

so we have

$$\mathrm{Var}(\widehat{I}_1) = \frac{1}{\frac{n}{2}(\frac{n}{2}-1)}\mathbb{E}\Big[\widehat{H}_1(X_1, X_2)^2 V(Y_1, Y_2)^2 \mathcal{K}_Z(Z_1, Z_2)^2\Big]. \tag{22}$$

Note that $V(Y_1, Y_2)^2 \le C\big\{\|Y_1\|_2^2\|Y_2\|_2^2 + \|g_Y(Z_1)\|_2^2\|Y_2\|_2^2 + \|Y_1\|_2^2\|g_Y(Z_2)\|_2^2 + \|g_Y(Z_1)\|_2^2\|g_Y(Z_2)\|_2^2\big\}$,

$$\begin{aligned}
\widehat{H}_1(X_1, X_2)^2 \le C\Big\{&\big[\mathbb{E}_{\eta_2^1}\mathcal{K}_X(X_1, \widetilde{X}_2^{(1)}) - \mathbb{E}_{\eta_2^1}\mathcal{K}_X(X_1, \widehat{X}_2^{(1)})\big]^2 + \big[\mathbb{E}_{\eta_1^1}\mathcal{K}_X(X_2, \widetilde{X}_1^{(1)}) - \mathbb{E}_{\eta_1^1}\mathcal{K}_X(X_2, \widehat{X}_1^{(1)})\big]^2 \\
&+ \big[\mathbb{E}_{\eta_1^1\eta_2^1}\mathcal{K}_X(\widehat{X}_2^{(1)}, \widetilde{X}_1^{(1)}) - \mathbb{E}_{\eta_1^1\eta_2^1}\mathcal{K}_X(\widehat{X}_2^{(1)}, \widehat{X}_1^{(1)})\big]^2 + \big[\mathbb{E}_{\eta_1^1\eta_2^1}\mathcal{K}_X(\widetilde{X}_1^{(1)}, \widetilde{X}_2^{(1)}) - \mathbb{E}_{\eta_1^1\eta_2^1}\mathcal{K}_X(\widetilde{X}_1^{(1)}, \widehat{X}_2^{(1)})\big]^2\Big\},
\end{aligned} \tag{23}$$

and $\big[\mathbb{E}_{\eta_2^1}\mathcal{K}_X(X_1, \widetilde{X}_2^{(1)}) - \mathbb{E}_{\eta_2^1}\mathcal{K}_X(X_1, \widehat{X}_2^{(1)})\big]^2 \le \mathcal{K}_X(X_1, X_1)\|\mathbb{E}_{\eta_2^1}\mathcal{K}_X(\cdot, \widetilde{X}_2^{(1)}) - \mathbb{E}_{\eta_2^1}\mathcal{K}_X(\cdot, \widehat{X}_2^{(1)})\|_{\mathbb{H}_X}^2$, so by part (a) and (b) of Assumption 4, we have $\widehat{I}_1 = O_p(n^{-1-\alpha_1})$. Under $H_{1n}$,

$$\begin{aligned}
\widehat{I}_1 = &\frac{1}{\frac{n}{2}(\frac{n}{2}-1)} \sum_{\substack{j\neq k \\ j,k\in\mathcal{J}^{(1)}}} \widehat{H}_1(X_j, X_k)\big[n^{-\alpha}\mathcal{G}(S_j)\big]^\top [\mathcal{R}_{nk}]\mathcal{K}_Z(Z_j, Z_k) \\
&+ \frac{1}{\frac{n}{2}(\frac{n}{2}-1)} \sum_{\substack{j\neq k \\ j,k\in\mathcal{J}^{(1)}}} \widehat{H}_1(X_j, X_k)\big[n^{-\alpha}\mathcal{G}(S_k)\big]^\top [\mathcal{R}_{nj}]\mathcal{K}_Z(Z_j, Z_k) \\
&+ \frac{1}{\frac{n}{2}(\frac{n}{2}-1)} \sum_{\substack{j\neq k \\ j,k\in\mathcal{J}^{(1)}}} \widehat{H}_1(X_j, X_k)\big[n^{-\alpha}\mathcal{G}(S_j)\big]^\top \big[n^{-\alpha}\mathcal{G}(S_k)\big]\mathcal{K}_Z(Z_j, Z_k) \\
&+ \frac{1}{\frac{n}{2}(\frac{n}{2}-1)} \sum_{\substack{j\neq k \\ j,k\in\mathcal{J}^{(1)}}} \widehat{H}_1(X_j, X_k)\big[\mathcal{R}_{nj}\big]^\top [\mathcal{R}_{nk}]\mathcal{K}_Z(Z_j, Z_k) \\
= &\widehat{I}_1^{(1)} + \widehat{I}_1^{(2)} + \widehat{I}_1^{(3)} + \widehat{I}_1^{(4)}.
\end{aligned}$$

We now show $\widehat{I}_1^{(1)} = O_p(n^{-1/2-\alpha-\alpha_1})$ (same result holds for $\widehat{I}_1^{(2)}$ and the proof is omitted). Note that $\mathbb{E}\big\{\widehat{H}_1(X_j, X_k)\big[n^{-\alpha}\mathcal{G}(S_j)\big]^\top [\mathcal{R}_{nk}]\mathcal{K}_Z(Z_j, Z_k)\big\} = 0$, by equation (23), we have

$$\begin{aligned}
\mathrm{Var}(\widehat{I}_1^{(1)}) \le \frac{C}{n}\Big\{&\mathbb{E}\big\{\big[\mathbb{E}_{\eta_2^1}\mathcal{K}_X(X_1, \widetilde{X}_2^{(1)}) - \mathbb{E}_{\eta_2^1}\mathcal{K}_X(X_1, \widehat{X}_2^{(1)})\big]^2\|n^{-\alpha}\mathcal{G}(S_1)\|_2^2\|\mathcal{R}_{n2}\|_2^2\mathcal{K}_Z(Z_1, Z_1)\mathcal{K}_Z(Z_2, Z_2)\big\} \\
&+ \mathbb{E}\big\{\big[\mathbb{E}_{\eta_1^1}\mathcal{K}_X(X_2, \widetilde{X}_1^{(1)}) - \mathbb{E}_{\eta_1^1}\mathcal{K}_X(X_2, \widehat{X}_1^{(1)})\big]^2\|n^{-\alpha}\mathcal{G}(S_1)\|_2^2\|\mathcal{R}_{n2}\|_2^2\mathcal{K}_Z(Z_1, Z_1)\mathcal{K}_Z(Z_2, Z_2)\big\} \\
&+ \mathbb{E}\big\{\big[\mathbb{E}_{\eta_1^1\eta_2^1}\mathcal{K}_X(\widehat{X}_2^{(1)}, \widetilde{X}_1^{(1)}) - \mathbb{E}_{\eta_1^1\eta_2^1}\mathcal{K}_X(\widehat{X}_2^{(1)}, \widehat{X}_1^{(1)})\big]^2\|n^{-\alpha}\mathcal{G}(S_1)\|_2^2\|\mathcal{R}_{n2}\|_2^2\mathcal{K}_Z(Z_1, Z_1)\mathcal{K}_Z(Z_2, Z_2)\big\} \\
&+ \mathbb{E}\big\{\big[\mathbb{E}_{\eta_1^1\eta_2^1}\mathcal{K}_X(\widetilde{X}_1^{(1)}, \widetilde{X}_2^{(1)}) - \mathbb{E}_{\eta_1^1\eta_2^1}\mathcal{K}_X(\widetilde{X}_1^{(1)}, \widehat{X}_2^{(1)})\big]^2\|n^{-\alpha}\mathcal{G}(S_1)\|_2^2\|\mathcal{R}_{n2}\|_2^2\mathcal{K}_Z(Z_1, Z_1)\mathcal{K}_Z(Z_2, Z_2)\big\}\Big\} \\
= &O(n^{-1-2\alpha-2\alpha_1}),
\end{aligned}$$

where the last equality follows from Assumption 13. So we have $\widehat{I}_1^{(1)} = O_p(n^{-1/2-\alpha-\alpha_1})$ under $H_{1n}$. For $\widehat{I}_1^{(4)}$, the conditional expectations of $\widehat{H}_1(X_j, X_k)\big[\mathcal{R}_{nj}\big]^\top [\mathcal{R}_{nk}]\mathcal{K}_Z(Z_j, Z_k)$ given $(\mathcal{R}_{nj}, S_j)$ or $(\mathcal{R}_{nk}, S_k)$ are zero, so $\widehat{I}_1^{(4)} = O_p(n^{-1-\alpha_1})$ by Assumption 13. Following similar approach as for $\widehat{I}_1^{(1)}$, we can show $\mathrm{Var}(\widehat{I}_1^{(3)}) = O(n^{-1-4\alpha-2\alpha_1})$, which implies $\widehat{I}_1^{(3)} = O_p(n^{-1/2-2\alpha-\alpha_1})$ under $H_{1n}$. So we have $\widehat{I}_1 = O_p(n^{-1/2-\alpha-\alpha_1} + n^{-1-\alpha_1})$ under $H_{1n}$.

Now consider $\widehat{I}_2$. Under $H_0$, $\widehat{I}_2 = O_p(n^{-1-\alpha_2})$ can be shown similar to the proof for $\widehat{I}_1$. Under $H_{1n}$, let

$$\widehat{H}_1^{(1)}(X_j, X_k) = \mathbb{E}_{\eta_k^1} \mathcal{K}_X(X_j, \widehat{X}_k^{(1)}) - \mathbb{E}_{\eta_k^1} \mathcal{K}_X(X_j, \widetilde{X}_k^{(1)}),$$

$$\widehat{H}_1^{(2)}(X_j, X_k) = \mathbb{E}_{\eta_j^1} \mathcal{K}_X(X_k, \widehat{X}_j^{(1)}) - \mathbb{E}_{\eta_j^1} \mathcal{K}_X(X_k, \widetilde{X}_j^{(1)}),$$

$$\widehat{H}_1^{(3)}(X_j, X_k) = \mathbb{E}_{\eta_j^1 \eta_k^1} \mathcal{K}_X(\widehat{X}_j^{(1)}, \widehat{X}_k^{(1)}) - \mathbb{E}_{\eta_j^1 \eta_k^1} \mathcal{K}_X(\widetilde{X}_j^{(1)}, \widetilde{X}_k^{(1)})$$

$$\widehat{H}_2^{(1)}(Y_j, Y_k) = Y_j^\top \widehat{g}_Y(Z_k) - Y_j^\top g_Y(Z_k),$$

$$\widehat{H}_2^{(2)}(Y_j, Y_k) = \widehat{g}_Y(Z_j)^\top Y_k - g_Y(Z_j)^\top Y_k,$$

$$\widehat{H}_2^{(3)}(Y_j, Y_k) = \widehat{g}_Y(Z_j)^\top \widehat{g}_Y(Z_k) - g_Y(Z_j)^\top g_Y(Z_k),$$

then we have

$$\widehat{I}_2 = \frac{1}{\frac{n}{2}(\frac{n}{2}-1)} \sum_{\substack{j \neq k \\ j,k \in \mathcal{J}^{(1)}}} U(X_j, X_k) \{\widehat{H}_2^{(1)}(Y_j, Y_k) + \widehat{H}_2^{(2)}(Y_j, Y_k) - \widehat{H}_2^{(3)}(Y_j, Y_k)\} \mathcal{K}_Z(Z_j, Z_k)$$

$$= \widehat{I}_2^{(1)} + \widehat{I}_2^{(2)} - \widehat{I}_2^{(3)},$$

$$\widehat{I}_2^{(1)} = \frac{1}{\frac{n}{2}(\frac{n}{2}-1)} \sum_{\substack{j \neq k \\ j,k \in \mathcal{J}^{(1)}}} U(X_j, X_k) \Big[g_Y(Z_j)\Big]^\top \Big[\widehat{g}_Y(Z_k) - g_Y(Z_k)\Big] \mathcal{K}_Z(Z_j, Z_k)$$

$$+ \frac{1}{\frac{n}{2}(\frac{n}{2}-1)} \sum_{\substack{j \neq k \\ j,k \in \mathcal{J}^{(1)}}} U(X_j, X_k) \Big[n^{-\alpha} \mathcal{G}(S_j)\Big]^\top \Big[\widehat{g}_Y(Z_k) - g_Y(Z_k)\Big] \mathcal{K}_Z(Z_j, Z_k)$$

$$+ \frac{1}{\frac{n}{2}(\frac{n}{2}-1)} \sum_{\substack{j \neq k \\ j,k \in \mathcal{J}^{(1)}}} U(X_j, X_k) \Big[\mathcal{R}_{nj}\Big]^\top \Big[\widehat{g}_Y(Z_k) - g_Y(Z_k)\Big] \mathcal{K}_Z(Z_j, Z_k)$$

$$= \widehat{I}_2^{(11)} + \widehat{I}_2^{(21)} + \widehat{I}_2^{(31)}.$$

For $\widehat{I}_2^{(11)}$, since the conditional expectations of the summand given $S_j$ or $S_k$ are zero, using Assumption 4, we can show $\widehat{I}_2^{(11)} = O_p(n^{-1-\alpha_2})$ (see the proof for $\widehat{I}_1$ under the null). For $\widehat{I}_2^{(31)}$, since the conditional expectations of the summand given $(\mathcal{R}_{nj}, S_j)$ or $S_k$ are zero, $\widehat{I}_1^{(31)} = O_p(n^{-1-\alpha_2})$ follows in the same way as $\widehat{I}_2^{(11)}$ (using part (b) of Assumption 4 and part (a) of Assumption 13). For $\widehat{I}_2^{(21)}$, we only have

$$\mathbb{E}\Big[U(X_j, X_k)\big[n^{-\alpha}\mathcal{G}(S_j)\big]^\top \big[\widehat{g}_Y(Z_k) - g_Y(Z_k)\big] \mathcal{K}_Z(Z_j, Z_k)\Big| S_j\Big]$$

$$= \mathbb{E}\Big[\mathbb{E}\big[U(X_j, X_k)\big| S_j, Z_k\big]\big[n^{-\alpha}\mathcal{G}(S_j)\big]^\top \big[\widehat{g}_Y(Z_k) - g_Y(Z_k)\big] \mathcal{K}_Z(Z_j, Z_k)\Big| S_j\Big]$$

$$= 0,$$

which implies

$$\mathrm{Var}(\widehat{I}_2^{(21)}) \leq \frac{C}{n} \mathbb{E}\Big[U(X_1, X_2)^2 \Big\{\big[n^{-\alpha}\mathcal{G}(S_1)\big]^\top \big[\widehat{g}_Y(Z_2) - g_Y(Z_2)\big]\Big\}^2 \mathcal{K}_Z(Z_1, Z_2)^2\Big]$$

$$\leq \frac{C}{n^{1+2\alpha}} \mathbb{E}\Big[U(X_1, X_2)^2 \|\mathcal{G}(S_1)\|_2^2 \|\widehat{g}_Y(Z_2) - g_Y(Z_2)\|_2^2 \mathcal{K}_Z(Z_1, Z_1)\mathcal{K}_Z(Z_2, Z_2)\Big].$$

By part (b) of Assumption 4 and part (a) of Assumption 13, $\widehat{I}_2^{(21)} = O_p(n^{-1/2-\alpha-\alpha_2})$ and we can conclude that $\widehat{I}_2^{(1)} = O_p(n^{-1/2-\alpha-\alpha_2} + n^{-1-\alpha_2})$. The proof for $\widehat{I}_2^{(2)} = O_p(n^{-1/2-\alpha-\alpha_2} + n^{-1-\alpha_2})$ is similar to $\widehat{I}_1^{(1)}$ and the proof for $\widehat{I}_2^{(3)} = O_p(n^{-1-\alpha_2})$ is similar to $\widehat{I}_1$ under the null. So we have $\widehat{I}_2 = O_p(n^{-1/2-\alpha-\alpha_2} + n^{-1-\alpha_2})$ under $H_{1n}$.

Now consider the term $\widehat{I}_3$. Denote $\widetilde{W}_1(j, k) = \mathbb{E}\{\widehat{H}_1(X_j, X_k)|Z_j, Z_k\}$ and $\widetilde{W}_2(j, k) = \mathbb{E}\{\widehat{H}_2(Y_j, Y_k)|Z_j, Z_k\}$, then we

have

$$
\widehat{I}_3 = \frac{1}{\frac{n}{2}(\frac{n}{2}-1)} \sum_{\substack{j \neq k \\ j,k \in \mathcal{J}^{(1)}}} \big[\widehat{H}_1(X_j, X_k) - \widetilde{W}_1(j,k)\big]\big[\widehat{H}_2(Y_j, Y_k) - \widetilde{W}_2(j,k)\big]\mathcal{K}_Z(Z_j, Z_k)
$$

$$
+ \frac{1}{\frac{n}{2}(\frac{n}{2}-1)} \sum_{\substack{j \neq k \\ j,k \in \mathcal{J}^{(1)}}} \widetilde{W}_1(j,k)\big[\widehat{H}_2(Y_j, Y_k) - \widetilde{W}_2(j,k)\big]\mathcal{K}_Z(Z_j, Z_k)
$$

$$
+ \frac{1}{\frac{n}{2}(\frac{n}{2}-1)} \sum_{\substack{j \neq k \\ j,k \in \mathcal{J}^{(1)}}} \big[\widehat{H}_1(X_j, X_k) - \widetilde{W}_1(j,k)\big]\widetilde{W}_2(j,k)\mathcal{K}_Z(Z_j, Z_k)
$$

$$
+ \frac{1}{\frac{n}{2}(\frac{n}{2}-1)} \sum_{\substack{j \neq k \\ j,k \in \mathcal{J}^{(1)}}} \widetilde{W}_1(j,k)\widetilde{W}_2(j,k)\mathcal{K}_Z(Z_j, Z_k)
$$

$$
= \widehat{I}_3^{(1)} + \widehat{I}_3^{(2)} + \widehat{I}_3^{(3)} + \widehat{I}_3^{(0)}. \tag{24}
$$

We now show that $\widehat{I}_3^{(2)}, \widehat{I}_3^{(3)} = O_p(n^{-1/2-(\alpha_1+\alpha_2)})$ and $\widehat{I}_3^{(0)} = O_p(n^{-2(\alpha_1+\alpha_2)})$ under both $H_0$ and $H_{1n}$. Note that $\mathbb{E}|\widehat{I}_3^{(0)}| \leq \mathbb{E}\{|\widetilde{W}_1(1,2)||\widetilde{W}_2(1,2)|\sqrt{\mathcal{K}_Z(Z_1, Z_1)\mathcal{K}_Z(Z_2, Z_2)}\}$ and

$$
|\widetilde{W}_1(1,2)| = \Big|\mathbb{E}\Big\{\big\langle \mathcal{K}_X(\cdot, \widehat{X}_2^{(1)}) - \mathcal{K}_X(\cdot, \widetilde{X}_2^{(1)}), \mathcal{K}_X(\cdot, \widetilde{X}_1^{(1)}) - \mathcal{K}_X(\cdot, \widehat{X}_1^{(1)})\big\rangle_{\mathbb{H}_X} \Big| Z_1, Z_2\Big\}\Big|
$$

$$
\leq \big\|\mathbb{E}_{\eta_2^1}\mathcal{K}_X(\cdot, \widehat{X}_2^{(1)}) - \mathbb{E}_{\eta_2^1}\mathcal{K}_X(\cdot, \widetilde{X}_2^{(1)})\big\|_{\mathbb{H}_X} \big\|\mathbb{E}_{\eta_1^1}\mathcal{K}_X(\cdot, \widetilde{X}_1^{(1)}) - \mathbb{E}_{\eta_1^1}\mathcal{K}_X(\cdot, \widehat{X}_1^{(1)})\big\|_{\mathbb{H}_X}.
$$

Similarly, $|\widetilde{W}_2(1,2)| \leq \|g_Y(Z_2) - \widehat{g}_Y(Z_2)\|_2 \|g_Y(Z_1) - \widehat{g}_Y(Z_1)\|_2$, so by part (b) of Assumption 4, we have $\mathbb{E}|\widehat{I}_3^{(0)}| = O(n^{-2(\alpha_1+\alpha_2)})$. For $\widehat{I}_3^{(2)}$, we have $\mathbb{E}\widehat{I}_3^{(2)} = 0$ and

$$
\mathrm{Var}(\widehat{I}_3^{(2)}) \leq \frac{C}{n^2}\mathbb{E}\Big\{\widetilde{W}_1(1,2)^2\big[\widehat{H}_2(Y_1, Y_2) - \widetilde{W}_2(1,2)\big]^2\mathcal{K}_Z(Z_1, Z_2)^2\Big\}
$$

$$
+ \frac{C}{n}\Big|\mathbb{E}\Big\{\widetilde{W}_1(1,2)\widetilde{W}_1(1,3)\big[\widehat{H}_2(Y_1, Y_2) - \widetilde{W}_2(1,2)\big]\big[\widehat{H}_2(Y_1, Y_3) - \widetilde{W}_2(1,3)\big]\mathcal{K}_Z(Z_1, Z_2)\mathcal{K}_Z(Z_1, Z_3)\Big\}\Big|.
$$

By part (b) of assumption 4 and equation (25), $\mathbb{E}\Big\{\widetilde{W}_1(1,2)^2\big[\widehat{H}_2(Y_1, Y_2) - \widetilde{W}_2(1,2)\big]^2\mathcal{K}_Z(Z_1, Z_2)^2\Big\} = O(n^{-2(\alpha_1+\alpha_2)})$. Since

$$
\widehat{H}_2(Y_1, Y_2) - \widetilde{W}_2(1,2) = \big[Y_1 - g_Y(Z_1)\big]^\top\big[\widehat{g}_Y(Z_2) - g_Y(Z_2)\big] + \big[Y_2 - g_Y(Z_2)\big]^\top\big[\widehat{g}_Y(Z_1) - g_Y(Z_1)\big]
$$

$$
\leq \big\|Y_1 - g_Y(Z_1)\big\|_2\big\|\widehat{g}_Y(Z_2) - g_Y(Z_2)\big\|_2 + \big\|Y_2 - g_Y(Z_2)\big\|_2\big\|\widehat{g}_Y(Z_1) - g_Y(Z_1)\big\|_2, \tag{25}
$$

$$
\widehat{H}_2(Y_1, Y_3) - \widetilde{W}_2(1,3) \leq \big\|Y_1 - g_Y(Z_1)\big\|_2\big\|\widehat{g}_Y(Z_3) - g_Y(Z_3)\big\|_2 + \big\|Y_3 - g_Y(Z_3)\big\|_2\big\|\widehat{g}_Y(Z_1) - g_Y(Z_1)\big\|_2
$$

by part (a) and (b) of Assumption 4, we have

$$
\mathbb{E}\Big\{\widetilde{W}_1(1,2)\widetilde{W}_1(1,3)\big[\widehat{H}_2(Y_1, Y_2) - \widetilde{W}_2(1,2)\big]\big[\widehat{H}_2(Y_1, Y_3) - \widetilde{W}_2(1,3)\big]\mathcal{K}_Z(Z_1, Z_2)\mathcal{K}_Z(Z_1, Z_3)\Big\} = O(n^{-2(\alpha_1+\alpha_2)}),
$$

which implies $\widehat{I}_3^{(2)} = O_p(n^{-1/2-(\alpha_1+\alpha_2)})$ and $\widehat{I}_3^{(3)} = O_p(n^{-1/2-(\alpha_1+\alpha_2)})$ follows similarly. For $\widehat{I}_3^{(1)}$, since $\mathbb{E}\widehat{I}_3^{(1)} = 0$

under $H_0$, we can show $\widehat{I}_3^{(1)} = O_p(n^{-1/2-(\alpha_1+\alpha_2)})$ in the same way as $\widehat{I}_3^{(2)}$. Under $H_{1n}$,

$$\widehat{I}_3^{(1)} = \frac{1}{\frac{n}{2}(\frac{n}{2}-1)} \sum_{\substack{j\neq k \\ j,k\in\mathcal{J}^{(1)}}} \left[\widehat{H}_1(X_j,X_k) - \widetilde{W}_1(j,k)\right]\left[Y_j - g_Y(Z_j)\right]^\top \left[\widehat{g}_Y(Z_k) - g_Y(Z_k)\right]\mathcal{K}_Z(Z_j,Z_k)$$

$$+ \frac{1}{\frac{n}{2}(\frac{n}{2}-1)} \sum_{\substack{j\neq k \\ j,k\in\mathcal{J}^{(1)}}} \left[\widehat{H}_1(X_j,X_k) - \widetilde{W}_1(j,k)\right]\left[Y_k - g_Y(Z_k)\right]^\top \left[\widehat{g}_Y(Z_j) - g_Y(Z_j)\right]\mathcal{K}_Z(Z_j,Z_k)$$

$$= \widehat{I}_3^{(11)} + \widehat{I}_3^{(21)}$$

$$\widehat{I}_3^{(11)} = \frac{1}{\frac{n}{2}(\frac{n}{2}-1)} \sum_{\substack{j\neq k \\ j,k\in\mathcal{J}^{(1)}}} \left[\widehat{H}_1(X_j,X_k) - \widetilde{W}_1(j,k)\right]\left[n^{-\alpha}\mathcal{G}(S_j)\right]^\top \left[\widehat{g}_Y(Z_k) - g_Y(Z_k)\right]\mathcal{K}_Z(Z_j,Z_k)$$

$$+ \frac{1}{\frac{n}{2}(\frac{n}{2}-1)} \sum_{\substack{j\neq k \\ j,k\in\mathcal{J}^{(1)}}} \left[\widehat{H}_1(X_j,X_k) - \widetilde{W}_1(j,k)\right]\left[\mathcal{R}_{nj}\right]^\top \left[\widehat{g}_Y(Z_k) - g_Y(Z_k)\right]\mathcal{K}_Z(Z_j,Z_k)$$

$$= \widehat{I}_3^{(111)} + \widehat{I}_3^{(112)}. \tag{26}$$

We now show $\widehat{I}_3^{(11)} = O_p(n^{-1/2-(\alpha_1+\alpha_2)}+n^{-(\alpha+\alpha_1+\alpha_2)})$ (same result holds for $\widehat{I}_3^{(21)}$ and the proof is omitted). Note that

$$\mathbb{E}|\widehat{I}_3^{(111)}| \leq \frac{1}{n^\alpha}\mathbb{E}\Big\{\|\mathcal{G}(S_1)\|_2\|\widehat{g}_Y(Z_2) - g_Y(Z_2)\|_2\|\mathcal{K}_X(X_1,\cdot) - \mathbb{E}_{\eta_1^1}\mathcal{K}_X(\widetilde{X}_1^{(1)},\cdot)\|_{\mathbb{H}_X}$$

$$\cdot \|\mathbb{E}_{\eta_2^1}\mathcal{K}_X(\widehat{X}_2^{(1)},\cdot) - \mathbb{E}_{\eta_2^1}\mathcal{K}_X(\widetilde{X}_2^{(1)},\cdot)\|_{\mathbb{H}_X}\sqrt{\mathcal{K}_Z(Z_1,Z_1)\mathcal{K}_Z(Z_2,Z_2)}\Big\}$$

$$+ \frac{1}{n^\alpha}\mathbb{E}\Big\{\|\mathcal{G}(S_1)\|_2\|\widehat{g}_Y(Z_2) - g_Y(Z_2)\|_2\|\mathcal{K}_X(X_2,\cdot) - \mathbb{E}_{\eta_2^1}\mathcal{K}_X(\widetilde{X}_2^{(1)},\cdot)\|_{\mathbb{H}_X}$$

$$\cdot \|\mathbb{E}_{\eta_1^1}\mathcal{K}_X(\widehat{X}_1^{(1)},\cdot) - \mathbb{E}_{\eta_1^1}\mathcal{K}_X(\widetilde{X}_1^{(1)},\cdot)\|_{\mathbb{H}_X}\sqrt{\mathcal{K}_Z(Z_1,Z_1)\mathcal{K}_Z(Z_2,Z_2)}\Big\}$$

$$= O(n^{-(\alpha+\alpha_1+\alpha_2)})$$

by Jensen's inequality and part (b) of Assumptions 4 and 13, so we have $\widehat{I}_3^{(111)} = O_p(n^{-(\alpha+\alpha_1+\alpha_2)})$. Since $\mathbb{E}\widehat{I}_3^{(112)} = 0$, following similar approach as for $\widehat{I}_3^{(2)}$ we can show $\widehat{I}_3^{(112)} = O_p(n^{-1/2-(\alpha_1+\alpha_2)})$ and conclude that $\widehat{I}_3^{(11)} = O_p(n^{-1/2-(\alpha_1+\alpha_2)}+n^{-(\alpha+\alpha_1+\alpha_2)})$. So part(b) is proved. $\qquad\square$

