# OpenReview forum: "Testing Conditional Mean Independence Using Generative Neural Networks"
_ICML.cc/2025/Conference — ICML 2025 poster_

### Official Review · Reviewer_WEw2 · 2025-03-12

**Overall Recommendation:** 4

**Summary:**

The paper introduces a new nonparametric test for conditional mean independence (CMI) that leverages deep generative neural networks to estimate conditional mean embeddings. The proposed method uses a novel population measure based on RKHS embeddings and constructs a test statistic in a multiplicative form that is robust to the slower convergence rates of nonparametric nuisance parameter estimators. To mitigate estimation errors, the authors combine sample splitting and cross-fitting with a generative moment matching network (GMMN) for sampling from the conditional distribution of covariates. The paper provides comprehensive theoretical guarantees (including asymptotic size control and power properties under local alternatives) and supports its claims via extensive simulation studies and real-world imaging applications (facial expression recognition and age estimation).

**Claims And Evidence:**

The central claims are well-supported by theoretical analysis and empirical results. In particular:

- The theoretical claims about precise asymptotic size control are supported by rigorous proofs and verified in simulation studies showing empirical sizes close to nominal levels. The claim of detecting local alternatives is validated through theoretical analysis in Theorem 5 and empirical power evaluations in simulations. However, the theoretical results rely on certain technical assumptions (e.g., on the decay rates of estimation errors) might limit the generality of the results in practice.

- The claim of strong empirical performance in high-dimensional settings is demonstrated through comprehensive simulations against multiple baseline methods and by experiment on real-world imaging applications.

**Essential References Not Discussed:**

While the paper cites a wide range of relevant literature, it would benefit from discussing recent advances in generative modeling (GANs and diffusion models) for conditional distribution estimation (i.e. trying different design of the generative network $\hat G$), for example:

1. Athey, S., Imbens, G. W., Metzger, J., and Munro, E. Using Wasserstein generative adversarial networks for the design of Monte Carlo simulations. Journal of Econometrics, 2021.

2. Baptista, R., Hosseini, B., Kovachki, N. B., and Marzouk, Y. Conditional sampling with monotone GANs: From generative models to likelihood-free inference. arXiv preprint arXiv:2006.06755, 2020.

3. Shi, Y., De Bortoli, V., Deligiannidis, G., and Doucet, A. Conditional simulation using diffusion Schrödinger bridges. In Uncertainty in Artificial Intelligence, pp. 1792–1802. PMLR, 2022.

4. Nguyen, B., Nguyen, B., Nguyen, H. T., & Nguyen, V. A. Generative conditional distributions by neural (entropic) optimal transport. In Proceedings of the 41st International Conference on Machine Learning (pp. 37761-37775), 2024.

**Experimental Designs Or Analyses:**

The simulation studies (Examples A1 and A2) are well-designed and cover a range of scenarios (both linear and nonlinear models, and sparse versus dense alternatives). The experimental analyses also include a comparison with multiple state-of-the-art methods.

However, while the results demonstrate clear benefits of the proposed test in terms of both size control and power, I believe a more detailed ablation study—particularly regarding the sensitivity to hyperparameter choices and kernel bandwidth selection—could strengthen the empirical section further.

**Methods And Evaluation Criteria:**

This work proposes to use generative models to approximate conditional distributions for RKHS-based testing, which is novel to overcome challenges in high-dimensional nonparametric estimation. The evaluations on both synthetic experiments (with clear sparse and dense alternatives) and applications on real imaging datasets are appropriate and provide convincing evidence of the method’s effectiveness.

**Other Comments Or Suggestions:**

n/a

**Other Strengths And Weaknesses:**

Good:

- The paper is in general clear to follow, though can be dense with notation in the first few pages.


Need to address:

- There is limited discussion of hyperparameter sensitivity (kernel bandwidths, network architectures)

**Questions For Authors:**

1. How does computational complexity scale with the dimensionality of X, Y, Z, and sample size n compared to existing methods?
2. Have you explored alternative conditional generative models beyond GMMNs (e.g., conditional GANs, score-based diffusion model, see Essential References Not Discussed)? How might they affect test performance?
3. How robust is your method to model misspecification when estimating conditional mean functions in highly nonlinear relationships?

**Relation To Broader Scientific Literature:**

I believe the authors have done good work on literature review within the CMI testing literature, clearly identifying limitations of existing methods.

**Theoretical Claims:**

The paper contains several theoretical contributions, with proofs detailed in the supplementary material. I reviewed the main steps in the proofs of Theorems 4, 5, and 6. Under the given assumptions (e.g., Assumptions 7 and 9), the arguments appear sound. However, some of those technical assumptions seem to be quite strong. Clarification on the practical implications of these assumptions would be beneficial.

---

> ### Author Rebuttal · Authors · 2025-03-29
>
> We greatly appreciate your valuable comments, which have helped lead to a much-improved manuscript. In the following, we present our point-by-point responses to your questions and will take into account all your suggestions in a revised version of our manuscript.
>
> **Generality and practical implications of the assumptions.** For a detailed discussion on Assumptions 7 and 9, please refer to our reply to Reviewer **tmTe**. In summary, our proposed CMI test is fully nonparametric, and Assumptions 7 and 9 do not impose explicit restrictions on the data distribution (e.g., boundedness, continuity, or sub-Gaussianity), enhancing its practical applicability. Furthermore, the double robustness property of the test statistic allows for mild assumptions on the error decay rates of nuisance parameters. For example, if we assume that $(Y, Z)$ follows a linear regression model, then $g_Y$ can be estimated at the $n^{-1/2}$ rate, which implies that the conditional distribution $P_{X|Z}$ only needs to be consistently estimated without strict rate requirements.
>
> Following your comments, we will include a brief discussion in the revised version of our paper, highlighting the generality and practical implications of these assumptions, particularly the more technical ones.
>
> **Hyperparameter sensitivity.** For the sensitivity to network hyperparameters, please refer to our reply to Reviewer **tmTe**. Regarding the choice of kernel bandwidths, we followed the median heuristic in our manuscript, as it is widely used in kernel-based tests. To further evaluate the sensitivity to bandwidth selection, we conducted additional simulations for Example A1 with a fixed sample size $n=400$ using bandwidths determined by either the mean pairwise distance or the “$\gamma$th quantile heuristic” for $\gamma $ $\in$ {25%, 75%}. Specifically, the bandwidth for $\mathcal{K}_X$ was set as the mean or $\gamma$th quantile of {$|X_j - X_k|_1 : j,k \in [n]$}, with similar choices for $\mathcal{K}_Z$. The empirical sizes for the test using the mean, 25\% quantile, and 75\% quantile bandwidths are 7\%, 6.8\%, and 7\%, respectively. The size-adjusted power under the sparse alternative are 98.6\%, 98.4\%, and 98.6\%, while under the dense alternative, it remains 100\% in all cases. These results indicate that the test’s empirical performance remains stable across different bandwidth selection methods.
>
> **Computational complexity.**  Given the trained neural networks and assuming Gaussian or Laplacian kernel functions, our statistic resembles the average of two U-statistics of degree two, with its value depending on pairwise distances between samples. As a result, the computational complexity scales linearly with the dimensions $(d_X + d_Y + d_Z)$ and quadratically with the sample size $n$. For comparison, the computational complexity of pMIT$_M$ (Cai et al., 2024) and DSP$_M$ (Dai et al., 2022), both DNN-based CMI tests focused on univariate $Y$, scales linearly with $n$. The network training complexity is $O(E \cdot n \cdot P)$, where $E$ is the number of epochs and $P$ is the total number of trainable parameters.
>
> In light of this comment, we will include a discussion on computational complexity in the revised version.
>
> **Alternative GNN structure and discussion on recent advances in generative modeling.** At the early stages of this paper, we experimented with conditional GANs similar to those in Shi et al. (2021) to approximate $P_{X|Z}$. The empirical performance of the test was comparable to the current approach using GMMN. However, a key drawback of GANs is their longer training time, as both the generator and discriminator must be trained simultaneously. In contrast, GMMN has a more efficient training process and yields a test statistic whose empirical performance closely matches the oracle statistic; see Table 2 in Section 3. We will incorporate a more detailed discussion of recent advancements in generative modeling, particularly those highlighted in your review, in the revised version of our paper.
>
> **Robustness to model misspecification.** A key strength of our proposed test is its fully nonparametric nature, as it does not assume a specific parametric form for the mean functions. Thanks to the universal approximation properties of neural networks, model misspecification is not a concern asymptotically if the network width increases with the sample size. However, in practice, network structures are fixed, which may introduce approximation errors.
>
> Fortunately, the double robustness property of our statistic mitigates sensitivity to these approximation errors, making our method more reliable than approaches that lack this property. As demonstrated by the simulation results in Section 3 for both linear and nonlinear models, our test's performance is comparable to the oracle test.

---

### Official Review · Reviewer_tmTe · 2025-03-12

**Overall Recommendation:** 3

**Summary:**

This paper proposes a novel statistical method to conditional mean independence (CMI) testing. First, the authors introduce a new population-level CMI measure and develop a bootstrap-based hypothesis testing framework that employs generative neural networks to approximate conditional mean functions. Its test statistic is constructed to reduce the influence of nonparametric estimation errors, ensuring asymptotic precision. The proposed method performs well in high-dimensional settings, and supports multivariate responses. Finally, the experiments on simulated and real-world data demonstrate the effectiveness of the proposed method.

##  update after rebuttal
 After reviewing the rebuttal addressed to me and those for other reviewers, I am willing to maintain my score.

**Claims And Evidence:**

Yes, the claims made in the submission appear to be supported by clear and convincing evidence.

**Essential References Not Discussed:**

No, the paper includes all essential and relevant references.

**Experimental Designs Or Analyses:**

Yes, the experimental designs and analyses are sound.

**Methods And Evaluation Criteria:**

Yes, the proposed hypothesis testing framework makes sense for the CMI testing  task.

**Other Comments Or Suggestions:**

See Weaknesses.

**Other Strengths And Weaknesses:**

Strengths:
- The method is shown to control Type I error while maintaining nontrivial  power.

- The proofs are clear and sound.

- Comparisons against existing CMI tests highlight superior empirical  performance of the proposed method.

Weaknesses:
- The method requires to train multiple deep neural networks, which increases computational cost.

- The theoretical results depends on some strong assumptions, such as assumption 7. The rationality of the assumptions in this paper should be discussed.

**Questions For Authors:**

No.

**Relation To Broader Scientific Literature:**

This paper proposes a novel framework for CMI testing problem with strong theoretical guarantee.

**Theoretical Claims:**

Yes. I check some proofs,  including Theorems 4，5，6.

---

> ### Author Rebuttal · Authors · 2025-03-29
>
> We greatly appreciate your valuable comments, which have helped lead to a much-improved manuscript. In the following, we present our point-by-point responses to your questions and will take into account all your suggestions in a revised version of our manuscript.
>
> **Computational cost.** Due to the sample splitting and cross-fitting framework, we need to train four neural networks (two GNNs and two DNNs) to construct our statistic, which is analogous to the two-split pMIT$_M$ test proposed in Cai et al. (2024). Regarding computation time, it takes 41.0 seconds for our method to complete one Monte Carlo simulation for Example A1 with a sample size of $n=400$. This is longer than competing methods but of the same order as pMIT$_M$; please refer to our reply to Reviewer **KpXH** for more details on computation time.
> The computation time for training the neural networks depends on the complexity of the network, which is primarily determined by the data structures. For our numerical results in Section 3, the network structures used are relatively simple (with only one hidden layer). These simple structures are easy to train and still yield satisfactory empirical performance. Importantly, our method does not rely on a specific machine learning algorithm or network structure. As long as the estimation error meets the requirements in Assumption 7, any new or different machine learning techniques and network architectures can be used to reduce the training cost.
>
> **Rationality of the assumptions.** Part (a) of Assumption 7 ensures that the (conditional) mean embeddings into the RKHSs, as well as the operator $\Sigma$, are well-defined. This assumption is commonly used in the literature of kernel-based conditional (mean) independence testing and holds for bounded kernels such as the Gaussian and Laplacian kernels.
>
> Part (b) of Assumption 7 requires the estimation errors of the neural networks to decay to zero at rates $n^{-\alpha_1}$ and $n^{-\alpha_2}$ for $\alpha_1, \alpha_2 \in (0, \infty)$ such that $\alpha_1 + \alpha_2 > 1/2$. Similar rate requirements appear in Cai et al. (2024) and Lundborg et al. (2024), where "black-box" estimators such as DNNs are used. As shown in Stone (1982), the minimax nonparametric regression decay rate for $\mathbb{E} [ |g_Y(Z) - \hat g_Y(Z)|^2] $ is $n^{2p/(2p+d_Z)}$. When estimating $g_Y$ with DNNs, Bauer and Kohler (2019) demonstrated that the decay rate can be $n^{2p/(2p+d^\ast)}$, where $p$ is the smoothness parameter and $d^\ast$ represents the intrinsic dimensionality.
> Regarding the estimation error of the conditional mean embedding of $P(X|Z)$, our requirement is actually less restrictive than the assumptions on the total variation distance between $P(X|Z)$ and its estimator (see Remark 8 in Appendix C), which were used in Shi et al. (2021). Shi et al. (2021) argue that their assumption holds in a wide range of settings, with examples provided in Berrett et al. (2019). Moreover, due to the double robustness property of our test statistic, we do not impose explicit constraints on the individual estimation errors of $g_Y$ and the mean embedding of $P_{X|Z}$. Instead, we only require their product to decay faster than $n^{-1/2}$. Notably, when $g_Y$ is sufficiently smooth, $\alpha_2$ can approach $1/2$, allowing $\alpha_1$ to remain very small to accommodate complex and high-dimensional distributions of $P_{X|Z}$ (e.g., when both $X$ and $Z$ are images).
> For Assumption 9, we allow the residual vector $Y - \mathbb{E}[Y|Z]$ to vary under local alternatives. This is a more general setting than in nonparametric regression models, where the residual is assumed to remain the same as under the null hypothesis; see Remark 10 in Appendix C.
> As suggested, we will include a brief discussion of these assumptions, particularly the more technical ones, in the revised version of our paper.
>
> **Reference**
>
> Stone, C. J. (1982): "Optimal global rates of convergence for nonparametric regression." Ann. Statist.
>
> Bauer and Kohler (2019): "On deep learning as a remedy for the curse of dimensionality in nonparametric regression." Ann. Statist.
>
> Berrett et al. (2019): "The Conditional Permutation Test for Independence While Controlling for Confounders." Journal of the Royal Statistical Society Series B: Statistical Methodology

---

### Official Review · Reviewer_KpXH · 2025-03-14

**Overall Recommendation:** 4

**Summary:**

This develops a novel method to test for conditional mean independence that works well in high-dimensions, gives asymptotic size control, and has nontrivial power against local alternatives. This depends on using deep learning to learn g_y and g_x using a bootstrap sample. They then use the test statistic in the unnumbered equation before equation 2. They then generate data under a null distribution and use that to approximate the p-value. They show theoretically  They then test the performance on synthetic data with regression examples along with real examples testing whether masking affects prediction accuracy.

**Claims And Evidence:**

The authors justify the claims that this method works well in testing CMI in high dimensions, both theoretically and empirically. In my opinion their evidence is sufficient but some of the comparisons are disappointing. The authors don't really explore the quality of neural network approximations, but given that proving theoretical guarantees even for more rudimentary tasks is difficult I don't fault them for that. But I would have preferred some material in the appendix seeing how some of the neural network parameters changing affects results. Also would be nice to have a sense of how long these methods take to run.

**Essential References Not Discussed:**

The references cited seem quite extensive.

**Experimental Designs Or Analyses:**

I checked the analyses. I suppose this is the most appropriate category to point out that from what I can tell, the paper doesn't analyze how the Monte Carlo method performs along with the number of bootstrap samples. This is important, because one possibility is that this is just a very high-powered poorly sized test but the errors in the approximations introduced address that.

**Methods And Evaluation Criteria:**

Hypothesis testing is straightforward for evaluation criteria. Given that this is designed for high dimensions, high dimensional regression and image questions make sense. One of the problems I do have with this paper is that none of the competitor methods are used on the image data from what I can tell. I would have assumed that it would be a straightforward improvement.

**Other Comments Or Suggestions:**

Please don't put assumptions in the appendix. It's annoying to have to look there. Also it's weird to have assumption 7 and 9 but not 1-6 and 8.

**Other Strengths And Weaknesses:**

None that are not previously described

**Questions For Authors:**

Unless I missed it why is the effect of choice of B not evaluated?

Are there direct comparisons on the image data?

**Relation To Broader Scientific Literature:**

This paper improves on extensive literature on conditional mean independence. Many kernel methods struggle with high dimensional data. While most tests have size guarantees, at least of the most pertinent methods also struggle at maintaining power at a parametric rate, or at least being backed with theoretical guarantees.

**Theoretical Claims:**

I checked the theoretical claims to the best of my ability.

---

> ### Author Rebuttal · Authors · 2025-03-29
>
> We greatly appreciate your valuable comments, which have significantly contributed to improving the quality of our manuscript. Below, we provide point-by-point responses to your questions and will incorporate all of your suggestions in the revised version of our manuscript.
>
> **Sensitivity to network parameters and computation time.** To evaluate how changes in neural network parameters affect the performance of the proposed test, we repeated the simulation for Example A1 with a fixed sample size of $n = 400$ but varied the network configurations. In the first case, we increased the number of hidden layers in both networks to two, keeping all other parameters unchanged. In the second case, we reduced the number of nodes in $ \mathbb{\widehat G}$ and $ \widehat g_Y$ by half (to 56 and 128, respectively).
>
> The results were consistent with those in Table 2 of the manuscript:
> empirical sizes: 5.8\% and 6\% (close to the original);
> size-adjusted power (sparse alternative): 98.6\% and 99.8\%;
> size-adjusted power (dense alternative): 100\% in both cases.
> This suggests that the test’s performance is robust to moderate changes in key network parameters.
>
> For other hyperparameters (e.g., batch size, learning rate), we used the Optuna automated search package to optimize them by minimizing the loss function defined in Equation (4) of Appendix A.
>
> Regarding computation time, our method takes 41.0 seconds to complete one Monte Carlo simulation for Example A1 with sample size $n=400$ using NVIDIA T4 GPU,  which is of the same order as the DNN based test pMIT$_M$. Detailed computation times for our competitors under the same setting are listed below.
>
> - **On Intel Xeon CPU.** DSP: 3.21 seconds; DSP$_M$: 14.9 seconds; DNN-pMIT: 5.13 seconds; DNN-pMIT$_e$: 5.13 seconds; DNN-pMIT$_M$: 25.0 seconds; DNN-pMIT$_e$$_M$: 25.0 seconds.
> - **On 11th gen intel core i7-11800h CPU.** XGB-pMIT: 0.167 seconds; XGB-pMIT$_e$: 0.167 seconds; XGB-pMIT$_M$: 1.56 seconds; XGB-pMIT$_e$$_M$: 1.56 seconds; PCM: 0.20 seconds; PCM$_M$: 1.07 seconds; VIM: 6.667 seconds.
>
> **Competitor methods on image data application.** We applied some competing methods to the image datasets in our initial submission, but due to space limitations, we have included some details in the appendix. For the image data application in Section 4.1, we compared our method with the DSP$_M$ approach developed by Dai et al. (2022), where a similar application was examined. For the application in Section 4.2, we compared with the pMIT$_M$ method introduced by Cai et al. (2024). The p-values for these comparison methods are included in Figures 1 and 3, and a detailed comparison is provided in Appendix B.
>
>
> **Sensitivity to numbers of Monte Carlo data and bootstrap sample.** We have conducted additional simulations to investigate how the number of Monte Carlo synthetic data ($M$) and the bootstrap number ($B$) influence the performance of our test. The results suggest that both the size and power of our testing procedure remain robust against these tuning parameters. Specifically, we repeated the simulation for Example A1 with $n=400$, varying $M$ in {$5,20,60$} and $B$ in {$200,500$}. The results are presented in the following table.
> |                                | M=5 B=200 | M=5 B=500 | M=20 B=200 | M=20 B=500 | M=60 B=200 | M=60 B=500 |
> |-----------------------|-----------|-----------|-------------|------------|-------------|---------------------|
> | Empirical Size                 | 6.2       | 6.6       | 5.8        | 6.2        | 6.8        | 6.4        |
> | Power under Sparse Alternative | 99.2      | 99        | 99.6       | 99.6       | 98.6       | 99         |
> | Power under Dense Alternative  | 100       | 100       | 100        | 100        | 100        | 100        |
>
> **Numbering and location of assumptions.**  Originally, we chose to place the assumptions in the Appendix to stay within page limits and to keep the focus on explaining the core ideas of our proposed test without introducing excessive technical details. For the next version, we plan to maintain them in the Appendix due to space constraints, but will add a few sentences in the main text to briefly summarize these assumptions. Regarding assumption labeling, we initially used the default formatting from the ICML LaTeX template, but we would be happy to relabel them separately from theorems and remarks if preferred.

---

### Decision · Program_Chairs · 2025-05-01

**Decision:**

Accept (poster)

**Comment:**

This paper introduces a new nonparametric test for conditional mean independence (CMI), that works well in high-dimensional settings, by leveraging deep generative neural networks to estimate the conditional mean.  A new population-level CMI measure is proposed and its sample version serves as the test statistic. To mitigate the estimation errors a combination of sample splitting and cross-fitting with a generative moment matching network (GMMN) for sampling from the conditional distribution of covariates. The theory provides theoretical guarantees for both the asymptotic size control and power under local alternatives. Numerical experiments support the theory.

One concern is the lack of comparisons on image data with competing methods. The authors addressed this in their response.

Overall, this work makes a significant contribution to the literature of testing CMI.